# `FlexTok`: Resampling Images into 1D Token Sequences of Flexible Length

**Roman Bachmann** [* 1 2]  **Jesse Allardice** [* 1]  **David Mizrahi** [* 1]  **Enrico Fini** [1]  **Oğuzhan Fatih Kar** [2]
**Elmira Amirloo** [1]  **Alaaeldin El-Nouby** [1]  **Amir Zamir** [2]  **Afshin Dehghan** [1]

https://flextok.epfl.ch

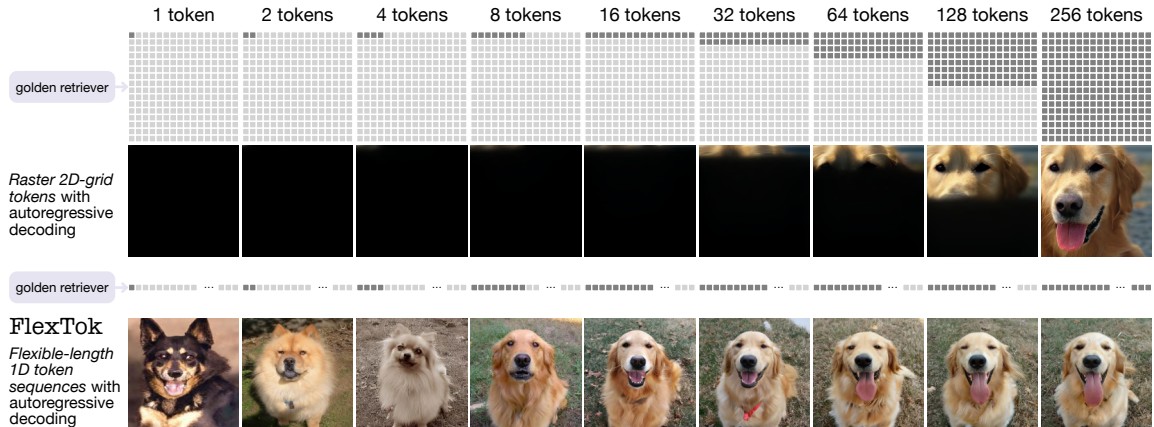

*Figure 1.* **Comparison of partial sequence generation: Raster-scan 2D-grid tokenizer vs. `FlexTok`.** `FlexTok` resamples images into a 1D sequence of discrete tokens of flexible length, describing images in a coarse-to-fine manner. When training autoregressive (AR) models on `FlexTok` token sequences, the class conditioning (here *"golden retriever"*) can be satisfied by generating as few as 8 tokens, whereas AR models trained on 2D tokenizer grids (here, LlamaGen (Sun et al., 2024)) need to always generate all tokens, no matter the complexity of the condition or image.

## Abstract

We introduce `FlexTok`, a tokenizer that projects 2D images into variable-length, ordered 1D token sequences. For example, a $256 \times 256$ image can be resampled into anywhere from 1 to 256 discrete tokens, hierarchically and semantically compressing its information. By training a rectified flow model as the decoder and using nested dropout, `FlexTok` produces plausible reconstructions regardless of the chosen token sequence length. We evaluate our approach in an autoregressive generation setting using a simple GPT-style Transformer. On ImageNet, this approach achieves an FID < 2 across 8 to 128 tokens, outperforming

TiTok and matching state-of-the-art methods with far fewer tokens. We further extend the model to support to text-conditioned image generation and examine how `FlexTok` relates to traditional 2D tokenization. A key finding is that `FlexTok` enables next-token prediction to describe images in a coarse-to-fine *"visual vocabulary"*, and that the number of tokens to generate depends on the complexity of the generation task.

## 1. Introduction

Image generation has advanced significantly in both quality and inference speed. Tokenization plays a crucial role in reducing the computational cost of training generative models (Van Den Oord et al., 2017; Esser et al., 2021). Recently, autoregressive (AR) image generation has shown competitive performance when scaled to billions of parameters (Sun et al., 2024).

Generative models such as diffusion (Rombach et al., 2022; Peebles & Xie, 2023), masked (Chang et al., 2022; 2023),

*Equal contribution  [1]Apple  [2]Swiss Federal Institute of Technology Lausanne (EPFL). Correspondence to: Roman Bachmann <roman.bachmann@epfl.ch>, Jesse Allardice <jallardice@apple.com>, David Mizrahi <d_mizrahi@apple.com>.

*Proceedings of the $42^{nd}$ International Conference on Machine Learning*, Vancouver, Canada. PMLR 267, 2025. Copyright 2025 by the author(s).

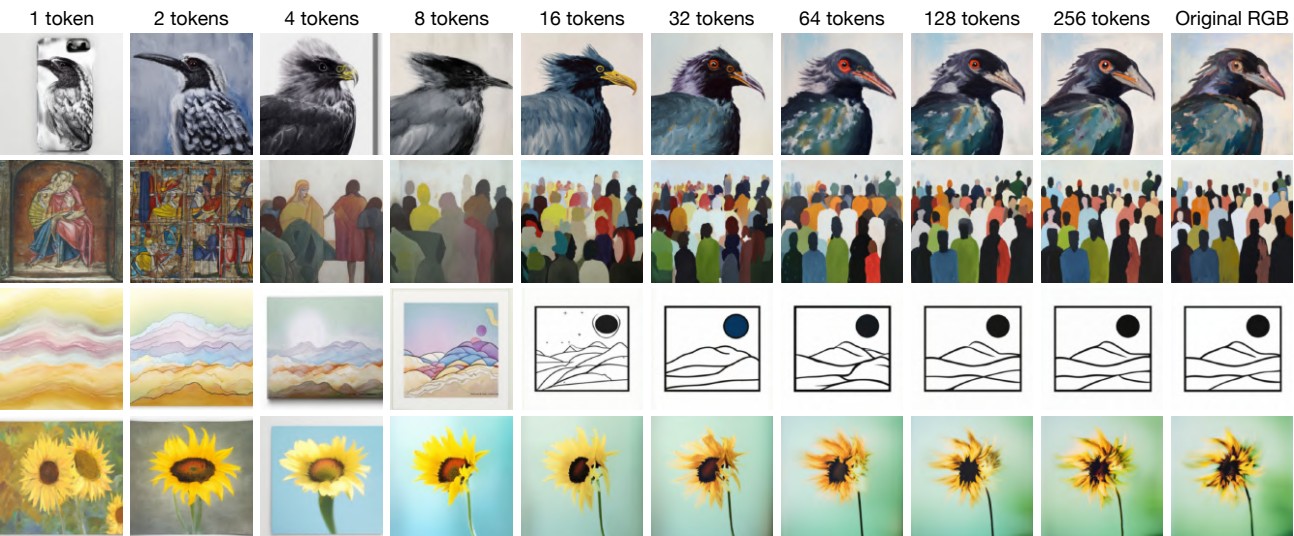

*Figure 2.* **Reconstruction examples using `FlexTok d18-d28` trained on DFN.** Notice how most of the images' semantic and geometric content is captured by fewer than 16 tokens. The first tokens already capture the high-level semantic concepts (e.g., *gray bird, people in colorful garments, mountain scene, yellow flower*), while more tokens are required to reconstruct more intricate scene details (e.g., *position and clothing of every person, brushstroke placement, etc.*). To showcase out-of-distribution reconstruction, we generated the original images using Midjourney v6.1 (Midjourney, 2024).

and autoregressive (Chen et al., 2020; Yu et al., 2022) models traditionally operate on 2D grids of continuous or discrete tokens. These representations maintain strong spatial alignment with the original pixel patches (Esser et al., 2021; Mentzer et al., 2023), but this means that the representation size is proportional to the image size, rather than dependent on the complexity of the image. Recent work (Yu et al., 2024a) has demonstrated the benefits of 1D tokenization schemes for improving computational efficiency while maintaining competitive quality. Similar to existing 2D grid tokenization schemes, these 1D token sequences are fixed-length, *regardless of the underlying image complexity*. In other words, no matter whether an image simply depicts a single object or a busy scene with intricate details, it is encoded into the same number of tokens. In turn, a conditional image generator trained to predict these tokens must always produce the full set of tokens, no matter the semantic complexity of the condition.

In this paper, we present `FlexTok`, a novel variable-length 1D tokenizer that is able to encode images into an ordered and content-dependent sequence of tokens. As visualized in Figure 2, earlier tokens capture the most high-level semantic and geometric information, while additional tokens progressively add finer details. The variable-length sequences, i.e. truncated subsequences of length 1, 2, 4, ..., 256, can be decoded into plausible images using an end-to-end trained rectified flow decoder. Figure 1 contrasts the rigid raster-scan ordering of traditional 2D tokenizers (top) with our hierarchical 1D approach that enables coarse-to-fine generation (bottom).

Our contributions are as follows:

**Flexible-length tokenization:** We present a 1D tokenization approach that can compress images into anywhere from 1 to 256 tokens. Through a combination of nested dropout and causal attention masking, our tokens are *naturally ordered* from coarse to fine details, enabling high-quality reconstruction even with very few tokens while providing progressively more detailed representations as more tokens are used.

**A new *"visual vocabulary"* to describe images:** Unlike classical 2D tokenizers which describe image content at each x,y location, `FlexTok` uses a hierarchical approach in which high-level aspects such as semantic and geometric concepts naturally emerge to be ordered first, while subsequent tokens add finer details. This allows a coarse conditioning to be fulfilled with relatively few predicted tokens, while a more detailed condition requires generating a larger number of tokens. For example, as shown in Figure 7, ImageNet-1k classes can be generated with as few as 8 tokens. In contrast, text conditions, which may involve more complex and varied prompts, can benefit from predicting up to 256 tokens.

**End-to-end trained rectified flow decoder:** To enable high-quality image reconstruction across varying token lengths, we incorporate a rectified flow objective into our tokenizer decoder. This proves essential for maintaining reconstruction quality even at extreme compression rates.

## 2. Related Work and Background

The primary role of an image tokenizer in generation tasks is to create compressed latent representations of images. Generative models need to learn a distribution over their outputs, but doing so directly in continuous, high-dimensional spaces (e.g., raw image pixels) is challenging. Compression helps by mapping images to a more compact latent space that removes imperceptible details (Rombach et al., 2022), while discretization enables the generative model to output per-token categorical distributions that can be sampled from.

Vector-quantized autoencoders (VQ-VAEs) (Van Den Oord et al., 2017; Razavi et al., 2019; Esser et al., 2021) have become a standard framework for learning these discrete representations. VQ-VAE models operate through three core components: (1) an encoder $Enc$ that maps input images $\boldsymbol{X} \in \mathbb{R}^{H \times W \times 3}$ to $D$-channel latent embeddings $\boldsymbol{Z} = Enc(\boldsymbol{X}) \in \mathbb{R}^{h \times w \times D}$ (usually $h \ll H$ and $w \ll W$), (2) a quantizer $Quant$ that maps these continuous latents to discrete codes from a learned codebook, and (3) a decoder $Dec$ that reconstructs the image $\hat{\boldsymbol{X}} = Dec(Quant(\boldsymbol{Z}))$. This approach has proven versatile, finding applications in image (Chang et al., 2022; 2023; Li et al., 2022), audio (Baevski et al., 2019), and video generation (Villegas et al., 2022; Hu et al., 2023; Kondratyuk et al., 2023), novel view synthesis (Yan et al., 2021), and large-scale multimodal pretraining (Lu et al., 2022; 2023; Mizrahi et al., 2023; Bachmann et al., 2024; Chameleon, 2024; Wang et al., 2024a).

Various improvements to this framework have been proposed, exploring different architectures (Yu et al., 2021), objective functions (Esser et al., 2021; Hu et al., 2023), codebook structures (Zhang et al., 2023; Yu et al., 2023), and the use of diffusion models as decoders (Shi et al., 2022; Mizrahi et al., 2023; Xu et al., 2024; Zhao et al., 2024). A key advancement is finite scalar quantization (FSQ) (Mentzer et al., 2023), which replaces the learned codebook with a projection to a small-dimensional latent space and quantizes values using fixed bins along each dimension. This approach maintains reconstruction quality while being simpler to implement and train.

Instead of maintaining a 2D spatial structure in the latent space, TiTok (Yu et al., 2024a) creates 1D representations using learned register tokens $\boldsymbol{R} \in \mathbb{R}^{K \times D}$ (Darcet et al., 2023). During encoding, these K register tokens are concatenated with image patch embeddings $\boldsymbol{P} \in \mathbb{R}^{h \times w \times D}$ for processing in a ViT encoder. The encoder output retains only the register token embeddings as $\boldsymbol{Z}_{1D}$, which capture the image content in a compact sequence. For reconstruction, the image patches are replaced with a grid of $h \times w$ learnable mask tokens $\boldsymbol{M} \in \mathbb{R}^{h \times w \times D}$. These mask tokens, guided by the quantized register embeddings, are transformed by the decoder into the reconstructed image: $\hat{\boldsymbol{X}} = Dec(Quant(\boldsymbol{Z}_{1D}) \oplus \boldsymbol{M})$.

Although effective at producing 1D token sequences, TiTok requires a two-stage training process and represents images using a fixed number of tokens. Compared to TiTok, FlexTok shows a superior reconstruction and generation quality (see Figure 4 and Table 1). To address the issue of the token sequence length depending entirely on the image height and width instead of its complexity, a range of concurrent works have proposed adaptive tokenization methods. While FlexTok can compress images into as little as a single token, ElasticTok's (Yan et al., 2024) FSQ tokenizer is limited to a minimum of 256 tokens and ALIT (Duggal et al., 2024) to 32 tokens. Generation is a key use case of tokenizers, and unlike ElasticTok, ALIT, or One-D-Piece (Miwa et al., 2025), we show in Section 5 that training on FlexTok tokens can produce strong generative models. Compared to our focus on AR models, ViLex (Wang et al., 2024b) and CAT (Shen et al., 2025) focus more on learning continous tokens to use in diffusion models. We discuss further differences with respect to FlexTok in detail in Appendix D.

## 3. Method

FlexTok is an autoencoder with a discrete 1D bottleneck, see Figure 3 for an overview. A ViT encoder maps 2D image patches into a 1D sequence using register tokens (Darcet et al., 2023; Yu et al., 2024a). The registers are discretized using FSQ (Mentzer et al., 2023), and then used as conditioning for a rectified flow model tasked with reconstructing the image. We use causal attention among register tokens followed by nested dropout (Rippel et al., 2014) to induce an ordering in the bottleneck representation. Paired with a rectified flow decoder, this design enables the model to decode any nested subset of tokens into plausible images.

### 3.1. 1D tokenization with a rectified flow decoder

**Register encoder and discrete bottleneck.** Similar to TiTok (Yu et al., 2024a), the encoder uses register tokens to resample the patched 2D VAE latents into a 1D sequence of discrete tokens. Specifically, we concatenate VAE latent patches with a set of learnable register tokens that act as read-write storage for the encoder. After encoding, the register tokens act as the bottleneck representation for the autoencoder, while the encoded patches are discarded. The register tokens are quantized into discrete tokens using FSQ. Instead of pixels, we operate on latent representations of a VAE-GAN, similar to SDXL's VAE (Podell et al., 2023), to abstract away the perceptual compression from this investigation. Using the VAE-GAN latent space enables us to greatly simplify our design space and reduce computational requirements as training generative models on pixels is expensive (Rombach et al., 2022).

*Figure 3.* **FlexTok overview. Stage 1**: FlexTok resamples 2D VAE latents to a 1D sequence of discrete tokens using a ViT with registers (Darcet et al., 2023). The FSQ-quantized bottleneck (Mentzer et al., 2023) representation is used to condition a rectified flow model that decodes and reconstructs the original images. FlexTok learns ordered token sequences of flexible length by applying nested dropout (Rippel et al., 2014) on the register tokens. **Stage 2**: We train class- and text-conditional autoregressive Transformers to predict 1D token sequences in a coarse-to-fine manner. As more tokens are predicted, the generated image becomes more specific, encoding high-level concepts first *(e.g., presence of a car)* followed by finer details *(e.g., car shape, brand, color)*.

**Rectified flow decoder.** The decoder's purpose is to generate perceptually plausible images conditioned on the compressed latent representations. When the bottleneck size is small, training a decoder with only a reconstruction loss can result in blurry reconstructions. To mitigate this, FlexTok uses a rectified flow model that is conditioned on the quantized register tokens $Quant(\boldsymbol{Z}_{1D})$ by concatenating them with noised VAE latent patches $\boldsymbol{X}_t = (1-t)\boldsymbol{X}_0 + t\epsilon$, where $\boldsymbol{X}_0$ are the clean VAE latents, $t \in [0,1]$ is a random time step, and $\epsilon \sim \mathcal{N}(0,1)$ is a random sample from the noise distribution. The decoder's objective is to predict the flow $\boldsymbol{U} = \epsilon - \boldsymbol{X}_0$ given the partially noised VAE latents and the encoded register tokens. We minimize the rectified flow loss $\mathcal{L}_{\text{RF}} = ||\hat{\boldsymbol{U}} - (\epsilon - \boldsymbol{X}_0)||^2$, given the predicted flow $\hat{\boldsymbol{U}} = Dec(Quant(\boldsymbol{Z}_{1D}) \oplus \boldsymbol{X}_t)$. We find that adding a REPA (Yu et al., 2024b) inductive bias loss $\mathcal{L}_{\text{REPA}}$ between an intermediate decoder layer and DINOv2-L (Oquab et al., 2023) features significantly improves convergence time and downstream generation performance (see Table 3 and Fig.12). The total FlexTok loss we optimize is $\mathcal{L}_{\text{FlexTok}} = \mathcal{L}_{\text{RF}} + \lambda_{\text{REPA}} \cdot \mathcal{L}_{\text{REPA}}$, with $\lambda_{\text{REPA}} = 1$. See Section 4 and Appendix C.2 for more implementation details.

### 3.2. Learning ordered 1D token sequences of flexible length

1D tokenizers like TiTok (Yu et al., 2024a) require training different tokenizers for each desired number of register tokens. As shown in the concurrent work ALIT (Duggal et al., 2024), fixed token sequence lengths do not take into account the inherent complexity of an image. Simple images can be compressed into as few as 32 tokens, while more complex ones require more tokens to faithfully reconstruct them. Neither TiTok nor ALIT demonstrate flexible tokenization below 32 tokens, and their image reconstruction performance deteriorates when the compression degree is high, see Figure 4 and Appendix K.3 for visual comparisons to FlexTok. Since we target AR generation with FlexTok, we additionally introduce a nested left-to-right ordering structure that naturally aligns with next-token prediction. We propose two techniques to introduce a 1D ordering and variable length into the token sequences.

**Nested dropout.** We train FlexTok to produce an ordered representation by randomly dropping the encoded register tokens in a nested manner during training (Rippel et al., 2014; Kusupati et al., 2022; Cai et al., 2024) . Specifically, given a register token sequence of length $K$, we randomly sample the number of units to keep $K_{keep} \in \{1, ..., K\}$ and remove the $K - K_{keep}$ last tokens by masking them out. By training the tokenizer in this manner, the encoder learns to compress the image content into the register tokens in an ordered manner, while the rectified flow decoder learns to reconstruct images given the variably-sized token sequences. As we show in Figure 5, this design enables FlexTok to capture the most important aspects of images in very few tokens. Simplistic images require few tokens to be faithfully compressed, while complex ones require longer token sequences. We note here that this ordering is not handcrafted and emerges purely from performing nested dropout on register tokens, and computing rectified flow and REPA losses. We mainly ablate two strategies of dropping tokens in a nested manner. In the first, we sample the number of tokens to keep uniformly, as described above. In the second variant, we uniformly sample them from an exponentially increasing set, e.g. $K_{keep} \in \{1, 2, 4, 8, 16, ..., K\}$. The latter variant addresses an issue with uniform nested dropout, where the last register tokens are passed to the decoder very rarely, meaning they are effectively trained for only a small fraction of gradient updates.

**Causal attention masks.** Orthogonal to the use of nested dropout, adding a causal attention mask to the encoded registers enforces a causal dependency structure between them (Ge et al., 2023). In this setting, the encoded image patches can all attend to each other but not to the registers, the register tokens can attend to all the patches, but the $i$-th register token may only attend to the $j$-th register token if $i \geq j$. In addition, the use of causal masks enables users to more efficiently encode images if they know ahead of time that they only want to keep $K_{keep} \ll K$ tokens.

### 3.3. Autoregressive image generation

To evaluate different design choices of `FlexTok` and compare to relevant baselines, we measure both reconstruction and generation performance. To that end, we train autoregressive Transformers to perform class-conditional generation on ImageNet-1k (Russakovsky et al., 2014), following LlamaGen (Sun et al., 2024), and text-to-image generation on DFN-2B (Fang et al., 2023).

## 4. Implementation

We break down the implementation into three distinct stages. In **Stage 0**, we train VAE models (Rombach et al., 2022) with continuous latents to perceptually compress images into 2D token grids. In **Stage 1**, we then train `FlexTok` tokenizers to resample these continuous 2D token grids into discrete 1D token sequences of flexible length. Finally, in **Stage 2**, we train autoregressive class-to-image and text-to-image models to evaluate the effectiveness of `FlexTok` in generative tasks.

**Stage 0: VAE training.** The rectified flow decoder (see Section 3.1) is a key design element of `FlexTok` that enables decoding arbitrary token subsequences. However, training such models directly in pixel space is computationally expensive (Rombach et al., 2022). The goal of this stage is therefore to facilitate **Stage 1** modeling by perceptually compressing images into more compact representations. We follow the architecture of the SDXL VAE (Podell et al., 2023), and train versions with 4, 8, and 16 channels on the DFN dataset (Fang et al., 2023). All subsequent experiments use the 16 channel VAE with a downsampling factor of 8. Please see Appendix C.1 for more VAE training details, and Table 2 for ablations on the number of latent channels.

**Stage 1: `FlexTok` training.** The `FlexTok` architecture consists of a Transformer encoder and decoder using a maximum of 256 registers tokens. After applying a 6-dimensional FSQ (Mentzer et al., 2023) bottleneck with levels [8, 8, 8, 5, 5, 5] (for an effective vocabulary size of 64 000), the encoded registers are randomly truncated using nested dropout. The decoder is a rectified

flow model that receives noised VAE latent patches and the (randomly masked) registers as input, and is tasked to predict the flow. We use adaLN-zero (Peebles & Xie, 2023) to condition the patches and registers separately on the current timestep, and REPA (Yu et al., 2024b) with DINOv2-L (Oquab et al., 2023) features to speed up convergence. We use 2x2 patchification in both the `FlexTok` encoder and decoder, which combined with the VAE's 8x downsampling yields a total 16x downsampling from pixels to patch tokens. All models are trained at a resolution of 256x256 pixels. The encoder and decoder dimensions $w$ are parameterized using their respective depths $d$, using a fixed aspect ratio of 64, i.e. $w = 64 \cdot d$. We train three `FlexTok` versions with different encoder and decoder sizes (separated by a hyphen), `d12-d12`, `d18-d18` and `d18-d28` after sweeping optimal hyperparameters at a small scale using $\mu$P (Yang et al., 2022). Depending on the downstream use case, we train `FlexTok` models on ImageNet-1k (Russakovsky et al., 2014) for subsequent class-conditional generation, and on DFN (Fang et al., 2023) for subsequent text-to-image modeling. See Appendix C.2 for further `FlexTok` implementation and training details.

**Stage 2: AR Transformer training.** Our autoregressive Transformer follows LlamaGen's Llama-inspired architecture (Sun et al., 2024; Touvron et al., 2023), using prenormalization with RMSNorm (Zhang & Sennrich, 2019) and a SwiGLU feedforward (Shazeer, 2020). Since our tokens lack a 2D grid structure, we use learned absolute positional embeddings instead of 2D RoPE (Su et al., 2024).

For class conditioning, we add a learned class embedding to an [SOI] token (Tian et al., 2024) and concatenate it with the image token sequence. The AR model, ranging from 49M to 1.3B parameters, predicts the token sequence from the `FlexTok` tokenizer. To enable comparisons with LlamaGen and TiTok, we train without $\mu$P.

For text-conditioned generation, our AR decoder crossattends to text embeddings from FLAN-T5-XL (Chung et al., 2024), projected to the model dimension via an MLP (Chen et al., 2023). We scale these text-conditioned AR models up to 3B parameters, using $\mu$P to maintain consistent behavior across scales.

Following standard practice, we employ conditioning dropout during training to enable classifier-free guidance at inference. For text-conditioned models, this involves randomly replacing text inputs with an empty string. For additional implementation and training details, see Appendix C.3.

## 5. Experiments

In this section, we experimentally evaluate the reconstruction performance of `FlexTok` across different numbers of

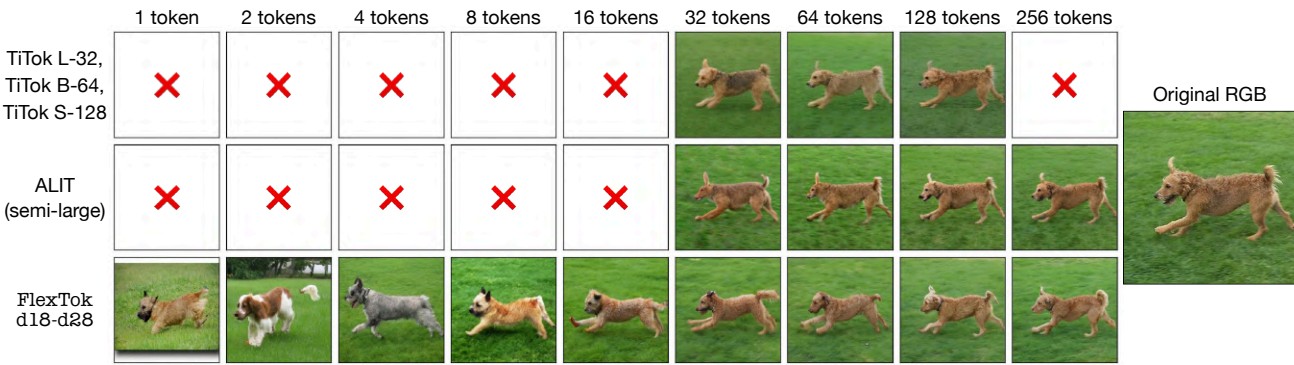

*Figure 4.* **Image reconstruction comparison between three different TiTok (Yu et al., 2024a) models, ALIT (Duggal et al., 2024), and FlexTok.** Compared to other 1D tokenizers, `FlexTok` is able to tokenize images in a highly semantic and ordered manner, all the way down to a single token, and all in a single model. For more visual comparisons, see Appendices K.1 to K.3.

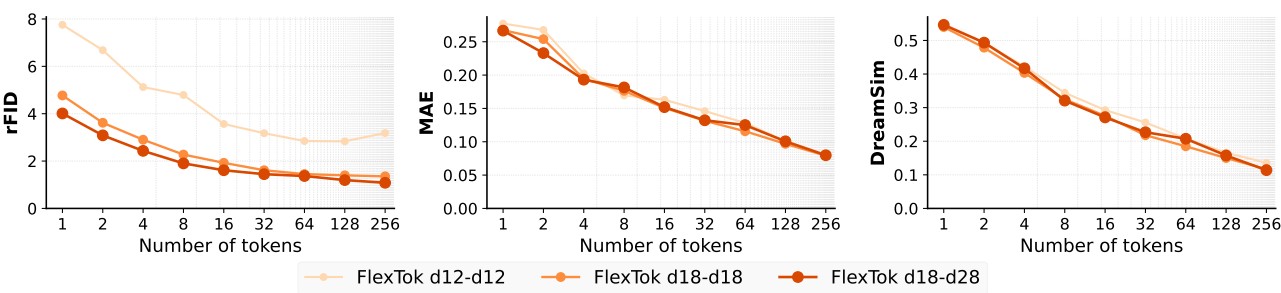

*Figure 5.* **FlexTok rate-distortion tradeoff.** We show ImageNet-1k **reconstruction** metrics for three different `FlexTok` sizes. The more tokens used, the closer the reconstructions get to the original RGB images. Scaling the tokenizer size significantly improves reconstruction FID, but is not as crucial in terms of MAE and DreamSim score. For each of the different `FlexTok` model sizes we use the optimal inference hyperparameters detailed in Appendix F. We show additional reconstruction metrics in Table 8.

tokens, assess its applicability for class-conditional and text-conditional image generation, and compare it to relevant baselines. We show that `FlexTok` can effectively compress images into 1D sequences of flexible length, enabling a novel *"visual vocabulary"* where images can be specified and generated in a coarse-to-fine manner.

### 5.1. Flexible-length tokenization

We demonstrate `FlexTok`'s variable-rate tokenization capability by evaluating its reconstruction performance on nested token sequences of different lengths. We perform comparisons using `FlexTok` models trained on ImageNet-1k, testing them on 256x256 pixel crops from the validation set (Russakovsky et al., 2014). In Figure 5 we measure reconstruction metrics (rFID, MAE, DreamSim) for $K_{keep} \in \{1, 2, 4, 8, 16, ..., 256\}$ tokens. `FlexTok` is able to generate plausible images, as measured through the rFID against the ImageNet-1k validation set, with as little as a single token.

As shown in Figure 2 with a `FlexTok d18-d28` model trained on DFN, and in Figure 4 with a `FlexTok d18-d28` model trained on ImageNet-1k, reconstructions

using the first few tokens capture high-level semantic features. As more tokens are used, both the alignment with the original image becomes more fine-grained and the image-wise reconstruction metrics (MAE and DreamSim) improve rapidly.

Please see Appendices K.1 to K.3 for additional reconstruction examples and comparisons. For linear probing experiments on the token sequences, see Appendix B.

We find the following properties particularly noteworthy: (1) By performing nested dropout on the registers, *a hierarchy emerges in which high-level concepts are ordered first.* (2) Through training `FlexTok` with a Rectified Flow decoder, *any token subsequence can be decoded into a plausible image.* (3) The token sequences specify a *distribution over images* that gets *more and more specific* with more tokens, see Figures 34 and 35 for visual examples.

### 5.2. Coarse-to-fine generation with increasing specificity

As shown in Section 5.1, `FlexTok` compresses images into ordered token sequences. This naturally leads us to explore the implications of *predicting* these sequences for autoregressive image generation. By training class- and text-

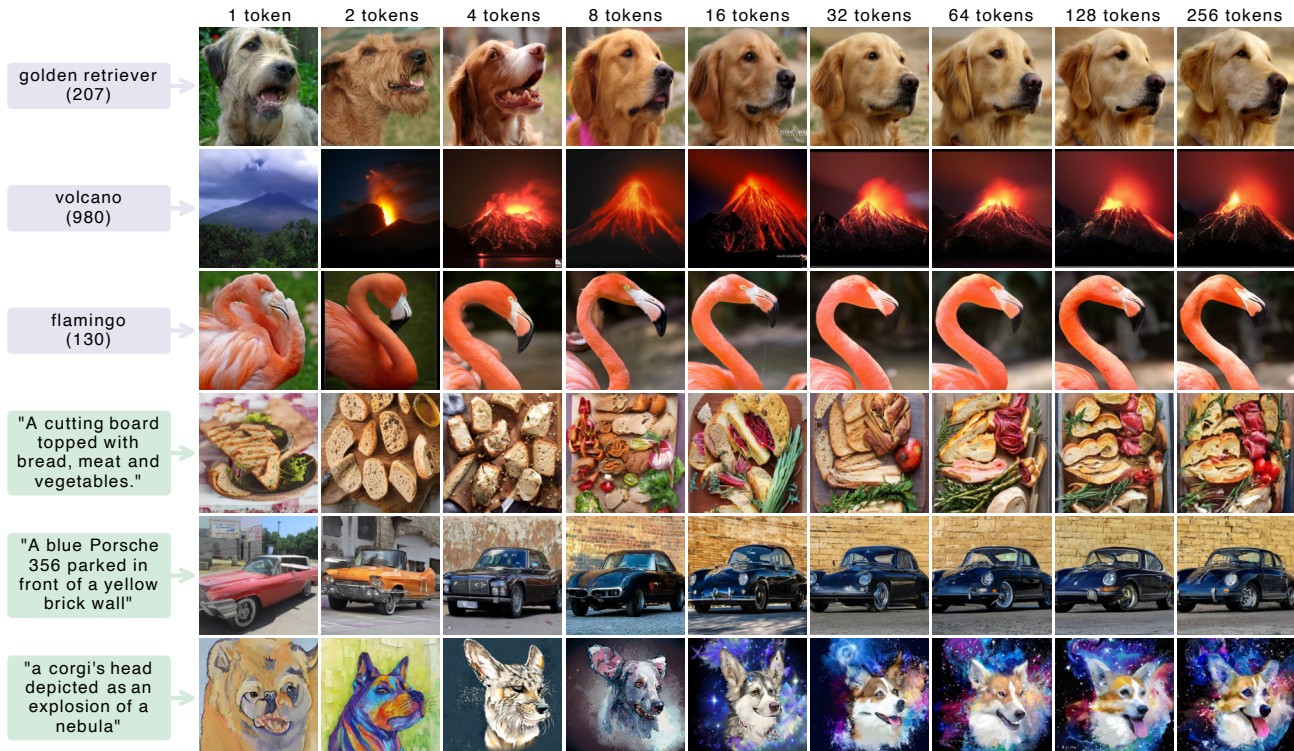

*Figure 6.* **Image generation examples with varying numbers of tokens.** Images generated with both class (top 3 rows) and text conditioning (bottom 3 rows) demonstrate that FlexTok-based models achieve high quality all the way down to a single token, and all within a single model. The conditioning alignment strengthens as more tokens are generated. For example with the prompt *"a corgi's head depicted as an explosion of a nebula"*, the first two tokens capture the high-level concept of *a artistic depiction of a dog*, while adding more tokens adds in further details such as the *dog breed* and the *nebula background*. For more visualizations, see Appendices K.4 and K.5.

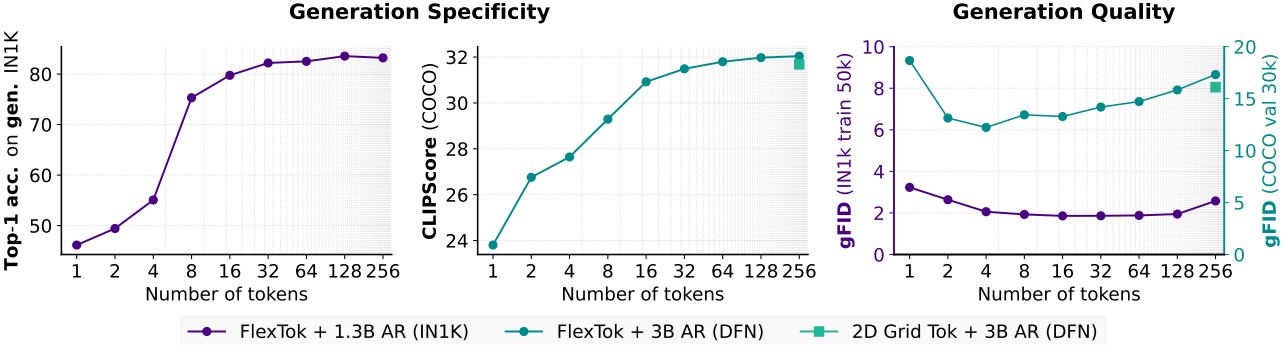

*Figure 7.* **Conditioning alignment and generation quality vs. number of tokens. Left:** For class-conditional generation with a 1.3B AR model, we compute DINOv2-L (Oquab et al., 2023) top-1 accuracy on generated images conditioned on ImageNet-1k class labels. **Center:** For text-conditional generation with a 3B AR model, we show CLIPScore relative to input prompts from the COCO 30k validation set (Lin et al., 2014), using a CLIP base model . **Right:** We measure class-conditional gFID on ImageNet-1k[*], and text-conditional gFID on COCO. The AR models use guidance scales of 1.0 (no guidance) and 2.5, respectively. We follow the optimal inference parameters described in Appendices F and G.

[*] *Note.* For evaluation of the class-conditioned image generation results, we follow the common practice of measuring the generation FID (gFID) of 50K generated samples relative to the reference statistics calculated *over the entire training split* of the ImageNet-1k dataset (Dhariwal & Nichol, 2021).

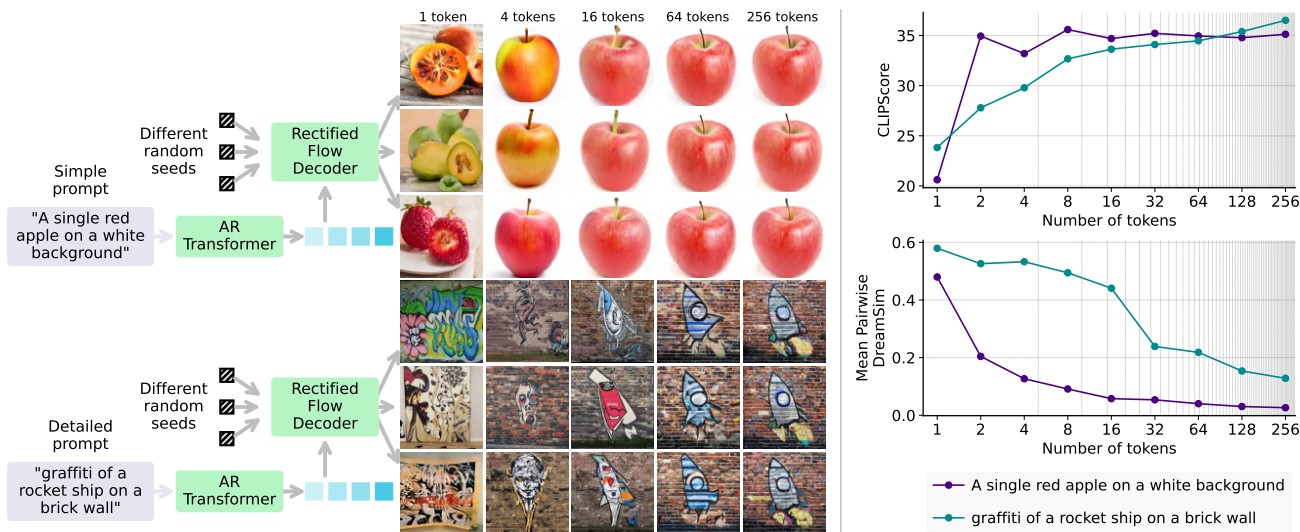

*Figure 8.* **Image generation with simple and detailed prompts.** Images generated with `FlexTok`-based models show that the number of tokens needed to fulfill the conditioning depends on prompt complexity. For a simple prompt, the desired image is achieved with as few as 4-16 tokens (as measured by CLIPScore), and semantic variation between different decoded images (as measured by pairwise DreamSim scores) vanishes quickly. In contrast, a detailed prompt requires the full 256-token sequence to fully meet the conditioning and shows greater variation at lower token counts as the `FlexTok` rectified flow decoder compensates for missing details. For each prompt, the `FlexTok` tokens are generated just once using the AR Transformer and then decoded with 10 random seeds in the rectified flow decoder.

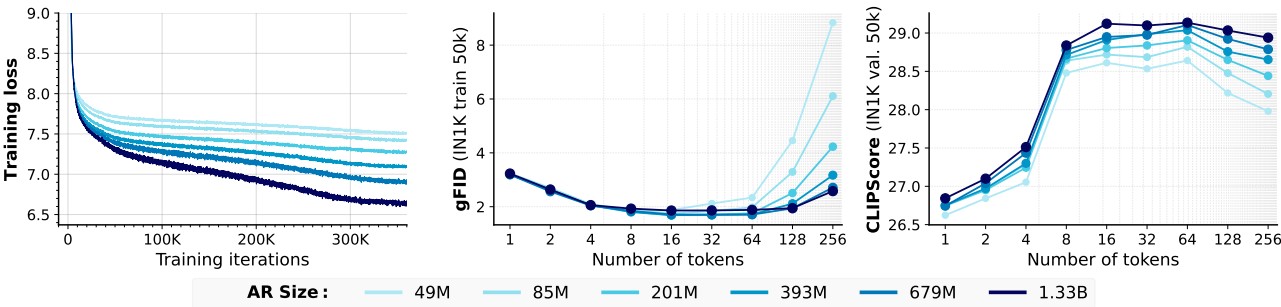

*Figure 9.* **Class-conditioned AR model scaling.** We show training loss, gFID and image generation CLIPScore values for the class-conditional models with the `FlexTok d18-d28` tokenizer. We calculate the CLIPScore using the text label of the classes from the ImageNet-1k validation set, and do not use classifier-free guidance for the AR Transformer (CFG scale = 1.0). We show additional generation metrics for the 1.33B AR model in Table 9.

conditional models, we find that `FlexTok` token sequences act as a *"visual vocabulary"*, allowing autoregressive models to describe images with increasing levels of specificity. Unlike conventional autoregressive models that generate images in a fixed raster-scan order on 2D token grids, our approach enables progressive refinement of image details. We also observe a clear *relationship between conditioning complexity and token requirements*. Simple conditions, like ImageNet class labels, can be fulfilled with as few as 16 tokens, while more complex ones, like open-ended text prompts, benefit from generating up to 256 tokens.

In Figure 6, we show that images generated by class- or text-conditional models become increasingly specific to their conditioning as more tokens are produced. Our quantitative results in Figure 7 confirm this trend, showing that alignment between the conditioning signal and the generated images improves with higher token counts. We measure alignment using DINOv2-L (Oquab et al., 2023) classification accuracy for class conditioning and CLIPScore for text conditioning. Notably, text-image alignment continues to improve as additional tokens are generated, whereas classification accuracy tapers off after the first few tokens and plateaus around 32 (Figures 7 and 9). Furthermore, we observe that generation quality remains consistent across all token sequence lengths, as measured by gFID, which we attribute to the strength of our rectified flow decoder.

Similarly, Figure 8 shows that the number of tokens needed to generate prompt-aligned images varies significantly based

on prompt complexity. Simple prompts like "a red apple" can produce satisfactory results with just 4 to 16 tokens, while detailed prompts like "graffiti of a rocket ship" benefit from using the full 256-token sequence. When fewer tokens are used, the model can still produce realistic images, but at the cost of greater variation between different random seeds. This variation decreases much more quickly for simple prompts than for detailed ones as token count increases, suggesting a fundamental relationship between prompt complexity and token requirements.

### 5.3. Scaling autoregressive model size

As shown in Figure 7, using our largest autoregressive models, the alignment between condition and generations generally improves with more predicted tokens. In this section, we investigate the scaling behavior of autoregressive class-conditional models trained on `FlexTok d18-d28` tokens, focusing on how model size impacts image-caption alignment and image fidelity. In Figure 9, we show that increasing the AR model size consistently improves the measured training loss. However, for the prediction of the first few (1-8) tokens, the generation FID (gFID) values for the decoded images are effectively independent of the autoregressive model size, indicating that these short initial parts of the sequences are easily learned even by small models. In contrast, for long sequences ($> 128$ tokens) the task becomes more challenging, and performance scales strongly with model size for both gFID and CLIPScore.

This highlights an important trade-off between the `FlexTok` and AR model: as more tokens are generated, the AR model takes on a larger role in shaping the image. With few tokens, `FlexTok`'s flow decoder drives most of the generation, resulting in low gFID even with smaller AR models. However, as the token count increases, an equally powerful AR model is needed to maintain strong performance.

### 5.4. System-level comparison

In Table 1, we compare our `FlexTok` models against several relevant baselines. We evaluate the tokenizers by training autoregressive models to perform ImageNet-1k class-conditional generation. Compared to previous 1D approaches `FlexTok` achieves superior reconstruction and generation quality at each token budget, all in a single model.

For comparison, we train a 2D grid-based tokenizer with a flow matching decoder, matching the `FlexTok d18-d28` tokenizer in parameter count, training steps, and source dataset. Unlike `FlexTok d18-d28`, it does not use register tokens, causal masking, or nested dropout. Text-conditioned image generation using an `FlexTok` tokenizer outperforms the 2D grid baseline when generating 2 to

*Table 1.* **System-level comparison on ImageNet-1k class-conditional generation.** `FlexTok` generates variable length token sequences from 1 to 256 tokens long. The `FlexTok` tokenizers are combined with 1.33B parameter AR Transformers for class-conditioned image generation. For each `FlexTok` model we follow the optimal inference parameters and use no classifier-free guidance in the AR model, as described in Appendices F and G. $\dagger$ indicates `FlexTok` results for a sequence of *32 tokens*. The "-re" suffix indicates the use of rejection sampling.

| Tokenizer | # tokens | Codebook size | rFID ↓ | gFID ↓ |
|---|---|---|---|---|
| Taming VQ-GAN-re | 16x16 | 16384 | 4.98 | 5.20 |
| MaskGIT VQ-GAN | 16x16 | 1024 | 2.28 | 6.06 |
| Open MAGVIT-v2 | 16x16 | 262144 | 1.17 | 2.33 |
| LlamaGen | 16x16 | 16384 | 2.19 | 3.06 |
| TiTok-L | 32 | 4096 | 2.21 | 2.77 |
| TiTok-B | 64 | 4096 | 1.70 | 2.48 |
| TiTok-S | 128 | 4096 | 1.71 | 1.97 |
| `FlexTok d12-d12` | 1-256 | 64000 | 4.20$^\dagger$ | 3.83$^\dagger$ |
| `FlexTok d18-d18` | 1-256 | 64000 | 1.61$^\dagger$ | 2.02$^\dagger$ |
| `FlexTok d18-d28` | 1-256 | 64000 | 1.45$^\dagger$ | 1.86$^\dagger$ |

128 tokens (Figure 7). A full 256-token sequence with `FlexTok` yields better text-image alignment (CLIPScore) than the 2D grid tokenizer, despite a slight gFID regression compared to the 2D grid-based model.

## 6. Discussion & Conclusion

In this work, we show the potential of a flexible sequence length tokenizer for image reconstruction and generation. Beyond enabling high-fidelity reconstructions with very few tokens, we demonstrate through training class- and text-conditional AR models that the `FlexTok` token sequences specify a *"visual vocabulary"* that enables generation in a coarse-to-fine ordering.

Our experiments suggest that depending on the complexity of the generation task, a model may be trained to stop the generation early as soon as the condition is fulfilled. While `FlexTok` can semantically compress images into as little as a single token, representing highly dense or structured content like text requires more tokens and training objectives that prioritize semantically meaningful concepts. We believe that these present exciting research directions to speed up and improve autoregressive image generation using adaptive compute budgets.

Looking ahead, we anticipate that `FlexTok`-like tokenizers, which adapt to the intrinsic complexity of the input data, could be applicable to other domains with high redundancy, such as audio and video. Training generative models on representations that can be both very compact and semantic, or very long and detailed, may enable further explorations into long-horizon video generation, understanding, as well as visual reasoning.

## Acknowledgments

We thank Justin Lazarow and Miguel Angel Bautista Martin for their valuable feedback on earlier versions of this manuscript. This work was supported as part of the Swiss AI initiative by a grant from the Swiss National Supercomputing Centre (CSCS) under project ID a08 on Alps.

## Impact Statement

This paper presents work whose goal is to advance the field of Machine Learning. There are many potential societal consequences of our work, none which we feel must be specifically highlighted here.

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

# Appendix

## A. Ablation of `FlexTok` Design Choices

We explore the design space offered by the `FlexTok` framework. The effects of the VAE choice, structure applied to the tokens to induce an ordering, the decoder loss formulation, and the use of inductive bias losses are all investigated in the goal of converging on a high-quality and compact 1D tokenizer that can resample images into variable-length token sequences.

**Default ablation setup.**   Unless otherwise stated:

- All models are trained on images of resolution 256x256.

- We use a 16-channel VAE and produce the VAE latent space by sampling from the Gaussian distributions, rather than just taking the mode.

- We use a FSQ quantization with 6 dimensions, each bucketed into levels [8, 8, 8, 5, 5, 5], for an effective vocabulary size of $64\,000$.

- We use a 2x2 patchification inside the `FlexTok` encoder and decoder that acts on the VAE latent space which itself has a downsample factor of 8. This yields an effective spatial downsample factor of 16 from pixels to patch tokens.

- All `FlexTok` ablation models have encoder and decoder sizes `d12-d12`, and are trained for 50B patch tokens on ImageNet-1k (Russakovsky et al., 2014). We measure one patch token as a 16x16 pixel patch.

- To evaluate the ablation models for class conditional image generation, we train 393M parameter AR Transformers on the resulting token sequences. The AR models are trained for 94B tokens on ImageNet-1k (300 epochs).

**Evaluation Setup**   We evaluate our tokenizers using an array of metrics which probe their reconstruction and generation properties. To evaluate image reconstruction we calculate reconstruction FID (rFID) (Heusel et al., 2017), DreamSim (Fu et al., 2023), and Mean Absolute Error (MAE) on the full validation split of the ImageNet-1k dataset (Russakovsky et al., 2014). During generation, we use a classifier-free guidance scale of 1.5 for the AR model. For evaluation of the class-conditioned image generation results, we follow the common practice of measuring the generation FID (gFID) of 50K generated samples relative to the reference statistics calculated over the entire training split of the ImageNet-1k dataset (Dhariwal & Nichol, 2021).

### A.1. VAE choice ablation

We observe a strong correlation between the VAE reconstruction quality and the number of latent channels (Tab. 2), and find that increasing the number of latent channels above 4 can significantly improve the `FlexTok` reconstruction quality (Fig. 10). This finding is in line with the observations that the VAE latent channel size should be scaled with size of the subsequent latent diffusion model (Esser et al., 2024). To optimize for the largest-sized tokenizer's performance, we select the 16-channel VAE for our final setup.

*Table 2.* **VAE reconstruction performance.** We measure MSE and rFID on the COCO-30k validation set (Lin et al., 2014).

| Model | Arch. | # Latent Channels | MSE ↓ | rFID ↓ |
|---|---|:---:|---|---|
| SDXL VAE | SDXL VAE | 4 | 0.0038 | 1.13 |
| ours | SDXL VAE | 4 | 0.0043 | 1.53 |
| ours | SDXL VAE | 8 | 0.0028 | 0.66 |
| ours | SDXL VAE | 16 | 0.0013 | 0.35 |

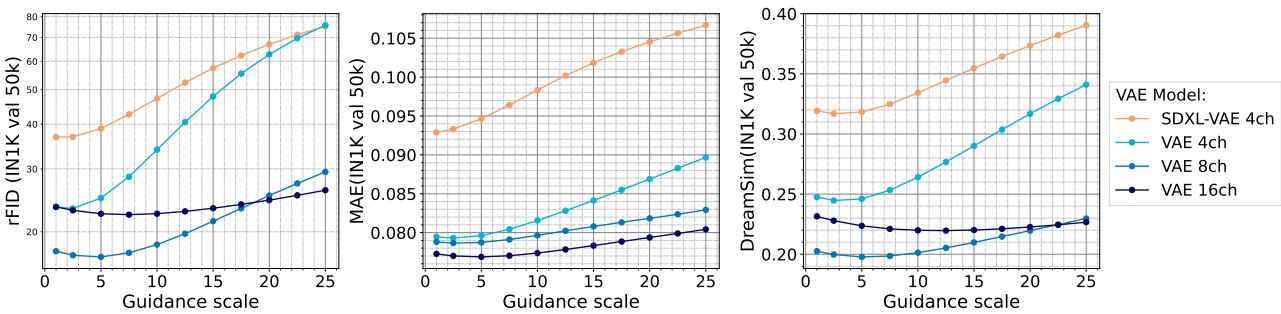

*Figure 10.* **Guidance scale ablation for different VAE choices.** We train `FlexTok d12-d12` models on different VAE choices. We use adaptive projected guidance (Sadat et al., 2024), and evaluate on the ImageNet-1k (Russakovsky et al., 2014) validation set.

### A.2. Resampling strategy ablation

*Table 3.* **Ablation of resampling strategies**. We compare 2D grid tokenization with 1D tokenization, the use of a rectified flow decoder, the training noise schedule, as well as the use of an auxiliary REPA (Yu et al., 2024b) loss. We compare on reconstruction MAE, DreamSim, and rFID, as well as ImageNet-1k class-conditional generation gFID.

| Token Structure | Loss Formulation | Noise | REPA | MAE ↓ | DreamSim ↓ | rFID ↓ | gFID ↓ |
|---|---|---|---|---|---|---|---|
| $16 \times 16$ (2D) | MSE | - | ✗ | 0.059 | 0.288 | 51.27 | 35.93 |
| $16 \times 16$ (2D) | Rectified flow | mode(0.25) | ✗ | 0.083 | 0.272 | 32.85 | 23.29 |
| 256 (1D) | Rectified flow | mode(0.25) | ✗ | 0.078 | 0.220 | 23.28 | 22.16 |
| 256 (1D) | Rectified flow | uniform | ✗ | 0.078 | 0.222 | 23.63 | 21.35 |
| 256 (1D) | Rectified flow | logit-normal | ✗ | 0.082 | 0.208 | 19.03 | 18.69 |
| 256 (1D) | Rectified flow | mode(0.25) | ✓ | **0.075** | **0.128** | **5.98** | **7.40** |

We first compare the resampling strategy (2D grid tokenization versus 1D register tokenization), as well as the use of a rectified flow decoder compared to a simple decoder optimized with MSE. Note that in none of these experiments do we apply any ordering strategies. We reserve those investigations for Appendix A.4.

The results in rows 1 and 2 in Table 3 show that the use of a rectified flow decoder significantly improves both rFID and gFID, while getting worse MAE. When switching from a 2D to a 1D tokenizer (row 2 vs. 3), we see improvements across all metrics, with the largest improvements being on rFID. We also ablate three different choices of noise schedules following Esser et al. (2024), namely a uniform noise schedule (row 4), a logit-normal schedule with location $m = 0$ and scale $s = 1$ (row 5), and a mode sampling schedule with scale $s = 0.25$ (row 3). While the logit-normal schedule shows the strongest performance, we note that we observe inference instabilities due to the earliest and latest timesteps being undertrained. We chose the mode sampling schedule for our models due to the comparatively better reconstruction performance.

### A.3. Use of inductive bias loss.

We observe that rectified flow decoders converge slowly and investigate the use of inductive bias losses (Hu et al., 2023; Yu et al., 2024b) to 1) improve convergence time, and 2) distill the semantic inductive biases of a strong pre-trained vision model into the tokenizer to make the tokens more predictable. Following REPA (Yu et al., 2024b), we train a three-layer MLP to read out the activations of the decoded 2D patches in the first decoder layer, and project them to the DINOv2-L feature dimension. In addition to the rectified flow loss, we add a cosine similarity loss with weight 1.0 between the predicted features and the reference DINOv2-L features.

The last row of Table 3 compared to row 3 shows that the use of REPA significantly improves perceptual reconstruction metrics like DreamSim and rFID, as well as generation performance as measured by gFID. As shown in Figure 12, the use of an inductive bias loss significantly improves convergence time to high quality reconstructions. This is mirrored in our visual comparison in Figure 13, where reconstructions of models with REPA are significantly higher fidelity and more semantic. Interestingly, we find that even though the use of REPA improves the reconstruction and generation metrics significantly, it does not improve the convergence of the reconstruction loss during training of the tokenizer (Figure 11). Given the striking improvements in downstream reconstruction and generation metrics, we use REPA for all experiments from here on.

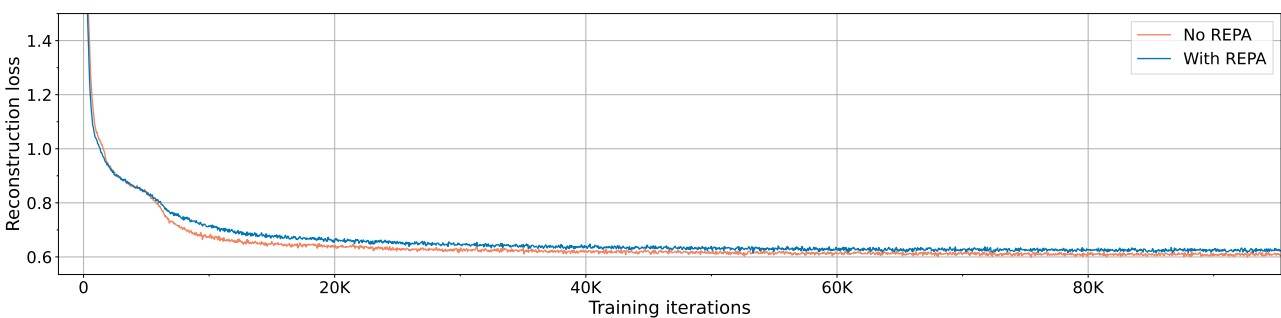

*Figure 11.* **REPA ([Yu et al., 2024b](#)) ablation loss curves.** We train `FlexTok d12-d12` models on ImageNet-1k, with and without REPA. The reconstruction loss shown here does not include the REPA loss contribution.

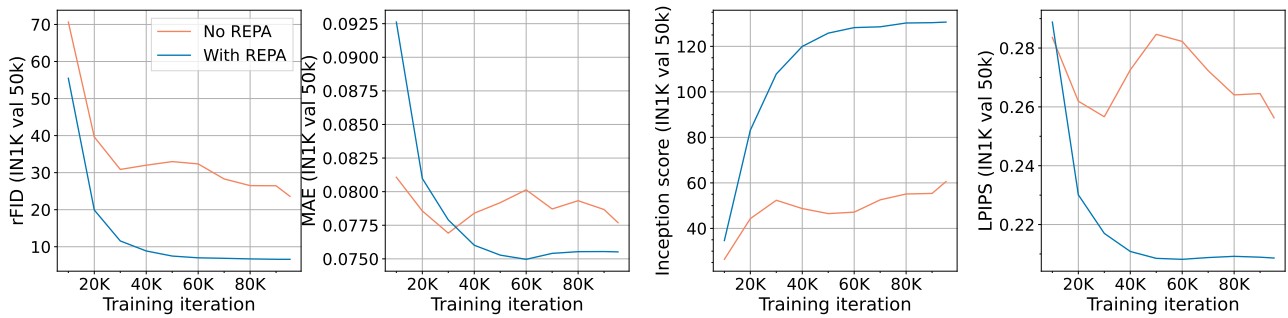

*Figure 12.* **REPA ([Yu et al., 2024b](#)) ablation evaluation curves.** We train `FlexTok d12-d12` models on ImageNet-1k, with and without REPA. The evaluation metrics are measured on the ImageNet-1k ([Russakovsky et al., 2014](#)) validation set.

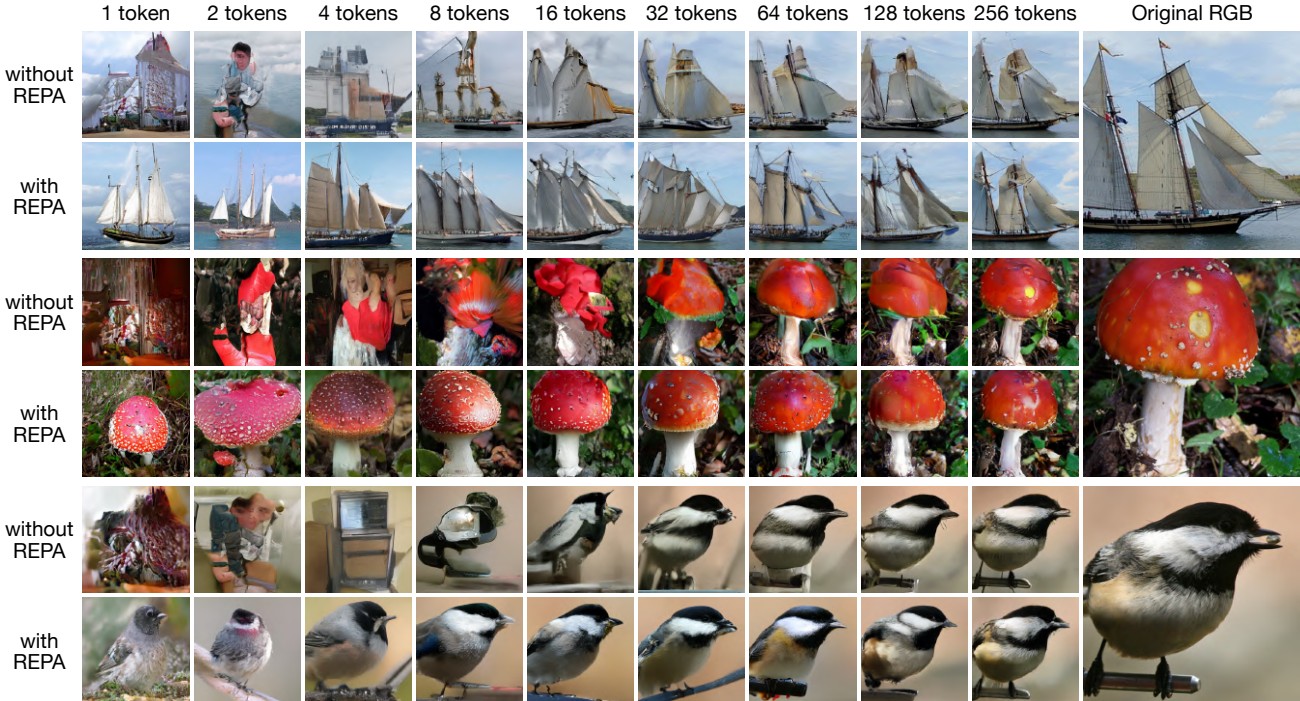

*Figure 13.* **Visual comparison of REPA ([Yu et al., 2024b](#)) ablation models.** We train `FlexTok d12-d12` models on ImageNet-1k, with and without REPA and show ImageNet-1k validation set reconstructions with different number of tokens.

## A.4. Structuring the 1D tokens and inducing an ordering

We ablate various ways to induce an ordering into the register tokens, starting with the 1D rectified flow model trained with REPA, as shown in the last row of Table 3 and first row of Table 4.

*Table 4.* **Comparison of strategies to induce an ordering**. We ablate the use of causal masks on the register tokens, as well as multiple nested dropout variants.

| Number of tokens | MAE ↓ | | | DreamSim ↓ | | | rFID ↓ | | | gFID ↓ | | |
| --- | --- | --- | --- | --- | --- | --- | --- | --- | --- | --- | --- | --- |
| | 4 | 32 | 256 | 4 | 32 | 256 | 4 | 32 | 256 | 4 | 32 | 256 |
| Baseline | ✗ | ✗ | **0.075** | ✗ | ✗ | 0.128 | ✗ | ✗ | 5.98 | ✗ | ✗ | 7.40 |
| Causal register mask | ✗ | ✗ | 0.080 | ✗ | ✗ | **0.118** | ✗ | ✗ | **3.28** | ✗ | ✗ | 5.04 |
| + Uniform nested dropout | 0.268 | **0.148** | 0.080 | 0.522 | **0.275** | 0.132 | 19.97 | **6.48** | 4.01 | 17.99 | **5.26** | **4.83** |
| + "Pow2" dropout | **0.209** | 0.159 | 0.088 | **0.440** | 0.323 | 0.175 | **10.39** | 8.09 | 7.22 | **9.51** | 6.63 | 5.78 |
| + "Unifpow2" dropout | 0.225 | 0.149 | 0.079 | 0.578 | 0.278 | 0.125 | 30.17 | 6.78 | 3.52 | 28.93 | 5.48 | 4.97 |

First, we ablate the use of a causal attention mask (see Section 3.2) applied to the encoder register tokens, see row 2 in Table 4. Compared to unstructured registers, this results in a significant improvement in both rFID and gFID. While the causal mask implicitly induces an ordering over the register tokens, it does not enable use of this model as a flexible-length tokenizer. To that end, we ablate three different nested dropout (Rippel et al., 2014) schedules (see Section 3.2). In "uniform nested dropout" we randomly draw $K_{keep} \in \{1, 2, 3, 4, ..., K\}$. For the "pow2" setting, we uniformly sample from powers of two, i.e. $K_{keep} \in \{1, 2, 4, 8, ..., K\}$. Finally, for "unifpow2", we sample as in "uniform nested dropout" but round to the next-highest power-of-two. Each of these choices puts more or less weight on different subsets of the token sequences. For example, when performing uniform nested dropout each valid number of tokens is only trained on roughly $\frac{1}{256}$ of all samples, while "pow2" and "unifpow2" reduce the number of possible sequence lengths from 256 down to 9. "unifpow2" favors larger token sequences, which is preferable when dealing with images that require more tokens. That said, its rFID and gFID for lower number of tokens is lower due to being relatively undertrained, and "pow2" can present a more balanced approach across the range of sequence lengths (see Figure 14). This improvement of the reconstruction quality for short sequences does come at the cost of a slight regression on the reconstruction and generation performance with the fully 256 token sequences. In Figure 15 we show guidance scale sweeps for all ablation models.

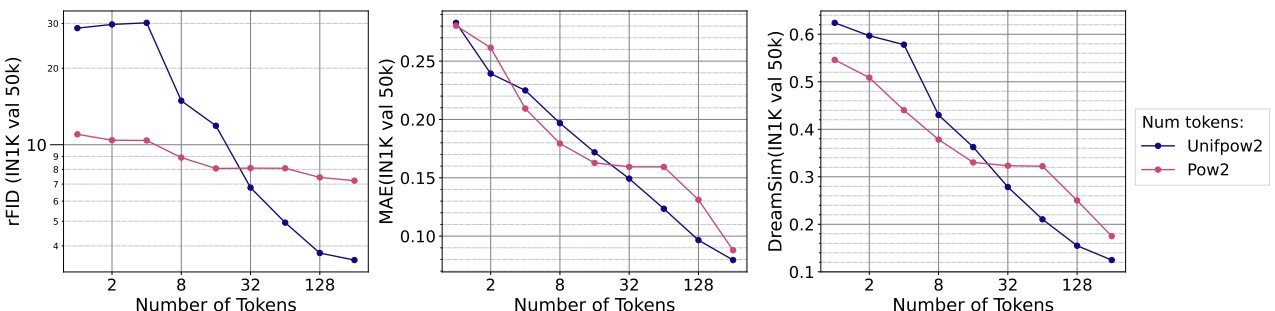

*Figure 14.* **Pow2 vs Unifpow2 nested dropout schedule as a function of the number of tokens.** We train `FlexTok d12-d12` models on ImageNet-1k with different nested dropout schedules and evaluate on the Imagenet-1k validation set. Pow2 presents a more balanced approach across sequence lengths, while Unifpow2 is preferable for higher sequence lengths.

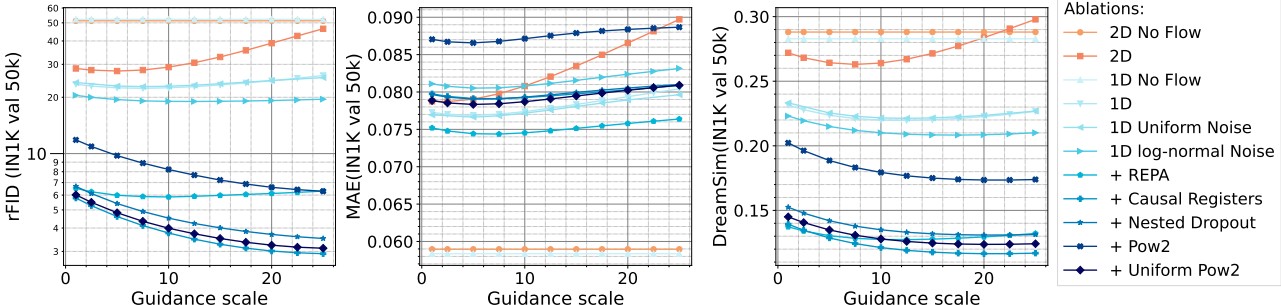

*Figure 15.* **Classifier-free guidance ablations.** For all ablation models, we perform guidance scale sweeps using adaptive projected guidance (Sadat et al., 2024). Evaluation metrics are measured on the ImageNet-1k validation set.

# B. Evaluating the Representation Quality of `FlexTok` Tokenizers

In this subsection, we evaluate the quality of the representations learned by the `FlexTok` tokenizer. Specifically, we perform linear evaluation on the quantized register tokens produced by the `FlexTok` encoder (i.e., the tokens passed as input to the flow model). Importantly, we do not use patch representations, focusing only on the quantized register tokens. For linear evaluation, we train a linear classifier on ImageNet-1k's training set and evaluate it on the test set, keeping the tokenizer frozen throughout. Since a single representation is needed for the linear classifier, we follow the approach of Yu et al. (2024a), where the quantized register tokens are concatenated to form a single feature vector per image. For the rest, we follow the probing setup in Fini et al. (2024). Given the 6-dimensional FSQ latents, this results in feature vectors of size $6 \times$ num_register_tokens.

To tune the linear evaluation recipe, we conduct a hyperparameter search over the following grid: Learning rate $\in [1 \times 10^{-3}, 5 \times 10^{-4}, 2 \times 10^{-4}]$, weight decay $\in [0.1, 0.05, 0.02]$, and minimum crop scale $\in [0.1, 0.2, 0.3, 0.4, 0.5, 0.6, 0.7, 0.8, 0.9]$. Using the optimal hyperparameters found (learning rate $5 \times 10^{-4}$, weight decay $0.05$, crop scale $[0.4, 1.0]$), we perform a sweep over the number of register tokens to collect the results shown in Figure 16. These experiments are conducted for all models (`FlexTok d12-d12`, `FlexTok d18-d18`, `FlexTok d18-d28`) with a batch size of 1024, using 8 H100 or A100 GPUs for each experiment.

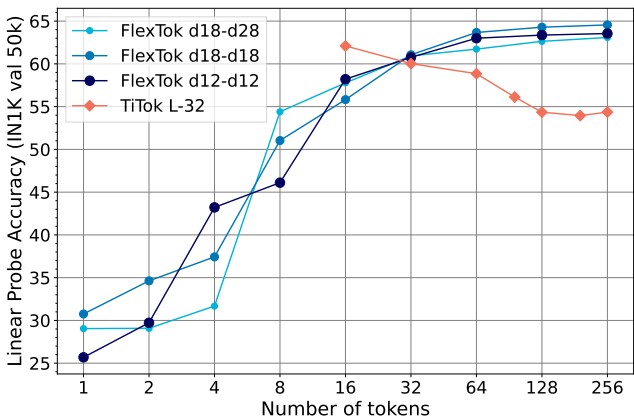

*Figure 16.* **Linear probing experiments.** ImageNet-1k top-1 accuracy on the frozen tokenizer trunk when varying number of tokens.

**Observations.** These experiments are interesting because there is no guarantee that the representations learned by the tokenizer will be linearly separable. However, as our results demonstrate, the representations are indeed linearly separable. The results, presented in Figure 7, reveal several key insights:

- *Improved performance with additional tokens:* Increasing the number of register tokens consistently improves performance. This behavior contrasts with TiTok, where the opposite occurs. The main difference lies in the training approach: TiTok trains a separate model for each configuration (i.e., number of tokens), while we learn a unified model capable of handling all configurations. In our model, when adding more tokens we retain the previous ones unchanged, e.g for $K_{\text{keep}} = 16$ the first 8 tokens will be the same as in the case $K_{\text{keep}} = 8$. Consequently, the linear layer has strictly more information to improve linear separability. In the worst case, the linear layer can simply ignore the additional tokens if they do not contribute useful information. For this reason we expect that performance will monotonically increase with more tokens, which is verified by the experimental evidence. In contrast, TiTok trains a different model for each token configuration, resulting in distinct feature spaces for each model.

- *Superior peak performance:* The best performance achieved by our model significantly surpasses TiTok, with our top-performing configuration reaching 64.6% top-1 accuracy on ImageNet.

- *Trends across encoder and decoder sizes:* A larger encoder improves linear separability when all register tokens are activated; conversely, a larger decoder size leads to a slight degradation in linear separability.

# C. Implementation and Training Details

In this Section, we list implementation and training details for the VAEs in Appendix C.1, the resamplers in Appendix C.2, and the autoregressive models in Appendix C.3.

## C.1. VAE details

We train a series of VAE models following the Stable Diffusion methodology (Rombach et al., 2022). The variational autoencoder architecture is the same as the SDXL VAE (Podell et al., 2023) except we investigate varying the size of the latent dimension from 4 up to 8 and 16 channels. For the discriminator loss we use a 3-layer Patch-GAN discriminator (Isola et al., 2017) that is applied after 5k steps of training and combine it with perceptual (Dosovitskiy & Brox, 2016), KL and reconstruction losses. We train all the VAEs using a learning rate of 1e-3, weight decay of 0.05, model EMA decay of 0.992, spatial down-sampling factor of 8, KL loss weight of 1e-6, discriminator loss weight of 0.5, reconstruction and perceptual loss weights of 1.0, a batch size of 128 and for 305k steps (40B patches) on images from the DFN dataset (Fang et al., 2023). During inference, we produce the VAE latent space by sampling from the Gaussian distribution, rather than just taking the mode.

The trained VAEs are evaluated on the COCO validation split using both mean squared error and reconstruction Fréchet inception distance (rFID) (Heusel et al., 2017) metrics (see Table 2). We find that increasing the latent channel dimension significantly improves the performance of the reconstructions produced by the VAEs. Critical for the upper bound of the performance for the subsequent tokenizer and autoregressive models, the VAE trained with 16 channels achieves a competitive rFID of 0.354. Furthermore, when training `FlexTok d12-d12` tokenizers on these four VAE variants, we show in Figure 10 that choosing Stage 0 VAEs with a low number of channels can have detrimental consequences on the Stage 1 `FlexTok` reconstruction performance. This finding is consistent with Esser et al. (2024), and we choose to train `FlexTok` models using our 16-channel VAE to avoid prematurely putting an upper-bound on its performance.

## C.2. `FlexTok` details

See Table 5 for a detailed breakdown of the resampler tokenizer architecture and training settings. All encoders and decoders are Transformers, whose hidden dimension $w$ is parameterized by the number of layers $d$ using a fixed aspect ratio of 64, i.e. $w = 64 \cdot d$. The number of attention heads is set to $d$. Both the encoder and decoder operate on $2 \times 2$ patches of VAE latents to reduce the sequence lengths they need to process. The registers are a randomly initialized and learnable parameter of shape $K \times d$, concatenated with the VAE patches. We use FlexAttention (PyTorch Team: Horace He, Driss Guessous, Yanbo Liang, Joy Dong, 2024) to create an encoder attention mask in which all patch tokens can attend to each other but not the registers, the registers can attend to all patch tokens, but the $i$-th register token can only attend to the $j$-th register token if $i \geq j$.

We use FSQ as the quantization bottleneck with levels $[8,8,8,5,5,5]$, for an effective codebook size of $64\,000$. When performing nested dropout, we replace the dropped tokens with a learnable mask token.

The `FlexTok` decoders are rectified flow models that are conditioned on the encoder's register tokens by concatenating them with the noised VAE latent patches. Unlike the encoder, the decoder computes full self-attention between the registers and patch tokens, and only the patch tokens are output. The VAE latents are sampled according to a noise schedule that follows Stable Diffusion 3's mode sampling with heavy tails scheme, using scale parameter $s = 0.25$. We use adaLN-Zero (Peebles & Xie, 2023) to condition the registers and image patches *using separate sets of adaLN weights* applied to the same continuous timestep embeddings. As shown in Table 5, the adaLN parameters make up half the total decoder parameters, but we note that their contribution to the overall decoder FLOPS is negligible as they are computed on the time embedding tokens (once for the registers and once for the noised patches). We leave the reduction of this parameter cost to future research. To enable the use of classifier-free guidance (Ho, 2022), we randomly replace the entirety of the encoded registers by a learned null-condition with probability 0.2. The decoder's target is the flow, see Section 3.1.

In addition to the rectified flow loss $\mathcal{L}_{\text{RF}}$, we use REPA (Yu et al., 2024b) to speed up convergence of `FlexTok`. Specifically, we project the first decoder layer activations using a 3-layer MLP with the same dimension as the decoder and ratio 4.0, upsample them to size $37 \times 37$, and compute the cosine similarity with 1024-dimensional DINOv2-L (Oquab et al., 2023) features. The REPA loss $\mathcal{L}_{\text{REPA}}$ is weighted equally to the rectified flow loss, i.e. $\mathcal{L} = \mathcal{L}_{\text{RF}} + 1.0 \cdot \mathcal{L}_{\text{REPA}}$.

The resampler models are either trained on ImageNet-1k (Russakovsky et al., 2014) for downstream use in class-conditional

*Table 5.* **FlexTok training settings.** Model and training configuration for three different model sizes of our resampler tokenizers. See Appendix C.2 for further training details.

| Configuration | FlexTok d12-d12 | FlexTok d18-d18 | FlexTok d18-d28 |
|---|---|---|---|
| Encoder depth $d_{enc}$ | 12 | 18 | 18 |
| Decoder depth $d_{dec}$ | 12 | 18 | 28 |
| Encoder dim. $w_{enc}$ | 768 | 1152 | 1152 |
| Decoder dim. $w_{dec}$ | 768 | 1152 | 1792 |
| Encoder Transformer parameters | 84.9M | 286.7M | 286.7M |
| Decoder Transformer parameters | 84.9M | 286.7M | 1.1B |
| Decoder adaLN (Peebles & Xie, 2023) parameters | 84.9M | 286.7M | 1.1B |
| Max. num. registers $K$ | | 256 | |
| Register attention mask | | Causal (see Section 3.2) | |
| Register nested dropout mode | | Powers of two: *1, 2, 4, 8, 16, 32, 64, 128, 256* (see Section 3.2) | |
| FSQ (Mentzer et al., 2023) levels | | [8,8,8,5,5,5] | |
| VAE channels | | 16 | |
| VAE downsampling factor | | 8 | |
| Patch size | | $2 \times 2$ | |
| Feedforward activation | | SwiGLU (Shazeer, 2020) | |
| Rectified flow decoder | | ✓ | |
| Decoder adaLN-Zero time emb. (Peebles & Xie, 2023) | | ✓ | |
| Noise mode sampling param. $s$ ((Esser et al., 2024), Sec. 3.1) | | 0.25 | |
| Condition dropout prob. | | 0.2 | |
| REPA (Yu et al., 2024b) layer | | 1 | |
| REPA (Yu et al., 2024b) model | | DINOv2-L (Oquab et al., 2023) | |
| REPA (Yu et al., 2024b) projection | | 3-layer MLP with decoder dim. | |
| REPA (Yu et al., 2024b) loss weight | | 1.0 | |
| Training length ($n$ tokens) | | 200B | |
| Warmup length ($n$ tokens) | | 4B | |
| Warmup learning rate | | 1e-6 | |
| Learning rate schedule | | Cosine decay | |
| Model EMA decay | | 0.998 | |
| Optimizer | | AdamW (Loshchilov & Hutter, 2017) | |
| Opt. momentum | | $\beta_1, \beta_2 = 0.9, 0.99$ | |
| Learning rate $\eta$ | | 5.62e-4 | |
| Batch size | | 2048 | |
| $\mu$P (Yang et al., 2022) base dim. | | 256 | |
| Weight decay timescale $\tau_{iter}$ (Wang & Aitchison, 2024) | | $n_{iter} = 381\,470$ | |
| Gradient clipping norm | | 1.0 | |
| Dataset | | ImageNet-1k (Russakovsky et al., 2014) or DFN-2B (Fang et al., 2023) | |
| Image resolution | | $256^2$ | |
| Augmentations | | RandomResizedCrop, RandomHorizontalFlip, Normalize | |
| Data type | | bfloat16 (Burgess et al., 2019) | |

image generation, or on DFN-2B (Fang et al., 2023) for training autoregressive text-to-image models. The image-caption dataset contains a mixture of original and synthetic captions. During training, we randomly select crops of size $256^2$ using random scales in [0.8, 1.0] and aspect ratio in [0.75, 1.3333]. The resulting images are randomly flipped horizontally with a probability of 0.5, and normalized to the range [-1,1].

All models are trained for a total of 200B tokens seen, where one token is counted as a $2 \times 2$ VAE patch, i.e. a $16 \times 16$ grid of pixels. For images of size $256^2$, this amounts to 256 tokens per sample. We linearly warm up the learning rate for 4B

tokens, and decay it using cosine decay. The model is trained using the AdamW (Loshchilov & Hutter, 2017) optimizer, and we swept the learning rate and batch size at a small scale using $\mu$P (Yang et al., 2022), which we transfer directly to all other settings. We note here that we did not sweep these hyperparameters for every resampler setting, but only on the base setting of a rectified flow resampler without REPA, causal register masks, nor nested dropout. We automatically set the weight decay following the interpretation that AdamW with weight decay can be understood as an exponential moving average (EMA) of recent updates (Wang & Aitchison, 2024). Concretely, we compute the weight decay such that it corresponds to averaging over all training iterations, setting $\tau_{iter}$ to the total number of iterations $n_{iter}$. The weight decay $\lambda$ is computed as $\lambda = \frac{1}{n_{iter} \cdot \eta}$, using learning rate $\eta$. For the final model evaluations, we additionally use an EMA of the weights with decay rate 0.998. See the exact settings in Table 5 and the ImageNet-1k training curves in Figure 17.

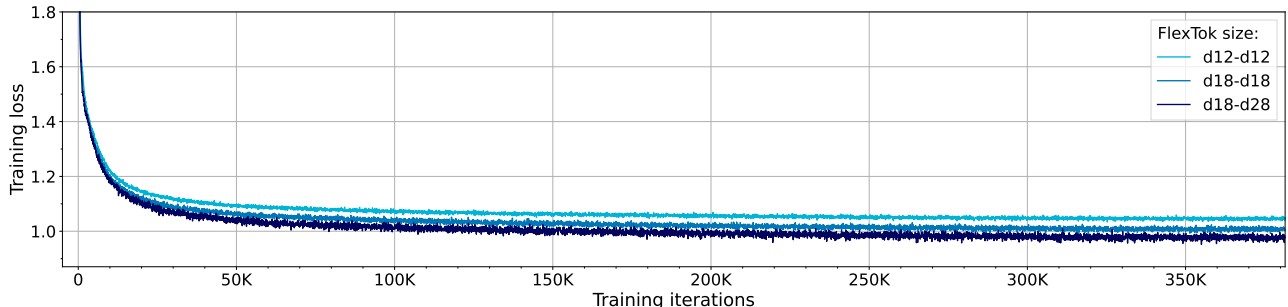

*Figure 17.* `FlexTok` **training loss curves.** The `FlexTok` of different sizes shown here are trained for 200B tokens on ImageNet-1k (Russakovsky et al., 2014). We plot the total loss, i.e. $\mathcal{L}_{\text{RF}} + \mathcal{L}_{\text{REPA}}$.

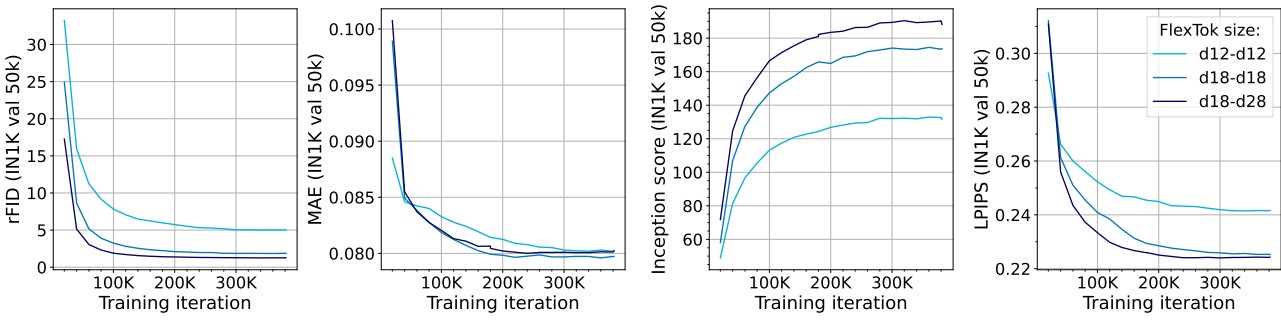

*Figure 18.* `FlexTok` **evaluation curves.** We evaluate the resamplers during training at periodic intervals on the entire ImageNet-1k validation set, showing rFID, MAE, Inception score, and LPIPS.

### C.3. Autoregressive model details

See Table 6 and Table 7 for a detailed breakdown of the class conditional and text-conditional autoregressive transformer architectures and training settings. The AR component of the models are casual decoder Transformers, similar to the `FlexTok` tokenizer modules, the hidden dimension $w$ is parameterized by the number of layers $d$ using a fixed aspect ratio of 64, i.e. $w = 64 \cdot d$. The number of attention heads is set to $d$. We use a MLP ratio of 4 for the FFN hidden dim relative to the attention hidden.

In the class-conditional models we do not apply $\mu$P (Yang et al., 2022) and instead scale the learning rate inversely with the model width. To mitigate overfitting to the ImageNet-1k dataset we apply dropout with 0.1 probability to the FFN, attention, and projection modules in the Transformer decoder blocks and we apply random hornizontal flipping of the images. Additionally we produce 10 random crops per image prior to tokenization(Sun et al., 2024). Whereas, for the text-conditional models we do apply $\mu$P, use a learning rate of 1e-2 for all model sizes, don't use dropout in the Transfomrer decoder blocks, take square center crops and apply no data augmentations to the training images. When scaling up the AR transformer for the text conditioned models we warmup the learning rate for the same number of tokens as the model's parameter count (Porian et al., 2024).

*Table 6.* **Class conditioned AR training settings.** Model and training configuration for different model sizes of our AR transformers.

| Configuration | AR 49M | AR 85M | AR 201M | AR 393M | AR 679M | AR 1.33B |
|---|---|---|---|---|---|---|
| Num. non-embedding Parameters | 49M | 85M | 201M | 393M | 679M | 1.33B |
| Decoder depth $d_{dec}$ | 10 | 12 | 16 | 20 | 24 | 30 |
| Decoder dim. $w_{dec}$ | 640 | 768 | 1024 | 1280 | 1536 | 1920 |
| Cross Attn. dim. | n/a | n/a | n/a | n/a | n/a | n/a |
| MLP Ratio | | | 4 | | | |
| Max. Sequence Length | | | 256 | | | |
| Attention mask | | | Causal | | | |
| Vocab Size | | | 64,000 | | | |
| Feedforward activation | | | SwiGLU (Shazeer, 2020) | | | |
| Positional Encoding | | | Learned Embedding | | | |
| Conditioning dropout prob. | | | 0.1 | | | |
| FFN, Attn. and Projection dropout prob. | | | 0.1 | | | |
| Training length ($n$ tokens) | | | 94B (300 Epochs) | | | |
| Warmup length ($n$ tokens) | | | 9.4B | | | |
| Initial warmup learning rate | | | 1e-6 | | | |
| Learning rate schedule | | | Cosine decay | | | |
| Optimizer | | | AdamW (Loshchilov & Hutter, 2017) | | | |
| Opt. momentum | | | $\beta_1, \beta_2 = 0.9, 0.95$ | | | |
| Learning rate $\eta$ | 1.2E-3 | 1E-3 | 7.5E-4 | 6E-4 | 5E-4 | 4E-4 |
| Final learning rate | | | $\eta \times$1E-2 | | | |
| Batch size | | | 1024 | | | |
| $\mu$P (Yang et al., 2022) base dim. | | | n/a | | | |
| Weight decay | | | 0.05 | | | |
| Weight decay timescale $\tau_{iter}$ (Wang & Aitchison, 2024) | | | n/a | | | |
| Gradient clipping norm | | | 1.0 | | | |
| Dataset | | | ImageNet-1k (Russakovsky et al., 2014) | | | |
| Image resolution | | | $256^2$ | | | |
| Augmentations | | | 10 options RandomResizedCrop, RandomHorizontalFlip, Normalize | | | |
| Data type | | | bfloat16 (Burgess et al., 2019) | | | |

*Table 7.* **Text conditioned AR training settings.** Model and training configuration for different model sizes of our AR transformers.

| Configuration | AR 113M | AR 382M | AR 1.15B | AR 3.06B |
|---|---|---|---|---|
| Num. non-embedding Parameters | 113M | 382M | 1.15B | 3.06B |
| Decoder depth $d_{dec}$ | 12 | 18 | 26 | 36 |
| Decoder dim. $w_{dec}$ | 768 | 1152 | 1664 | 2304 |
| Cross Attn. dim. | 12 | 18 | 26 | 36 |
| MLP Ratio | | 4 | | |
| Max. Sequence Length | | 256 | | |
| Attention mask | | Causal | | |
| Vocab Size | | 64,000 | | |
| Feedforward activation | | SwiGLU (Shazeer, 2020) | | |
| Positional Encoding | | Learned Embedding | | |
| Conditioning dropout prob. | | 0.1 | | |
| FFN, Attn. and Projection dropout prob. | | 0.0 | | |
| Training length ($n$ tokens) | | 284B | | |
| Training FLOPs | 1.93E+20 | 6.39E+20 | 1.90E+21 | 5.00E+21 |
| Warmup length ($n$ tokens) | | | | |
| Initial warmup learning rate | | 1e-6 | | |
| Learning rate schedule | | Cosine decay | | |
| Optimizer | | AdamW (Loshchilov & Hutter, 2017) | | |
| Opt. momentum | | $\beta_1, \beta_2 = 0.9, 0.95$ | | |
| Learning rate $\eta$ | | 1e-2 | | |
| Final learning rate | | $\eta \times$1E-2 | | |
| Batch size | | 8192 | | |
| $\mu$P (Yang et al., 2022) base dim. | | 256 | | |
| Weight decay | | 0.05 | | |
| Weight decay timescale $\tau_{iter}$ (Wang & Aitchison, 2024) | | $n_{iter} = 135\,594$ | | |
| Gradient clipping norm | | 1.0 | | |
| Dataset | | DFN-2B (Fang et al., 2023) | | |
| Image resolution | | $256^2$ | | |
| Augmentations | | n/a | | |
| Data type | | `bfloat16` (Burgess et al., 2019) | | |

# D. Related Work

**Image tokenization.**    The goal of tokenization is to convert high-dimensional images into a more compact sequence of token representations, making diffusion and flow (Rombach et al., 2022; Podell et al., 2023; Ma et al., 2024; Esser et al., 2024; Yu et al., 2024b), masked (Chang et al., 2022; 2023; Li et al., 2022; Lu et al., 2022; 2023; Mizrahi et al., 2023; Bachmann et al., 2024), or autoregressive (Chen et al., 2020; Ramesh et al., 2021; Yu et al., 2022; Sun et al., 2024) image modeling more tractable. Usually, these tokens are learned using an autoencoder objective with a discrete (Van Den Oord et al., 2017; Razavi et al., 2019) or continuously regularized (Rombach et al., 2022) bottleneck. In addition, auxilliary perceptual and discriminator losses (Esser et al., 2021; Yu et al., 2021) are commonly used to abstract away imperceptible details, while latent bias losses (Hu et al., 2023) have been shown to facilitate downstream token prediction. In the discrete case, vector quantization (VQ) (Van Den Oord et al., 2017) has been the de facto standard, but more recent techniques such as LFQ (Yu et al., 2023) and FSQ (Mentzer et al., 2023) show promising scaling trends.

**Tokenizers with rectified flow decoding.**    A core property of image tokenizers is the ability to reconstruct perceptually plausible images from the heavily compressed tokens. While discriminator losses (Esser et al., 2021) are commonly used for this, their training can be notoriously difficult (Yu et al., 2024a). Instead, we opt to train the decoder as a rectified flow (Liu et al., 2022) model, similar to other works (Shi et al., 2022; Mizrahi et al., 2023; Bachmann et al., 2024; Xu et al., 2024; Zhao et al., 2024), as this approach has proven successful in scalable image generation with both coarse and fine conditioning – a property that is particularly useful for our model, `FlexTok`, which can handle a wide range of conditioning specificity.

**Structured tokenization.**    So far, most of the aforementioned image tokenization methods project images into a fixed-size 2D grid. Methods like VAR (Tian et al., 2024) and Infinity (Han et al., 2024) showed impressive scaling trends when projecting images into a structured multi-scale latent representation, while TiTok (Yu et al., 2024a), TexTok (Zha et al., 2024), and DisCo-Diff (Xu et al., 2024) explored doing away with the 2D grid entirely and projecting images into an unstructured but highly compact 1D sequence. Still, the number of tokens an image is represented with is either fixed in the case of Titok, or depends on the image resolution in the case of Infinity.

**Structured and adaptive tokenization.**    To address the issue of the token sequence length depending entirely on the height and width instead of the image complexity, a range of concurrent works have proposed adaptive tokenization methods:

- ElasticTok (Yan et al., 2024) is a joint image and video tokenizer that performs nested dropout during training between a pre-specified minimum and maximum number of tokens. The authors note that the minimum bound is set to 128 or 256 due to instabilities at lower values. Compared to ElasticTok, we observe stable training of `FlexTok` down to a single token. In addition, by training the decoder with a rectified flow objective, `FlexTok` is able to reconstruct high-fidelity images at any number of tokens. Comparatively, ElasticTok requires much higher number of tokens to reconstruct images with high perceptual quality due to being trained only with an MSE and LPIPS loss.

- ALIT (Duggal et al., 2024) presents an adaptive length tokenizer that recurrently encodes images into a growing sequence of tokens. This process can be dynamically halted, leading to adaptively sized image representations. Compared to ALIT, `FlexTok` encodes images in a single forward pass, and we efficiently enforce an ordering through causal masks and nested dropout. Since the tokenizers are trained on a 100-class subset of ImageNet-1k and not evaluated on generative tasks, its scaling and generation properties are not entirely experimentally demonstrated. In comparison, we scale both the `FlexTok` and downstream AR models, and demonstrate strong generative modeling capabilities.

- One-D-Piece (Miwa et al., 2025) is architecturally similar to TiTok (Yu et al., 2024a) and `FlexTok`, in that it uses a register encoder and decoder and is trained using nested dropout on token sequences between 1 and 256. While the One-D-Piece models achieve good reconstruction metrics at high number of tokens, the model fails to produce plausible images at lower number of tokens, unlike `FlexTok`. In addition, it requires a two-stage training approach like TiTok, while `FlexTok`'s Stage 1 is trained end-to-end. Similar to ElasticTok and ALIT, the authors do not evaluate the use of One-D-Piece tokens to train generative models.

- ViLex (Wang et al., 2024b) uses a pre-trained text encoder and diffusion model to learn variable-sized sets of "text tokens" that encode a given image. These soft tokens can be combined with image prompts to generate novel images, similar to textual inversion (Gal et al., 2022). The goal of `FlexTok` lies more in learning a generic set of coarse-to-fine-grained token sequences, which can both be highly semantic for short sequences and highly detailed for longer

sequences. We believe that the use of pre-trained diffusion models as the decoder presents exciting future research directions.

- CAT (Shen et al., 2025) presents a nested VAE architecture that can adaptively compress images into 8x, 16x, and 32x spatially down-sampled representations. Compared to `FlexTok`'s discrete 1D representations, these token grids are 2D and continuous.

**Ordered representation learning.**    Besides image generation, learning ordered representations has been a long-studied topic. Nested dropout (Rippel et al., 2014) learns a variably sized bottleneck representation through uniformly dropping latents from one side, while Matryoshka Represenation Learning (Kusupati et al., 2022) proposes to sample the representation dimensions from powers of two, and decodes each with a weight shared decoder. Matroshka Multimodal Models (Cai et al., 2024) proposes instead to adaptively pool 2D vision encoder representations into smaller 2D grid sizes. UViM (Kolesnikov et al., 2022) proposes to use nested dropout to learn codes that are more robust and easier to model with a downstream language model.

**Variable-rate compression.**    In lossy image compression, how to trade compression strength (rate) for reconstruction performance (distortion) has been a long-studied topic. Classical lossy image compression codecs (Wallace, 1992; Zern et al., 2024; Taubman & Marcellin, 2001) and more recent neural compression algorithms (Mentzer et al., 2020; El-Nouby et al., 2022; Hoogeboom et al., 2023) effectively enable users to choose this trade-off and are able to compress simple images to smaller file sizes compared to more complex ones. While classical JPEG images are transmitted and decoded in a raster-scan order, progressive schemes like progressive JPEG (Wallace, 1992) structure images in a coarse-to-fine manner, allowing users to very quickly reconstruct a low-quality version of an image.

## E. Additional ImageNet Reconstruction and Generation Metrics

To complement the ImageNet-1k reconstruction metrics (rFID, MAE, and DreamSim) shown in Figure 5, we provide additional image-wise reconstruction metrics in Table 8, including PSNR, SSIM, and LPIPS, measured on the ImageNet-1k validation set. We also supplement the class-conditional generation results from Figure 7 (right) and Figure 9 with additional metrics in Table 9, which includes CLIP score, gFID, sFID, Inception score, precision, and recall measured against the complete ImageNet-1k training set.

*Table 8.* `FlexTok d18-d28` **ImageNet-1k reconstruction metrics**. For the largest `FlexTok d18-d28` model trained on ImageNet-1k, we show reconstruction metrics on the full validation set that measure distribution-level differences (reconstruction FID), as well as image-wise distortions in pixel-space (MAE, PSNR, and SSIM) and feature-space (LPIPS and DreamSim). Given a vocabulary size of 64 000, each token takes up 2 bytes of storage.

| # tokens | # bytes | rFID ↓ | MAE ↓ | PSNR ↑ | SSIM ↑ | LPIPS ↓ | DreamSim ↓ |
|---|---|---|---|---|---|---|---|
| 1 | 2 | 4.01 | 0.273 | 9.35 | 0.187 | 0.701 | 0.546 |
| 2 | 4 | 3.09 | 0.237 | 10.25 | 0.222 | 0.651 | 0.494 |
| 4 | 8 | 2.43 | 0.197 | 11.51 | 0.254 | 0.582 | 0.417 |
| 8 | 16 | 1.90 | 0.185 | 11.90 | 0.269 | 0.532 | 0.321 |
| 16 | 32 | 1.61 | 0.154 | 13.05 | 0.304 | 0.462 | 0.271 |
| 32 | 64 | 1.45 | 0.134 | 13.96 | 0.330 | 0.407 | 0.227 |
| 64 | 128 | 1.37 | 0.126 | 14.34 | 0.343 | 0.380 | 0.207 |
| 128 | 256 | 1.20 | 0.102 | 15.90 | 0.407 | 0.293 | 0.158 |
| 256 | 512 | 1.08 | 0.081 | 17.70 | 0.489 | 0.219 | 0.114 |

*Table 9.* **Class-conditional 1.33B AR model with** `FlexTok d18-d28` **generation metrics on ImageNet-1k**. We show generation metrics for a 1.33B class-conditional AR model trained on tokens from `FlexTok d18-d28`. The generation FID, sFID, Inception score, precision, and recall are measured using 50k generated samples by comparing against the full ImageNet-1k train set statistics, while the CLIP score is computed on the full validation set instead.

| # tokens | CLIP score ↑ | gFID ↓ | sFID ↓ | IS ↑ | Precision ↑ | Recall ↑ |
|---|---|---|---|---|---|---|
| 1 | 26.84 | 3.14 | 6.54 | 236.47 | 0.83 | 0.53 |
| 2 | 27.10 | 2.51 | 5.33 | 238.07 | 0.82 | 0.57 |
| 4 | 27.51 | 2.00 | 4.86 | 226.77 | 0.80 | 0.60 |
| 8 | 28.84 | 1.82 | 4.53 | 266.48 | 0.82 | 0.61 |
| 16 | 29.12 | 1.75 | 4.32 | 277.45 | 0.82 | 0.61 |
| 32 | 29.10 | 1.71 | 4.46 | 284.99 | 0.82 | 0.61 |
| 64 | 29.14 | 1.76 | 4.53 | 286.40 | 0.82 | 0.61 |
| 128 | 29.03 | 1.89 | 5.24 | 275.63 | 0.82 | 0.61 |
| 256 | 28.94 | 2.45 | 6.76 | 258.33 | 0.80 | 0.61 |

# F. `FlexTok` Inference Hyperparameter Sweeps

For trained `FlexTok` models, we sweep different inference-time hyperparameters relating to the rectified flow decoder, such as the number of denoising steps in Figure 19 and the classifier-free guidance scale when using adaptive projected guidance (APG) (Sadat et al., 2024) in Figure 21. We further show guidance sweeps when using a vanilla guidance formulation in Figure 20, and compare guidance scales for our largest 1D and 2D tokenizers in Figure 22.

## F.1. `FlexTok` flow decoder timestep sweeps

We sweep the number of denoising steps between 1 and 100 on `FlexTok d18-d28` (trained on ImageNet-1k) for different number of tokens. By default, we set the guidance scale to 7.5 and use APG (Sadat et al., 2024). We find that denoising for more than 25 steps leads to significant diminishing returns in terms of the reconstruction quality metrics, as shown in Figure 19. rFID and DreamSim metrics plateau, but the mean absolute error to the original images continues to increase, especially for low number of tokens. We choose 25 denoising steps for all subsequent ablations and use it as our default value for all model sizes, as it provides a good balance between inference speed and quality.

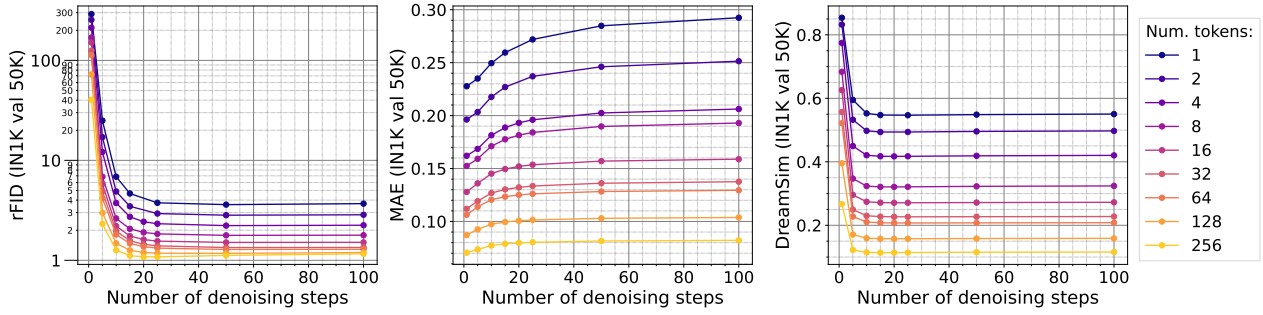

*Figure 19.* **Number of inference denoising steps ablation.** We sweep number of inference steps on `FlexTok d18-d28` (trained on ImageNet-1k) and find that 25 denoising step provides a good balance between inference speed and quality.

## F.2. Ablating standard classifier-free guidance

We find that using standard classifier-free guidance (Ho, 2022) in the flow matching decoder results in narrow basins of optimal reconstruction performance that are highly dependent on the number of tokens provided to the decoder (see Figure 20). We instead opt to use APG (Sadat et al., 2024), as discussed in Appendix F.3.

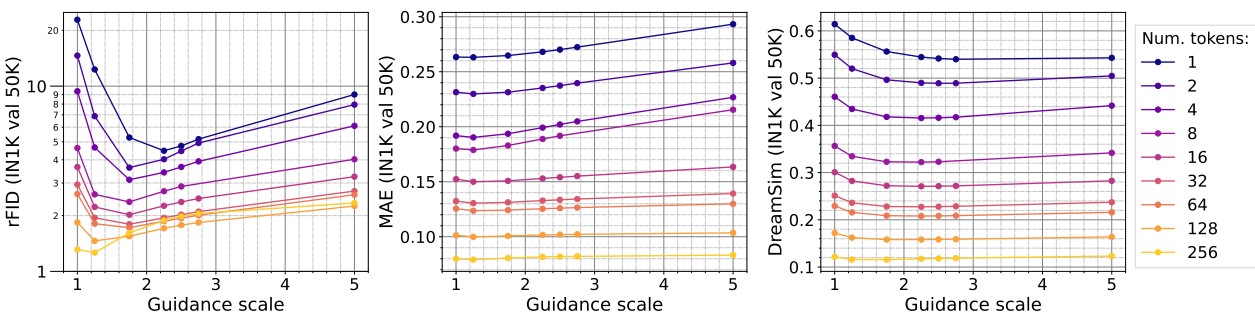

*Figure 20.* **Standard classifier-free guidance ablation.** We sweep guidance scales for `FlexTok d18-d28` (trained on ImageNet-1k) and find optimal guidance scale basins to be narrow and vary strongly across different number of tokens used.

## F.3. `FlexTok` normalized guidance sweeps

We would like to avoid cases where the optimal guidance scales varies greatly across different number of `FlexTok` tokens. As observed in Appendix F.2, standard classifier-free guidance can exhibit narrow optimality basins. For that reason, we explore the use of normalized guidance schemes like adaptive projected guidance (APG) (Sadat et al., 2024). We choose

hyperparameters for rescaling threshold as $r = 2.5$, parallel component as $\eta = 0$, and momentum as $\beta = -0.5$, and sweep the guidance scale. Compared to standard guidance, using APG (Sadat et al., 2024) yields optimal performance basins which are shallower, more aligned across differing numbers of tokens, and have overall improved rFID values. The basins of smaller `FlexTok` models are less aligned than the ones of larger models, and their optimal guidance values are higher. No matter the model size, the DreamSim metric improves and plateaus with higher guidance scales, while the MAE degrades slightly.

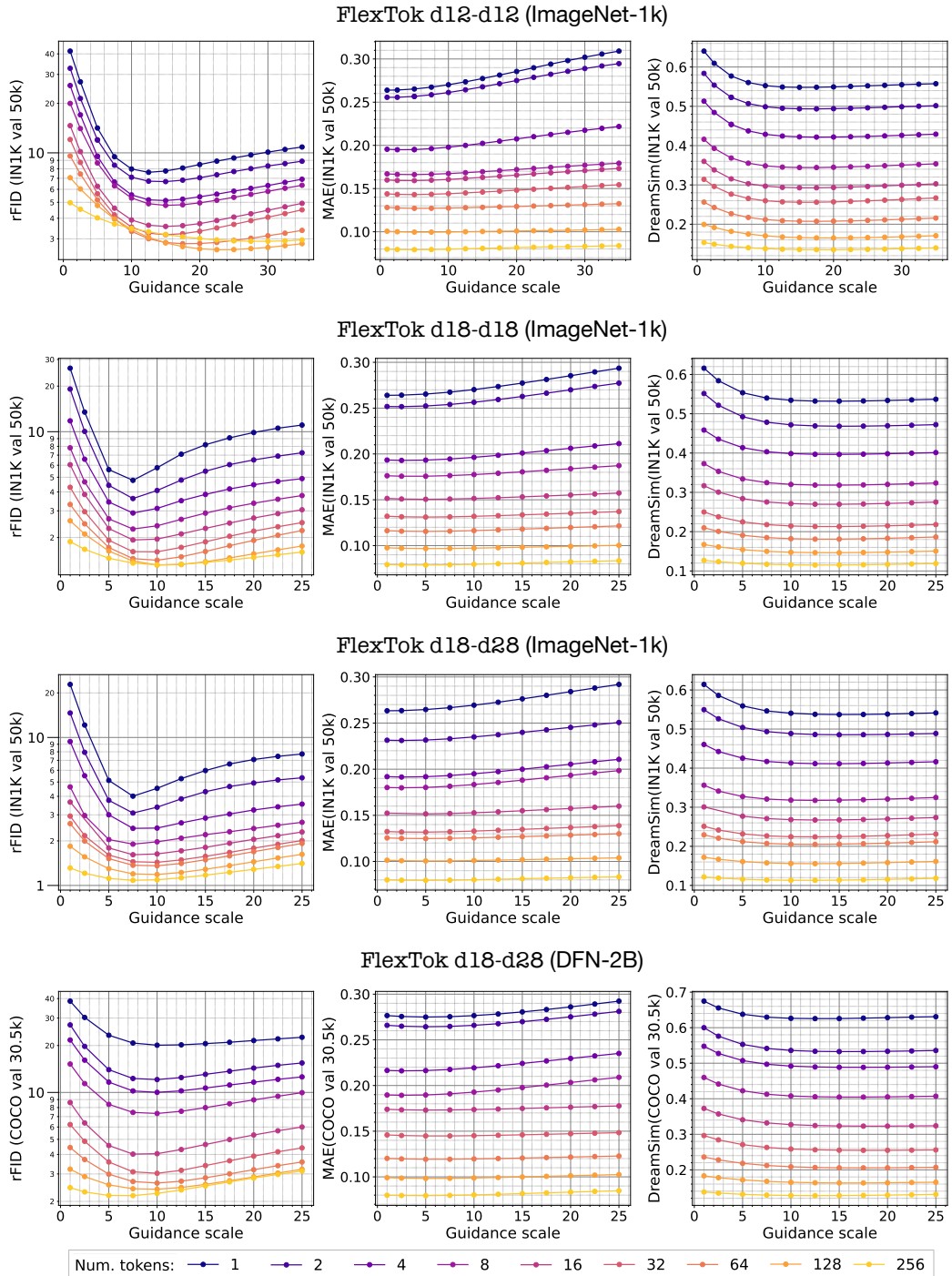

*Figure 21.* **Adaptive projected guidance (Sadat et al., 2024) ablation.** We sweep guidance scales for different `FlexTok` model sizes and number of tokens and find that APG results in smoother guidance scale basins than standard guidance.

### F.4. 1D vs 2D tokenizer classifier-free guidance ablation

For our largest tokenizer size `FlexTok d18-d28` (trained on DFN-2B), we compare classifier-free guidance basins with our controlled 2D grid tokenizer baseline on the COCO 30.5k validation set. Both models use APG (Sadat et al., 2024). As shown in Figure 22, we find that in terms of rFID and DreamSim score, the 1D tokenizer outperforms the 2D baseline by a large margin across guidance scale values, but observe that the 2D grid tokenizer performs better in terms of MAE.

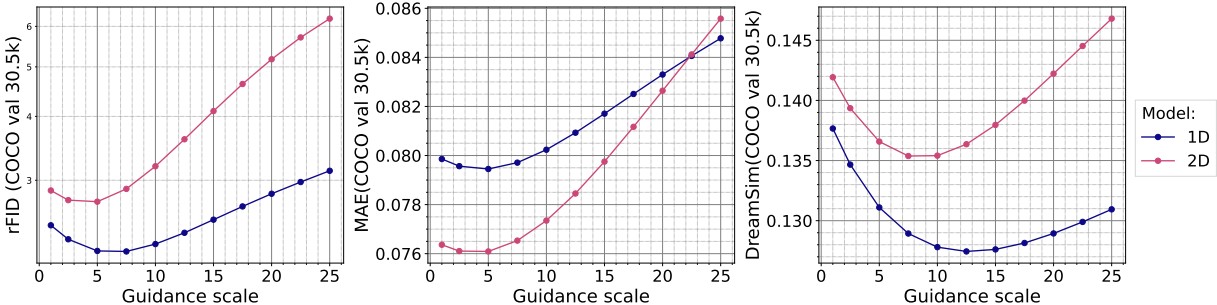

*Figure 22.* **1D vs 2D `FlexTok d18-d28` guidance scale ablation.** We sweep the APG guidance scale for `FlexTok d18-d28` (trained on DFN-2B) and the controlled 2D tokenizer baseline. The 1D `FlexTok d18-d28` model is evaluated using the full 256 tokens.

### F.5. Final `FlexTok` inference hyperparameters

From these observations we make the final choices of the inference hyperparameters for each of the model sizes and training datasets and use these for all subsequent image generation experiments. We detail them in Table 10 for ImageNet-1k tokenizers and 11 for DFN-2B tokenizers. We note that common timestep and guidance-scale distillation techniques for diffusion and flow models are applicable to `FlexTok` decoders, and we expect that such steps could reduce the computational requirements of running the decoder significantly.

*Table 10.* **ImageNet-1k `FlexTok` inference settings.** Chosen tokenizer inference configurations used for the three different model sizes trained on ImageNet-1k.

| Configuration | FlexTok d12-d12 | FlexTok d18-d18 | FlexTok d18-d28 |
|---|---|---|---|
| # denoising steps | 25 | 25 | 25 |
| Adaptive Projected Guidance (APG) (Sadat et al., 2024) | True | True | True |
| Decoder guidance scale | 15 | 7.5 | 7.5 |

*Table 11.* **DFN `FlexTok` inference settings.** Chosen tokenizer inference configurations used for the three different model sizes trained on DFN-2B.

| Configuration | FlexTok d18-d28 | 2D Grid d18-d28 |
|---|---|---|
| # denoising steps | 25 | 25 |
| Adaptive Projected Guidance (APG) (Sadat et al., 2024) | True | True |
| Decoder guidance scale | 7.5 | 5.0 |

# G. Autoregressive Class-Conditional Image Generation Hyperparameter Sweeps

For class-conditional AR Transformers trained on top of `FlexTok` tokenizers we sweep a variety of inference-time hyperparameters. We implement classifier-free guidance in the AR model as a update of the prediction logits $\hat{x}$ given by $\hat{x} = x_{\text{uncond}} + s \cdot (x_{\text{cond}} - x_{\text{uncond}})$. Here $x_{\text{cond}}$ and $x_{\text{uncond}}$ are the logits with and without the conditioning supplied to the model. Taking a fully trained AR 1.33B and `FlexTok d18-d28` we find that using no guidance, i.e. CFG scale $s = 1.0$, results in the best gFID values irrelevant of the top-k sampling or number of tokens generated before decoding to image space (Figures 23 and 24). A optimal gFID value with no guidance sets AR models with `FlexTok` tokenizers apart from previous examples in the literature. For example the optimal CFG value for the LlamaGen models was 2.0 for their class-conditional models (Sun et al., 2024). The ordering introduced by the causal mask and nested-dropout in the tokenizer could be the source of the property in the subsequent AR model. There are significant inference compute advantages if CFG is not applied, in particular the batch doesn't need to be increased by a factor of 2 to accommodate the unconditional samples. Besides the interesting dependence on the classifier-free guidance scale we find that increasing the top-k sampling threshold improves the gFID values. For optimal generation FID we find that no top-k sampling should be used.

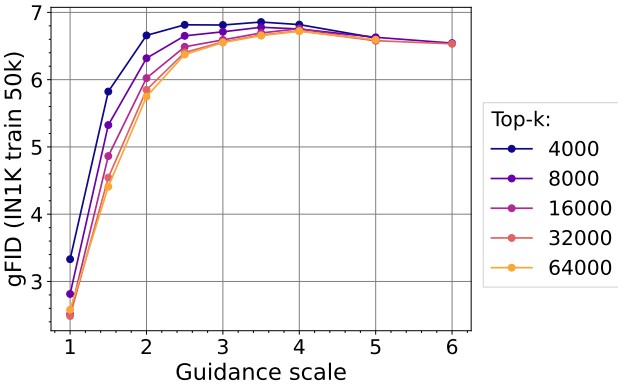

*Figure 23.* **Class-conditional 1.33B AR model with `FlexTok d18-d28` guidance and top-K ablation.** gFID measured with respect to the full ImageNet-1k train set statistics.

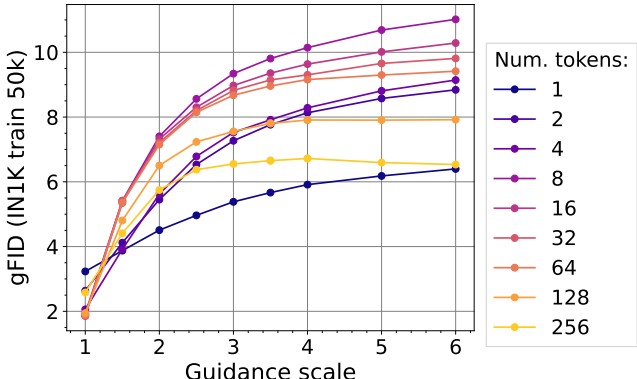

*Figure 24.* **Class-conditional 1.33B AR model with `FlexTok d18-d28` guidance and number of tokens ablation.** gFID measured with respect to the full ImageNet-1k train set statistics.

## H. Autoregressive Class-Conditional Image Generation Model Size

By scaling up the AR model size in the class conditioned models we observe consistent improvements in the training loss (Figure 25). Following the optimal inference parameters observed in Appendix G, no CFG and no top-k sampling, we evaluate the effect of scaling the AR model size on top of the `FlexTok d18-d28` tokenizer. Figure 26 shows that generation performance when producing only a few tokens is independant of model size. However the longer the sequence the more the larger AR model size limits token decoding errors. When generating the full 256 tokens scaling up the AR model size significantly improves the gFID values. We note that even with the largest AR model investigated here, with 1.33B parameters, we still observe a slight regression in the gFID values as the number of decoded tokens increases about 128.

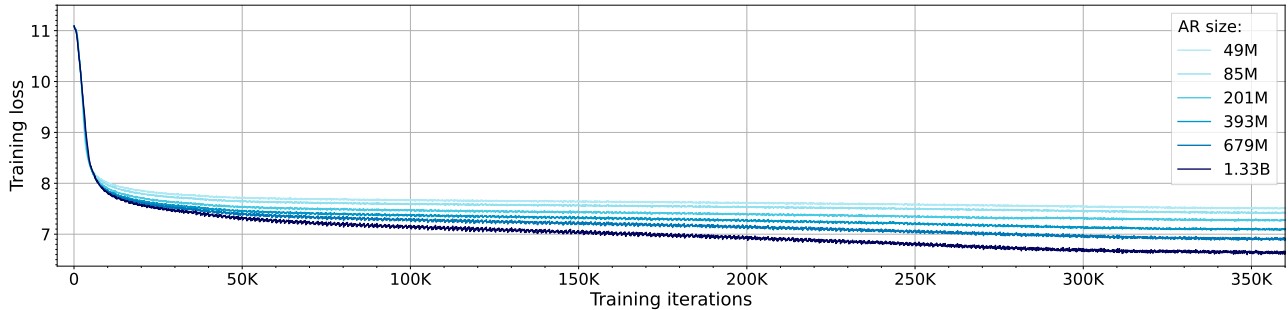

*Figure 25.* **Autoregressive class-conditional image generation loss curves.** The AR models of different sizes shown here are trained for 94B tokens on ImageNet-1k using the `FlexTok d18-d28` tokenizer trained on ImageNet-1k.

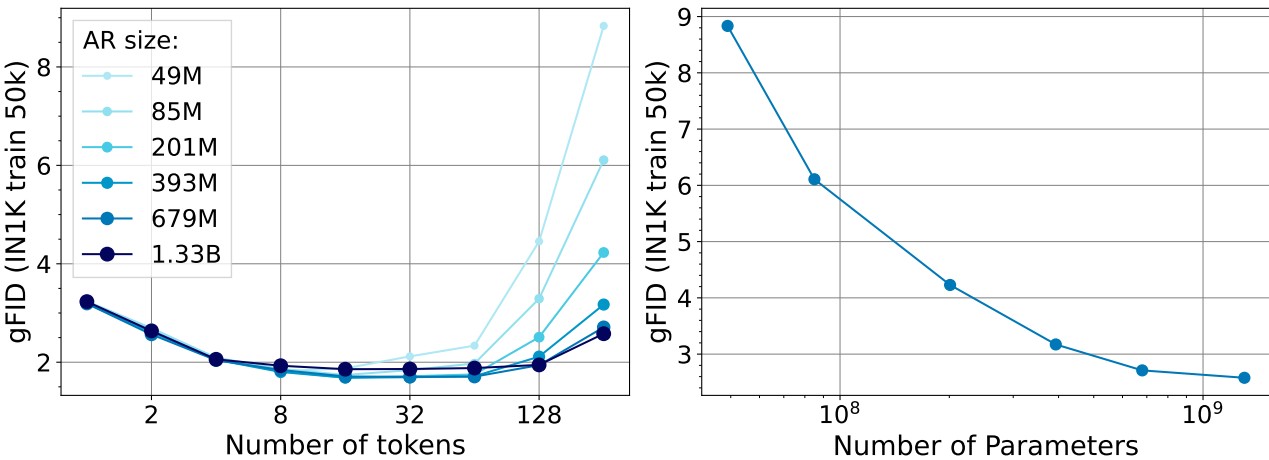

*Figure 26.* **Autoregressive class-conditional image generation model scaling.** Left shows the gFID values for each model size at varying numbers of generated tokens. The right figure shows the gFID with 256 tokens against the parameter count of the AR model. The gFID values are measured with respect to the full ImageNet-1k train set statistics. All AR models are trained using the `FlexTok d18-d28` tokenizer trained on ImageNet-1k. During generation we use the optimal inference parameters detailed in Appendix G.

Scaling up the tokenizer size significantly improves the generation quality (Figure 27). With a fixed AR model size of 1.33B we scale up the tokenizers size from `FlexTok d12-d12` to `FlexTok d18-d28` and find improvements in the measured gFID values. For some tokenizer sizes the reconstruction quality of the tokenizer is upper bounding the performance of the generative model.

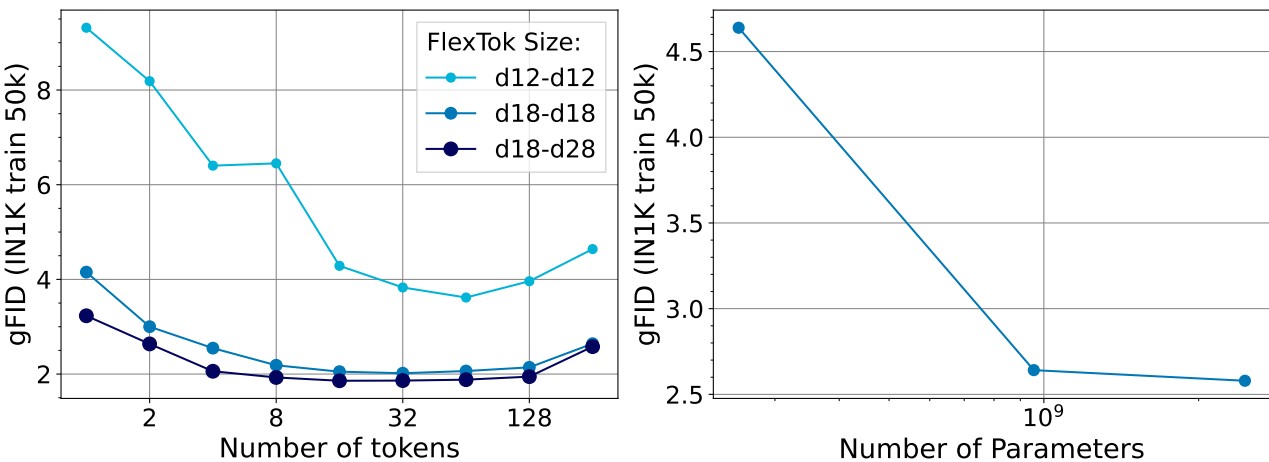

*Figure 27.* **Autoregressive class-conditional image generation `FlexTok` tokenizer scaling.** Left shows the gFID values for 1.33B AR models trained with each of the different sized `FlexTok` tokeninizers at varying numbers of generated tokens. The right figure shows the gFID with 256 tokens against the parameter in the `FlexTok` tokenizer. The gFID values are measured with respect to the full ImageNet-1k train set statistics. During generation we use the optimal inference parameters detailed in Appendix G.

# I. Autoregressive Text-Conditional Image Generation Inference Hyperparameter Sweeps

For text conditional AR Transformers trained with `FlexTok` tokenizers we sweep the hyperparameter classifier-free guidance scale and number of decoded tokens. Taking a fully trained AR 3.06B and `FlexTok d18-d28` we find that using classifier-free guidance improves the gFID for all lengths of generated token sequences (Figure 28). This observation in contrast to the class conditioned `FlexTok` based models where using CFG hurt gFID performance. The difference in the complexity of the conditioning and any distribution shift between the DFN training and COCO validation datasets could be a contributing factor here. In addition to the CFG improving gFID values we also find that the CLIPScore increases with increasing guidance.

We find that a CFG scale of 2.5 optimizes the gFID values across the variety of number of tokens generated. Using this CFG scale we see gFID values which sharply drops and then gradually increase as the number of tokens generated increases (Figure 29). In comparison we find that the text-image alignment measured by the CLIPScore only improves as the number of tokens is increased.

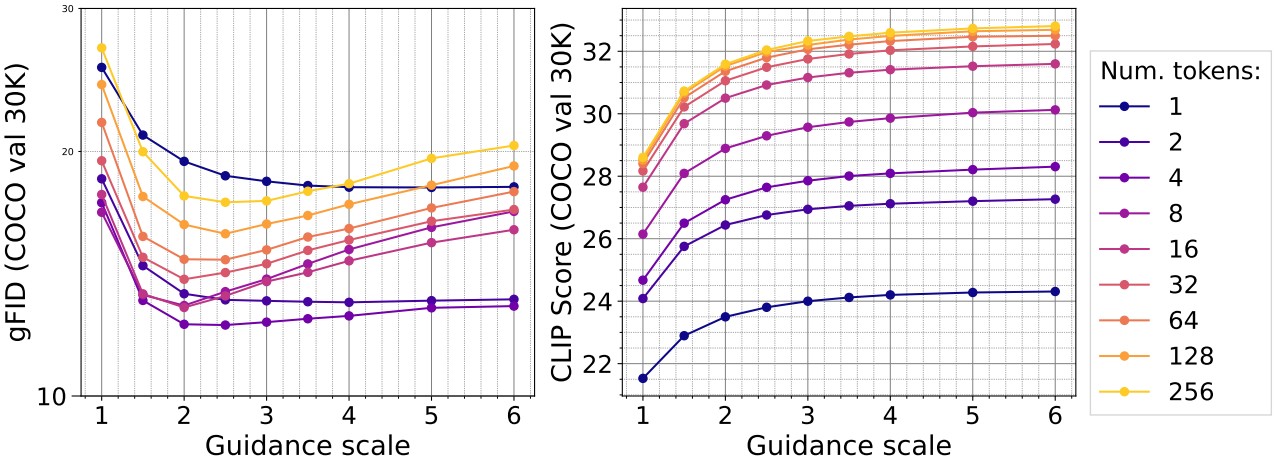

*Figure 28.* **Text-conditional 3.06B AR with `FlexTok d18-d28` guidance and number of tokens ablation.** gFID (left) and CLIPScore (right) metrics measured with respect to the COCO validation set at varying numbers of tokens.

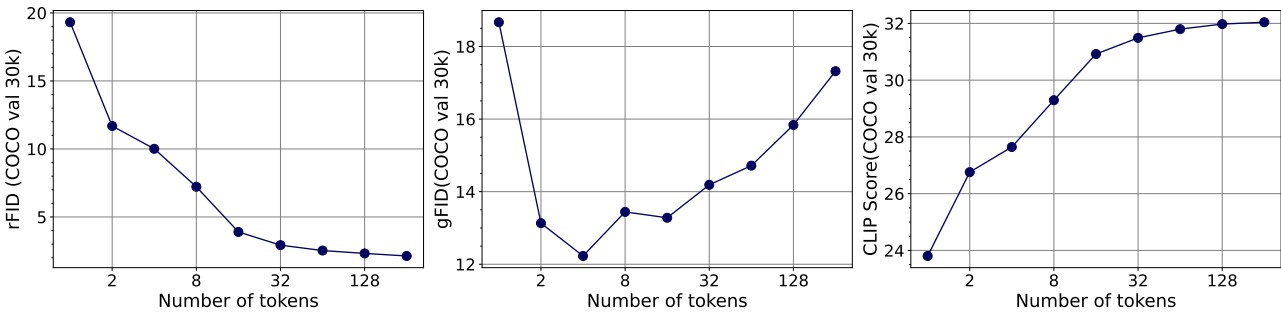

*Figure 29.* **Text-conditional 3.06B AR with `FlexTok d18-d28` image generation vs number of tokens.** All metrics are computed using a CFG scale of 2.5 during generation. Left shows the rFID of the DFN trained `FlexTok d18-d28` on the COCO validation set. The AR image generation gFID (middle) and CLIPScore (right) on the COCO validation set as functions of the number of generated tokens.

Figure 30 shows the 2D grid based tokenizer has a similar dependence of the evaluation metrics on the CFG scale used. We select a CFG scale of 2.5 as a balance between optimizing the gFID and CLIPScore metrics. Additionally this selected CFG scale is the same as the value used for the `FlexTok` based models enabling a balanced comparison of the two approaches.

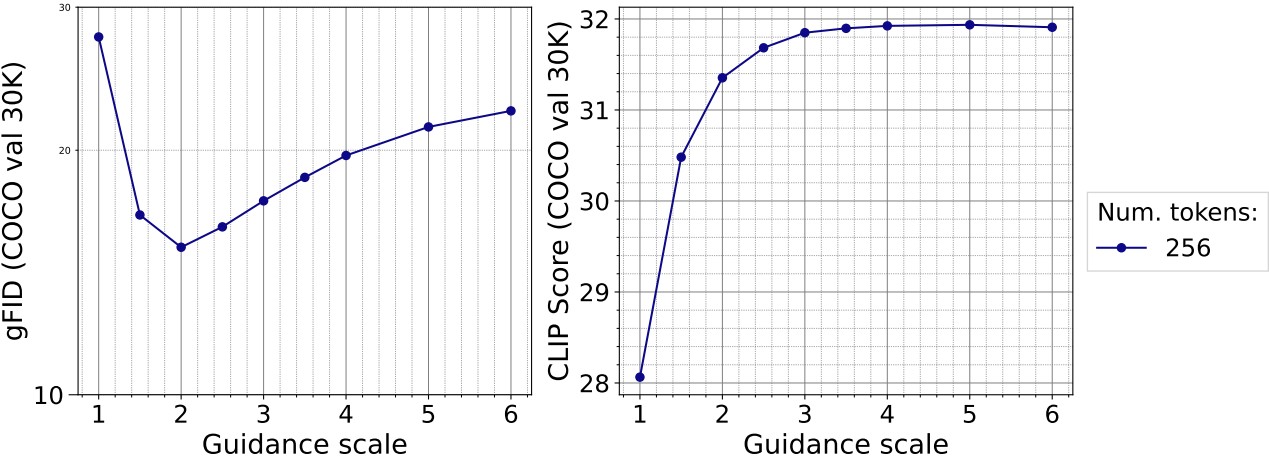

*Figure 30.* **Text-conditional 3.06B AR with 2D grid tokenizer guidance ablation.** gFID (left) and CLIPScore (right) metrics measured with respect to the COCO validation set at varying numbers of tokens.

## J. Autoregressive Text Conditional Image Generation Model Size

Following the optimal inference parameters observed in Appendix I, we evaluate the effect of scaling the AR model size while fixing the `FlexTok d18-d28` tokenizer. Scaling up the model size from 113M to 3.06B parameters results in lower final loss values being achieved for both the `FlexTok` and 2D grid tokenizer (Figure 31 and 32). When generating the full 256 tokens, scaling up the AR model size significantly improves the gFID values (Figure 33).

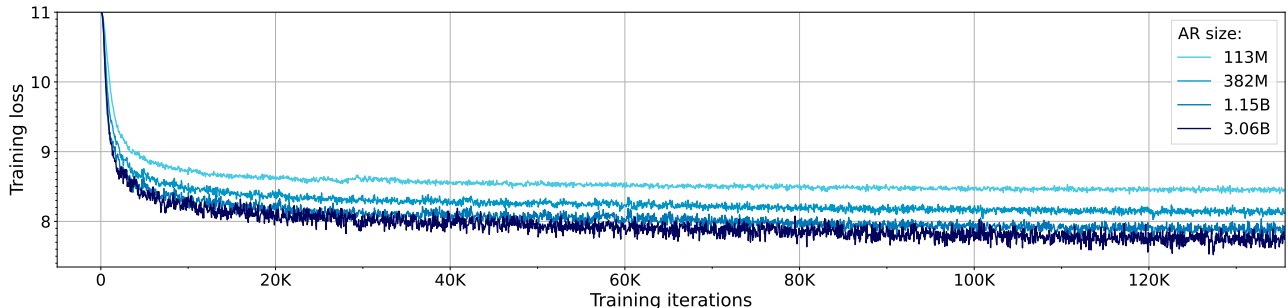

Figure 31. **`FlexTok` autoregressive text-conditional image generation loss curves.** The AR models of different sizes shown here are trained for 284B tokens on DFN using the `FlexTok d18-d28` tokenizer trained on DFN.

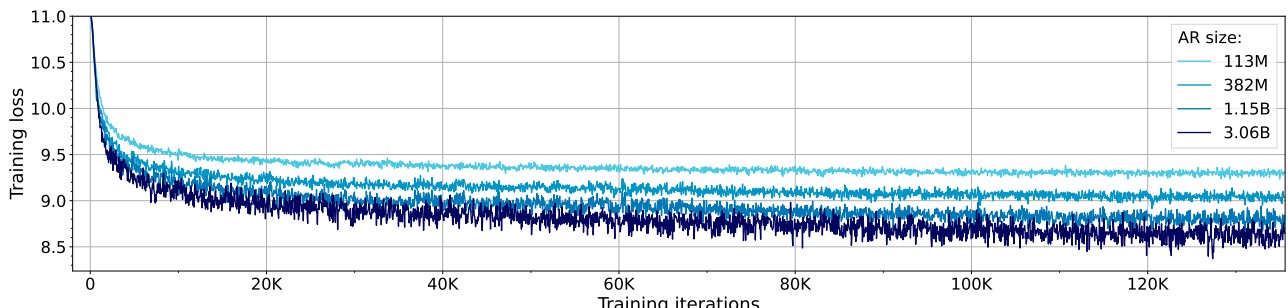

Figure 32. **2D grid tokenizer autoregressive text-conditional image generation loss curves.** The AR models of different sizes shown here are trained for 284B tokens on DFN using the `FlexTok d18-d28` tokenizer trained on DFN.

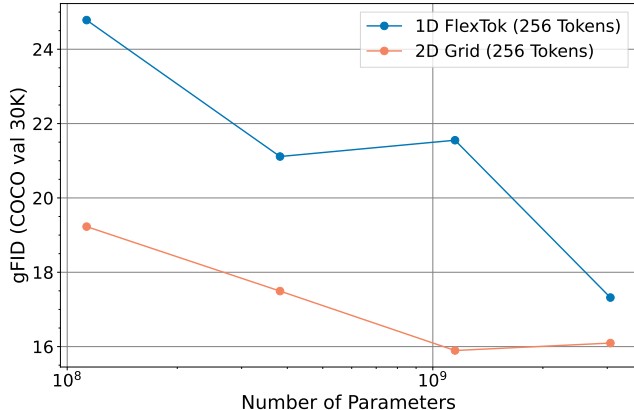

Figure 33. **Autoregressive text conditional image generation model scaling.** The gFID values for each model size as functions of the parameter count of the AR model. During AR generation we **generate 256 tokens so that the `FlexTok` and 2D grid tokenizer can be compared** and use the optimal inference parameters detailed in Appendix I. The gFID values are measured with respect to the COCO validation set. All AR models are trained using the tokenizers trained on DFN.

# K. Additional Visualizations

## K.1. **FlexTok** image reconstruction for different numbers of tokens – multiple samples per token sequence

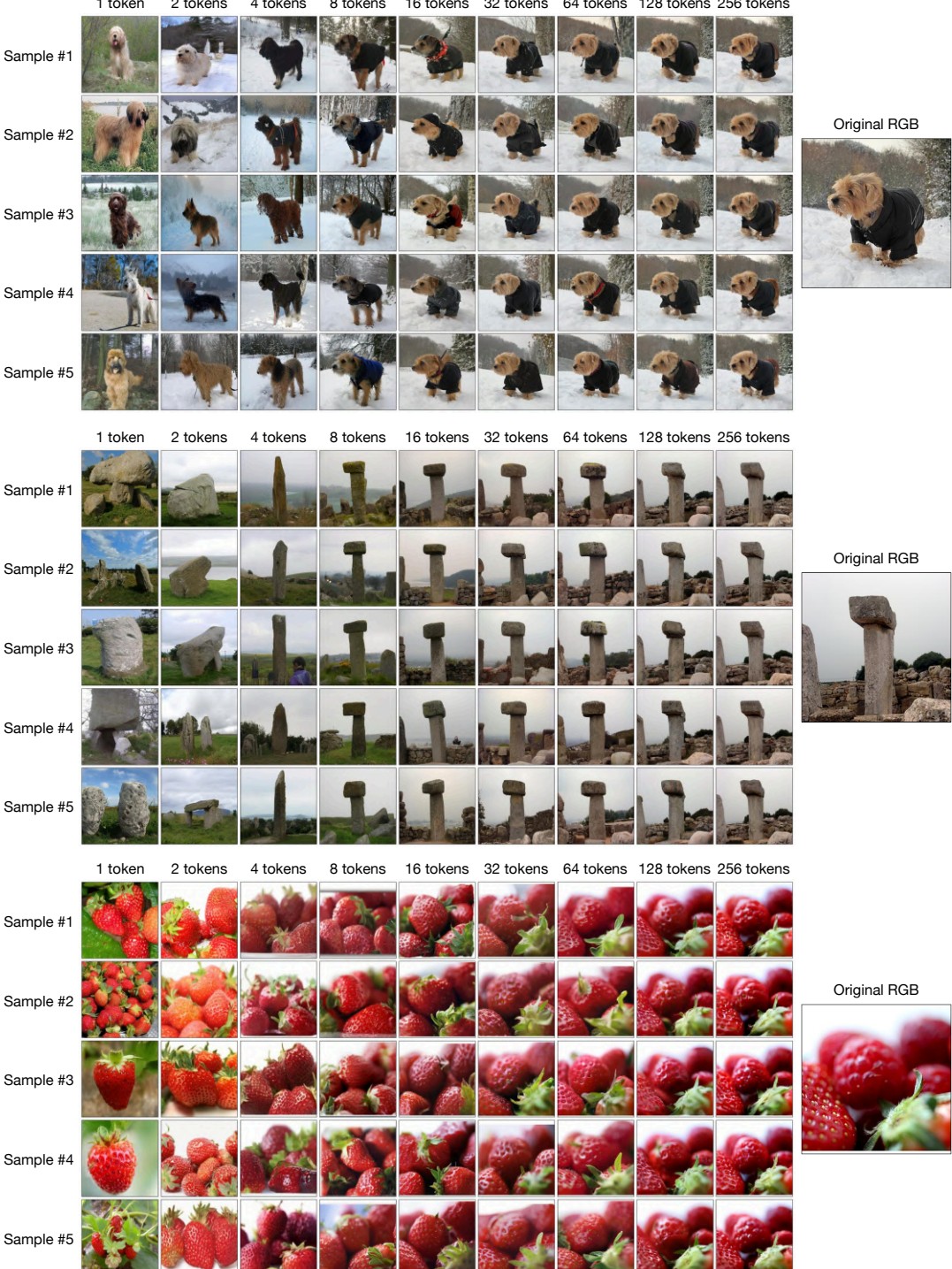

*Figure 34.* **FlexTok samples for different number of tokens.** We show `FlexTok d18-d28` (trained on ImageNet-1k) reconstructions for different numbers of tokens for ImageNet-1k validation set samples. We draw 5 random samples from the rectified flow decoder given the same token sequences. `FlexTok` token sequences define a distribution over images that gets narrower, and more specific to the original image, the more tokens are used.

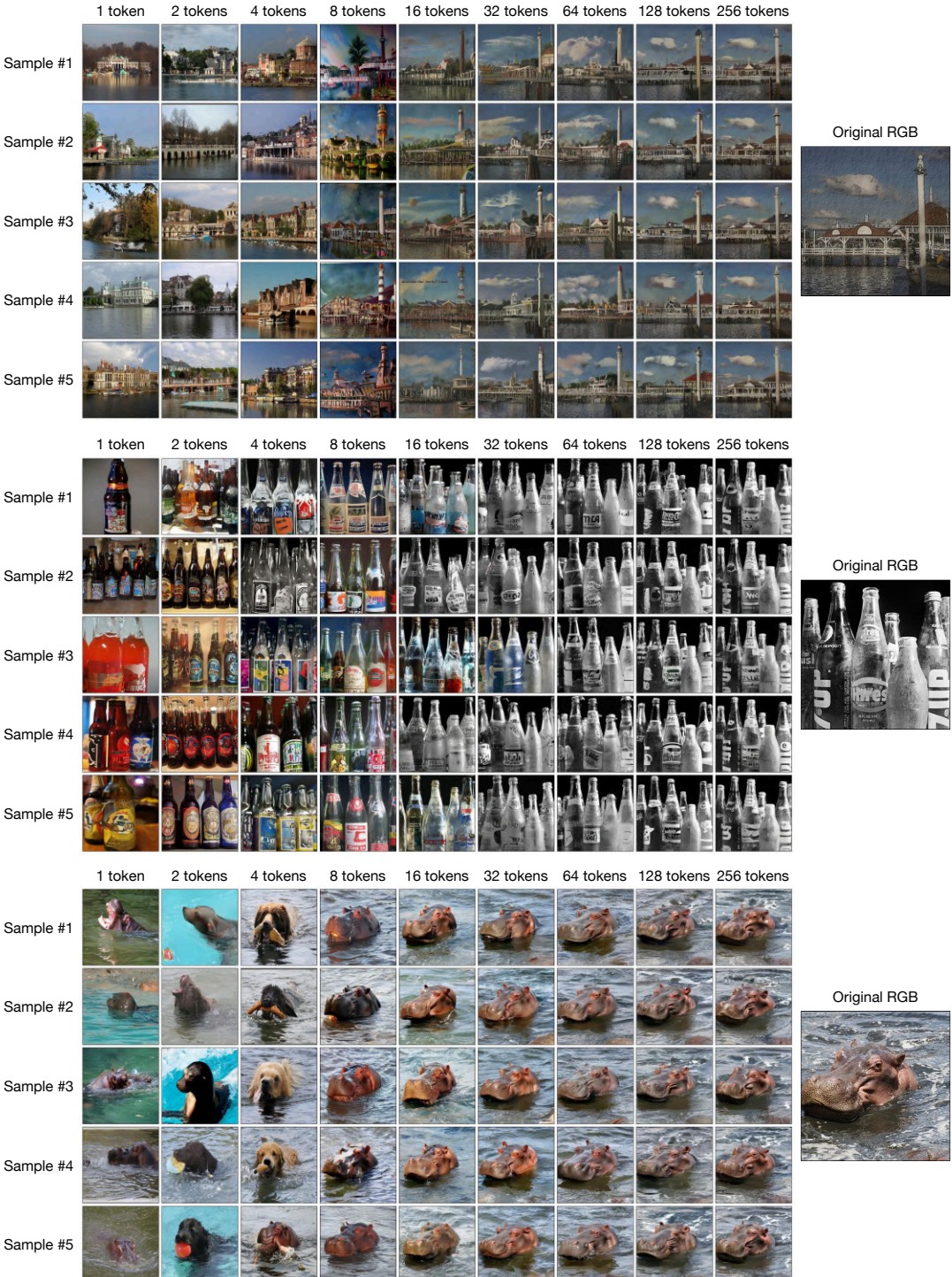

*Figure 35.* **FlexTok samples for different number of tokens.** We show `FlexTok d18-d28` (trained on ImageNet-1k) reconstructions for different numbers of tokens for ImageNet-1k validation set samples. We draw 5 random samples from the rectified flow decoder given the same token sequences. `FlexTok` token sequences define a distribution over images that gets narrower, and more specific to the original image, the more tokens are used.

## K.2. **FlexTok** image reconstruction for different numbers of tokens and tokenizer sizes

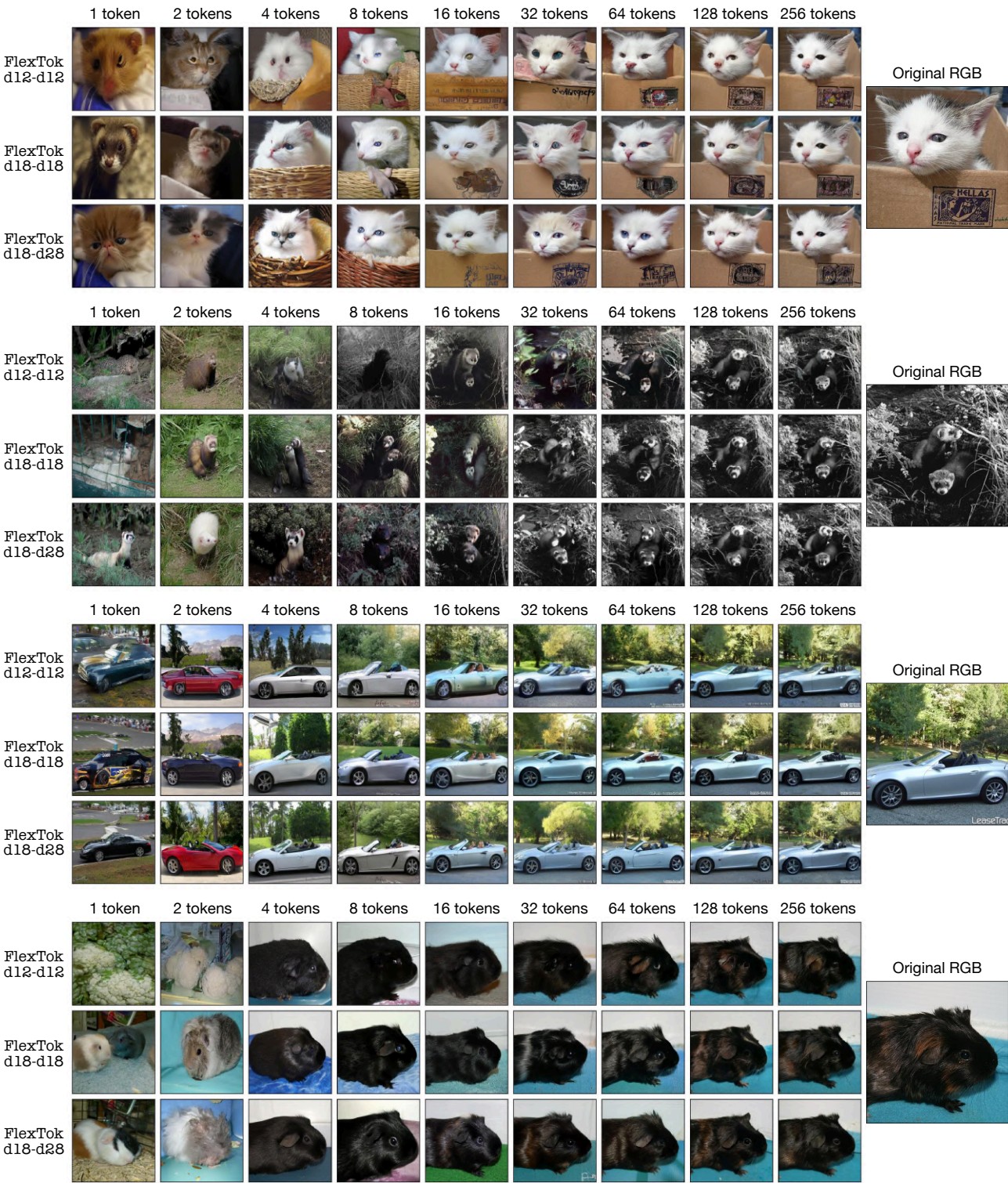

*Figure 36.* **FlexTok reconstructions for different numbers of tokens and model sizes.** We show `FlexTok` (trained on ImageNet-1k) reconstructions for different numbers of tokens and model sizes (`d12-d12`, `d18-d18`, `d18-d28`) for ImageNet-1k validation set samples.

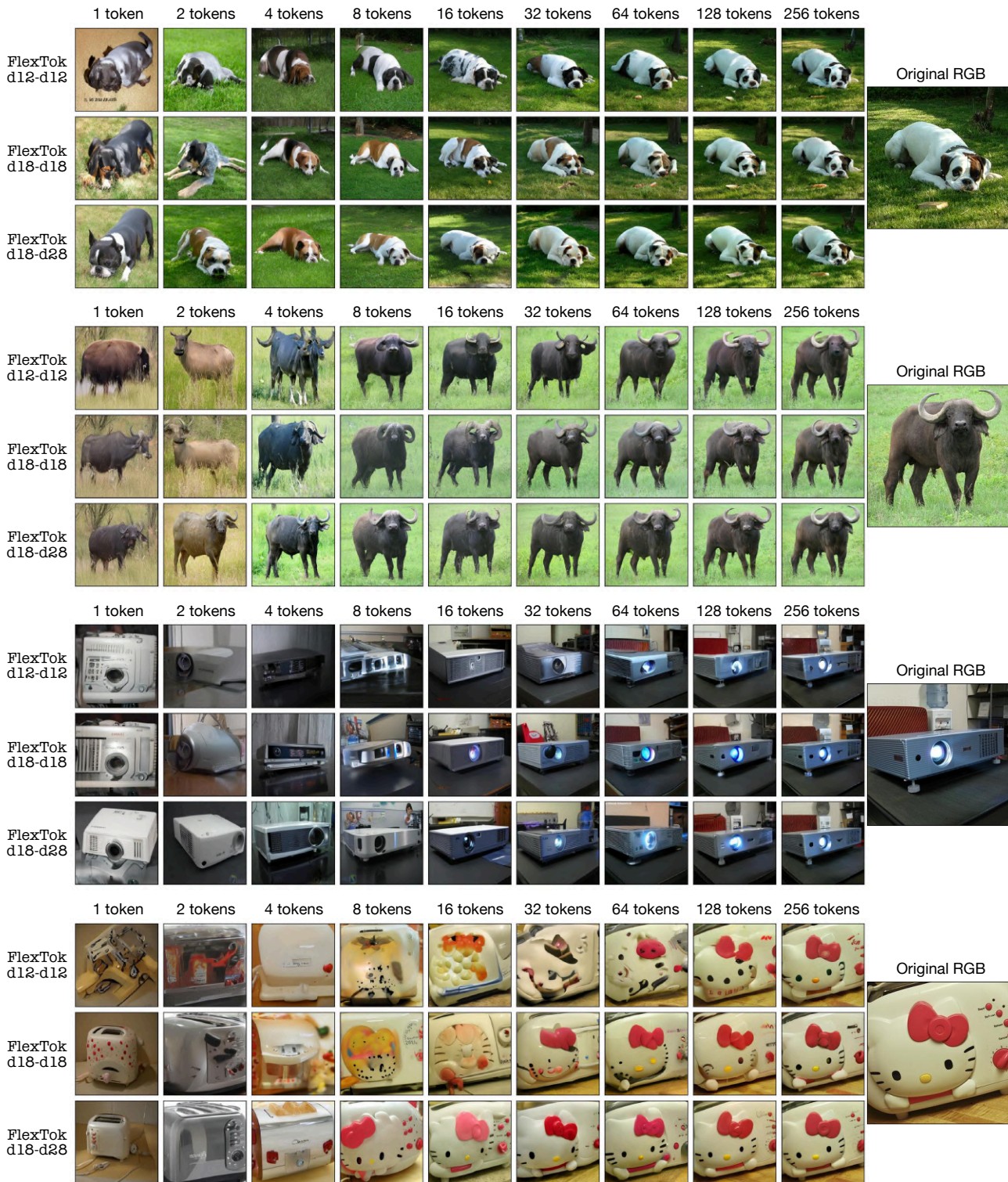

*Figure 37.* **FlexTok reconstructions for different numbers of tokens and model sizes.** We show `FlexTok` (trained on ImageNet-1k) reconstructions for different numbers of tokens and model sizes (`d12-d12`, `d18-d18`, `d18-d28`) for ImageNet-1k validation set samples.

### K.3. Image reconstruction comparison between **FlexTok**, TiTok, and ALIT

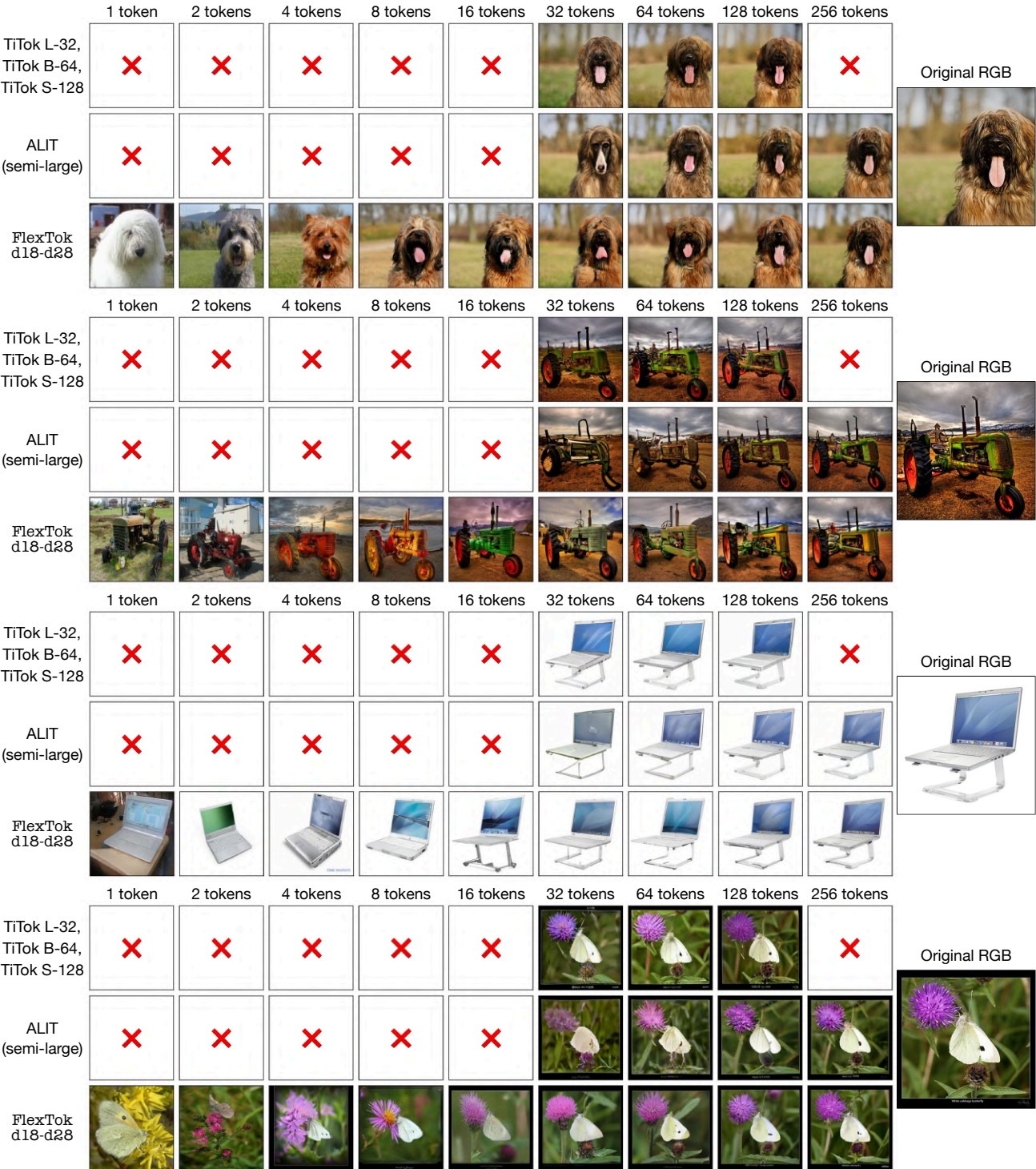

*Figure 38.* **Reconstruction comparison between FlexTok and baselines.** We show `FlexTok d18-d28` (trained on ImageNet-1k) reconstructions for different numbers of tokens and for ImageNet-1k validation set samples, and compare against three different TiTok (Yu et al., 2024a) models, and ALIT (Duggal et al., 2024).

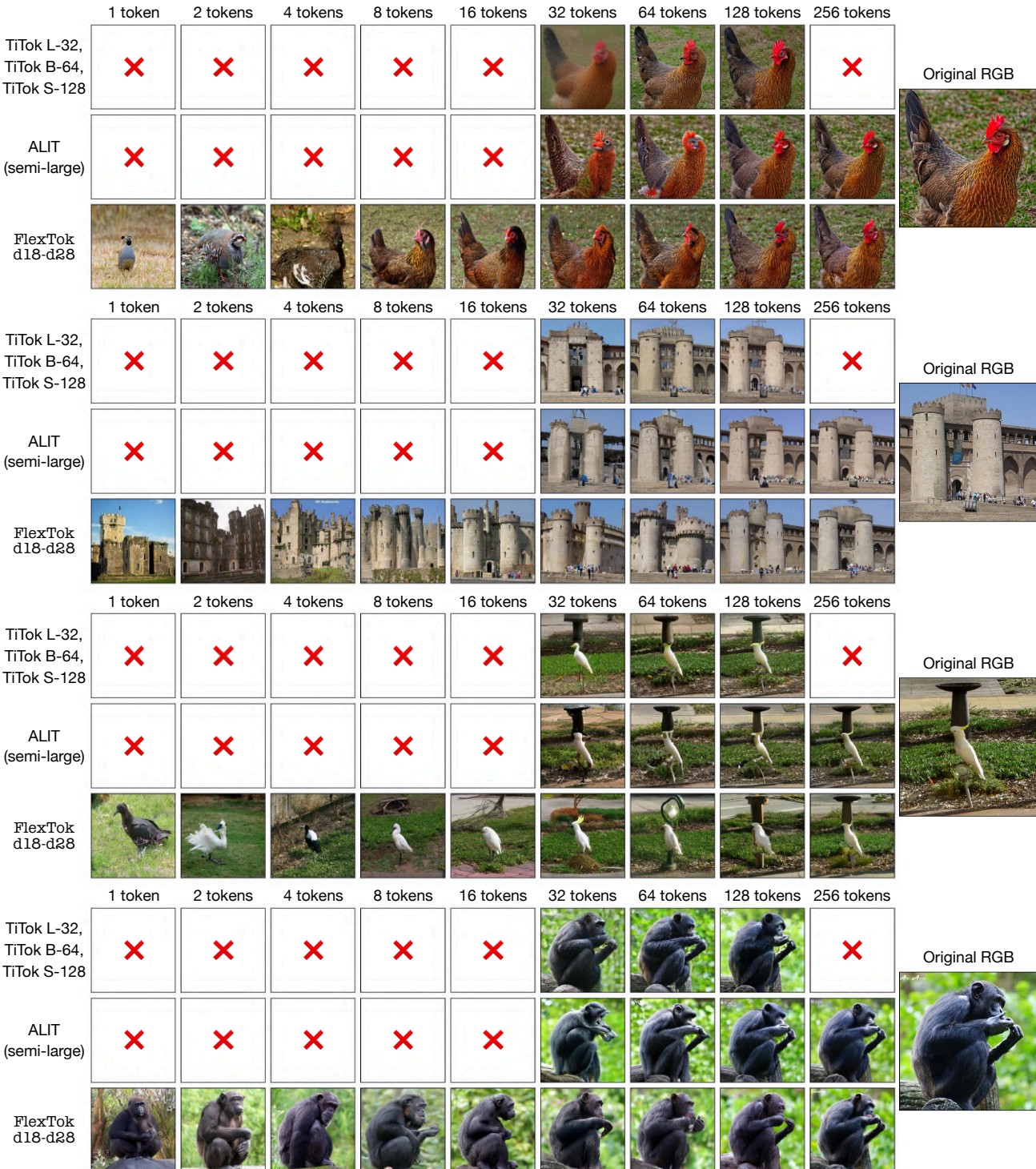

*Figure 39.* **Reconstruction comparison between `FlexTok` and baselines.** We show `FlexTok d18-d28` (trained on ImageNet-1k) reconstructions for different numbers of tokens and for ImageNet-1k validation set samples, and compare against three different TiTok (Yu et al., 2024a) models, and ALIT (Duggal et al., 2024).

## K.4. Class-conditional image generation visualizations

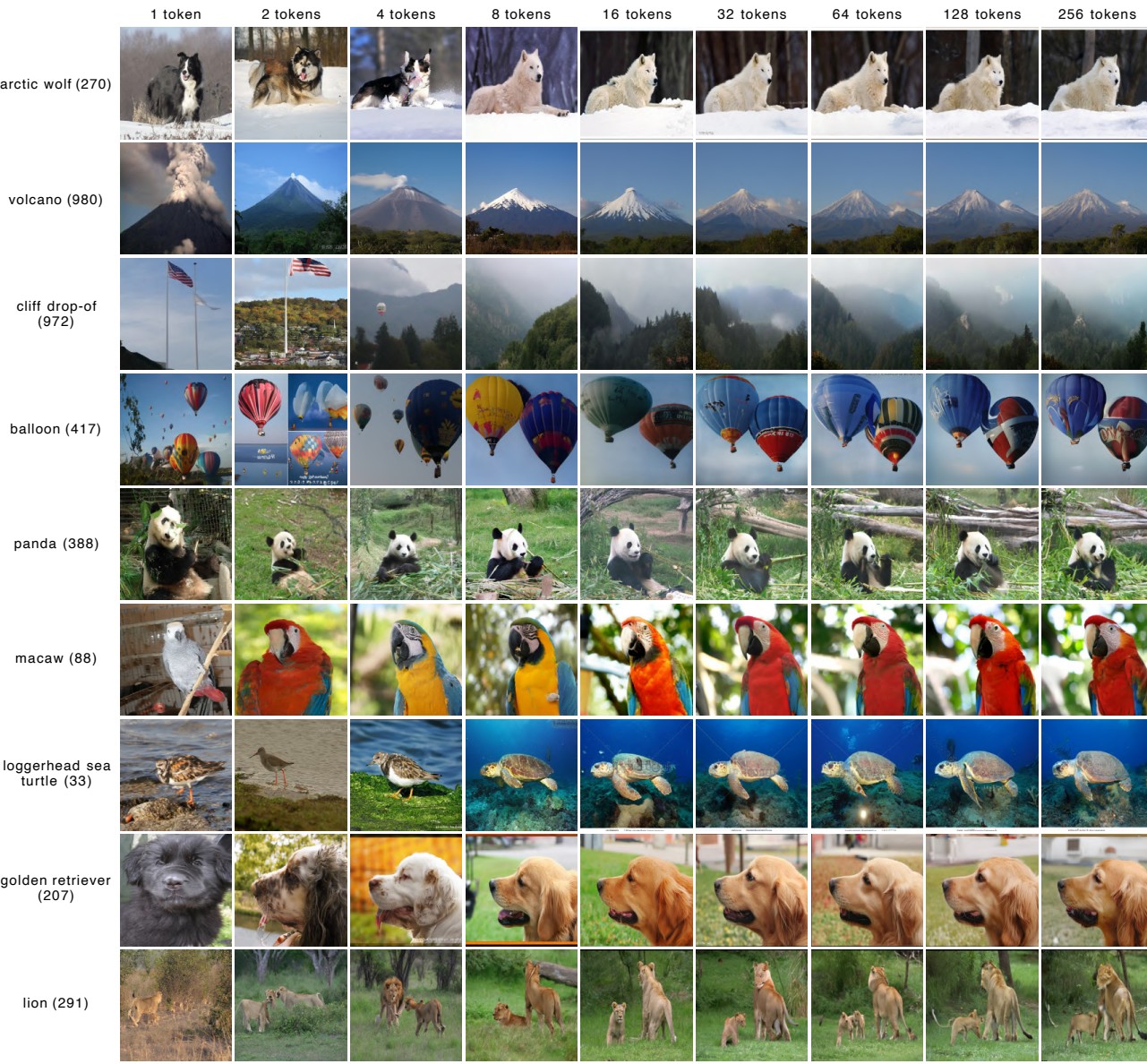

*Figure 40.* **FlexTok class-conditional image generations with varying numbers of tokens.** From left to right the number of tokens used increases in powers of 2 from 1 up to 256 tokens. Each row represents a different ImageNet-1k class index supplied to the auto-regressive image generator as conditioning. Images are generated using the `FlexTok d18-d28` tokenizer combined with a 1.33B parameter AR Transformer.

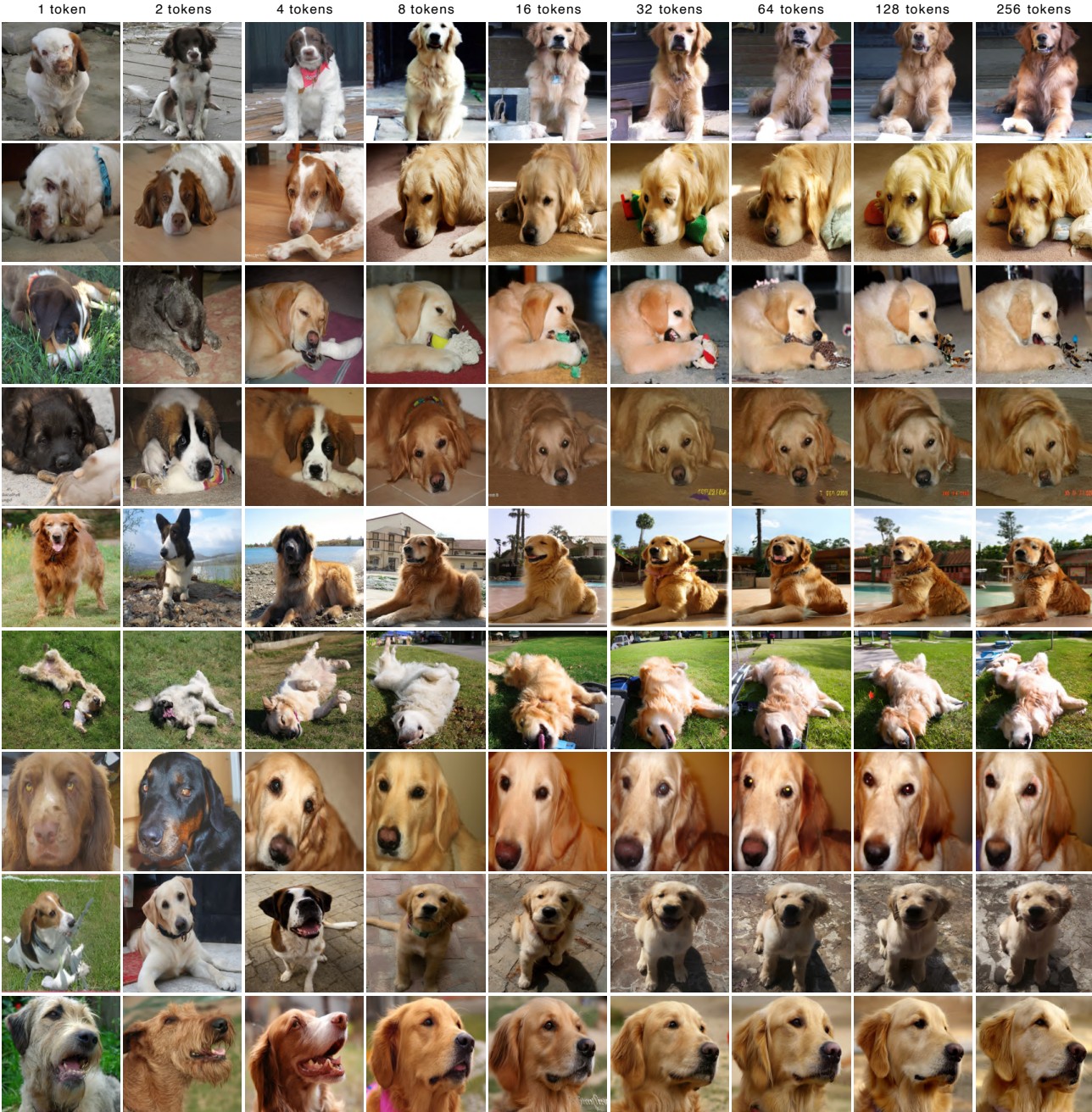

*Figure 41.* **FlexTok class-conditional image generations with varying numbers of tokens.** From left to right the number of tokens used increases in powers of 2 from 1 up to 256 tokens. Images are generated using the `FlexTok d18-d28` tokenizer combined with a 1.33B parameter AR Transformer. The model is conditioned on the class label from ImageNet-1k for "golden retriever" (207), using different random seeds for each row.

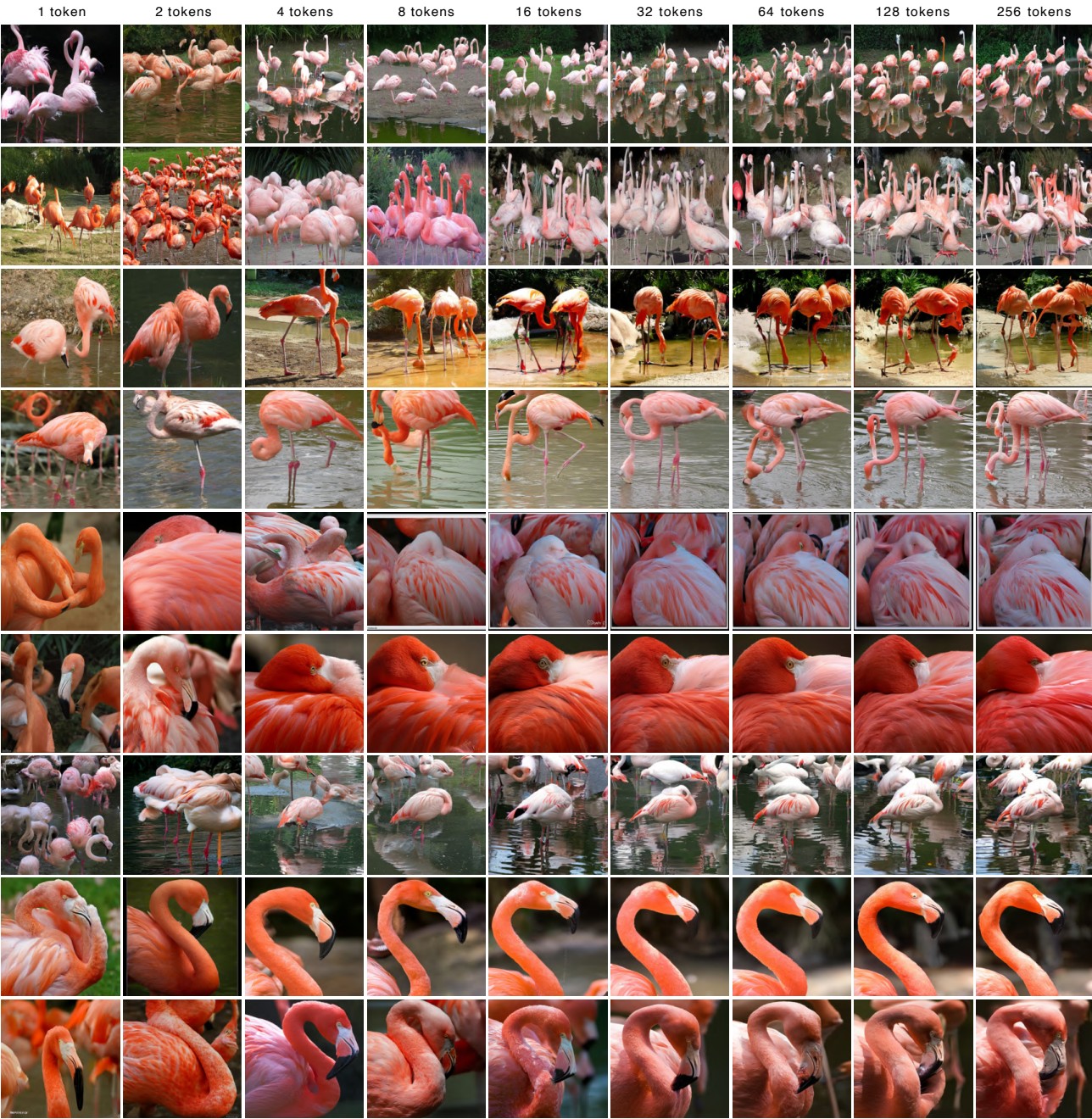

*Figure 42.* **FlexTok class-conditional image generations with varying numbers of tokens.** From left to right the number of tokens used increases in powers of 2 from 1 up to 256 tokens. Images are generated using the `FlexTok d18-d28` tokenizer combined with a 1.33B parameter AR Transformer. The model is conditioned on the class label from ImageNet-1k for "flamingo" (130), using different random seeds for each row.

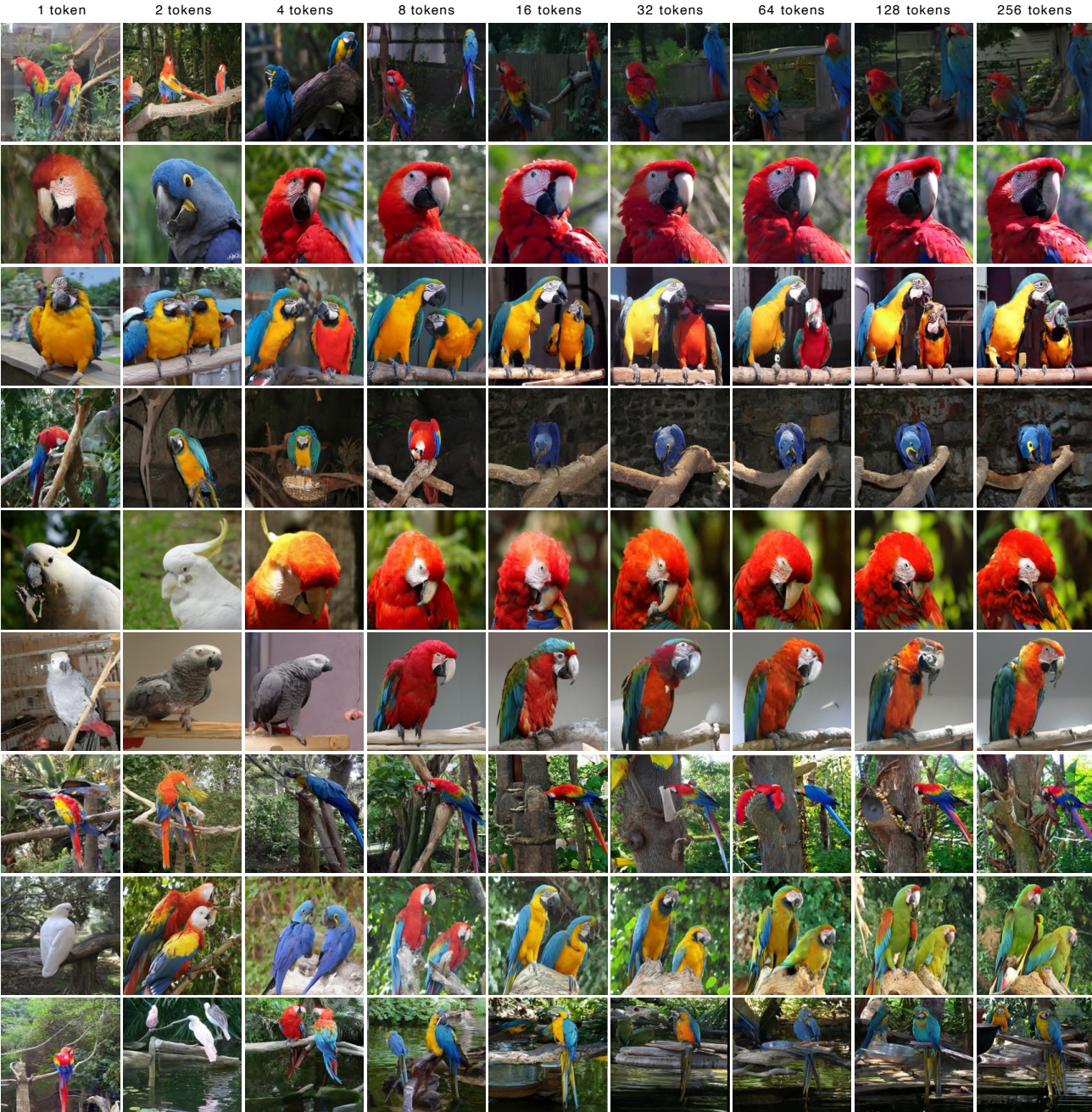

*Figure 43.* **FlexTok class-conditional image generations with varying numbers of tokens.** From left to right the number of tokens used increases in powers of 2 from 1 up to 256 tokens. Images are generated using the `FlexTok d18-d28` tokenizer combined with a 1.33B parameter AR Transformer. The model is conditioned on the class label from ImageNet-1k for "Macaw" (88), using different random seeds for each row.

## K.5. Text-conditional image generation visualizations

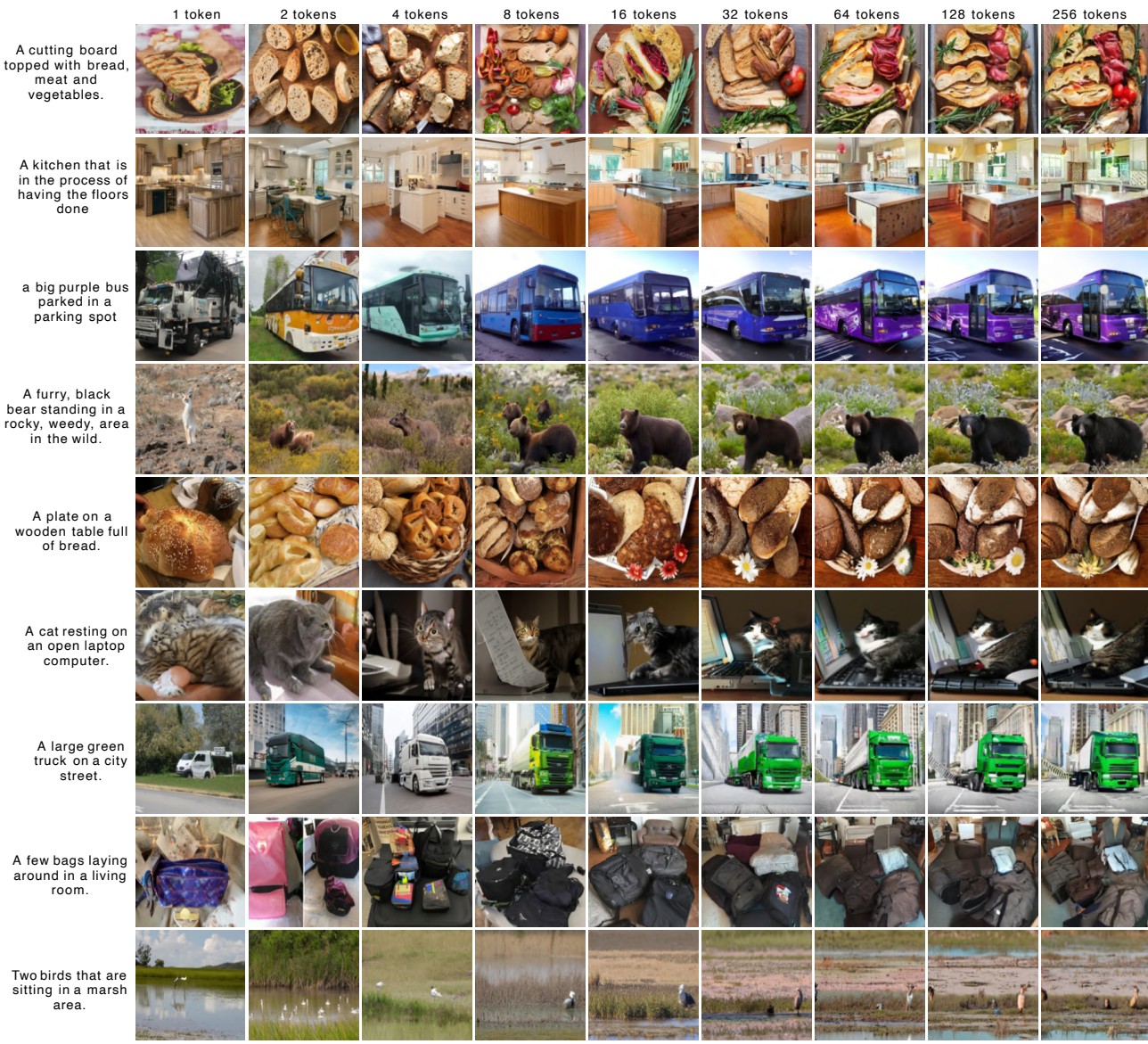

*Figure 44.* **FlexTok text-conditional image generations with varying numbers of tokens.** From left to right the number of tokens used increases in powers of 2 from 1 up to 256 tokens. Images are generated using the `FlexTok d18-d28` tokenizer trained on DFN combined with a 3.06B parameter AR Transformer also trained on DFN. For each row the AR Transformer is conditioned on the text embeddings of the prompts.

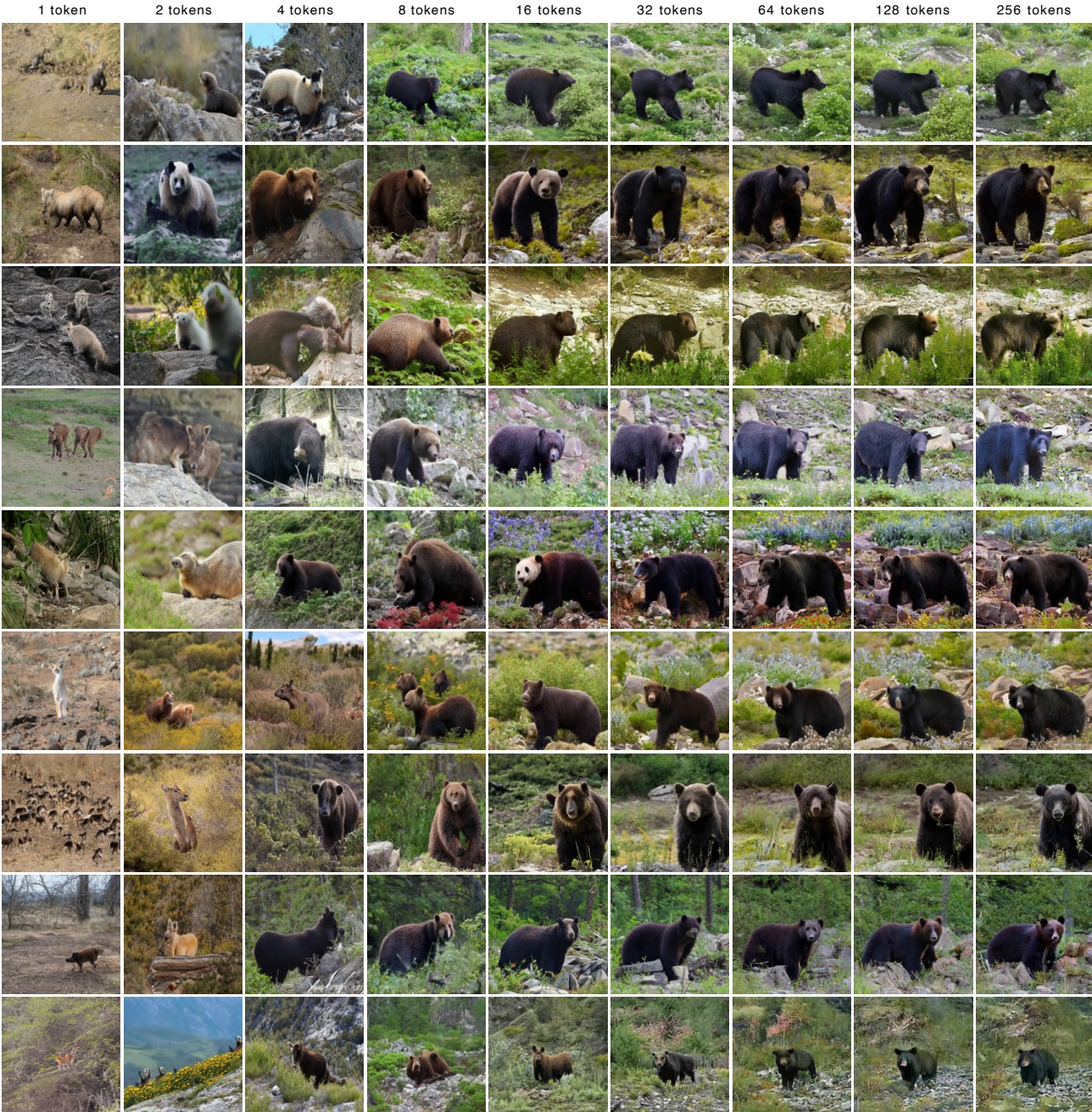

*Figure 45.* **FlexTok text-conditional image generations with varying numbers of tokens.** From left to right the number of tokens used increases in powers of 2 from 1 up to 256 tokens. Images are generated using the `FlexTok d18-d28` tokenizer trained on DFN combined with a 3.06B parameter AR Transformer also trained on DFN. For each row the AR Transformer is conditioned on the text embeddings of the prompt *"A furry, black bear standing in a rocky, weedy, area in the wild"*, but uses a different random seed.

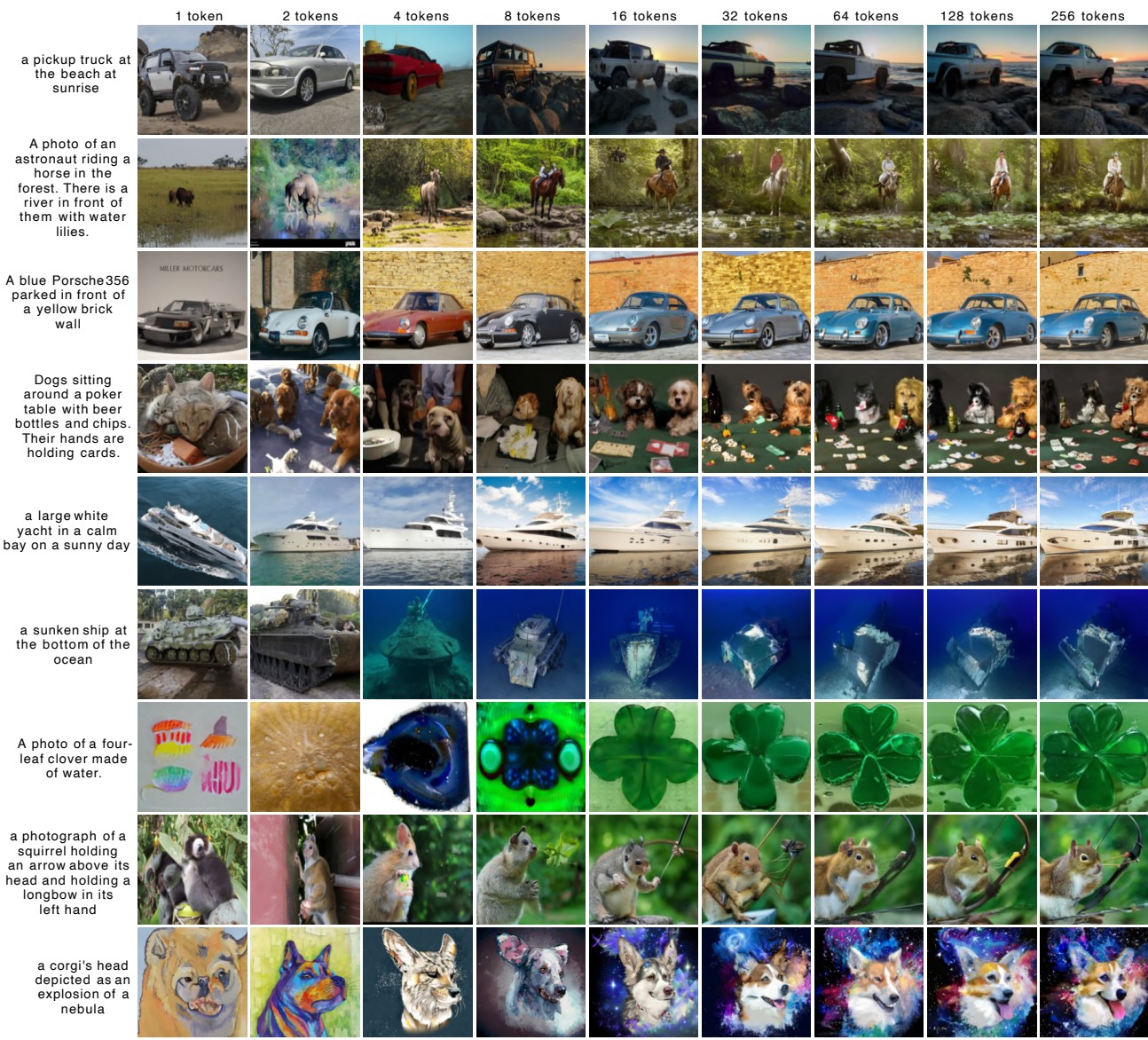

*Figure 46.* **FlexTok text-conditional image generations with varying numbers of tokens.** From left to right the number of tokens used increases in powers of 2 from 1 up to 256 tokens. Images are generated using the `FlexTok d18-d28` tokenizer trained on DFN combined with a 3.06B parameter AR Transformer also trained on DFN. For each row the AR Transformer is conditioned on the text embeddings of a different PartiPrompt (Yu et al., 2022).

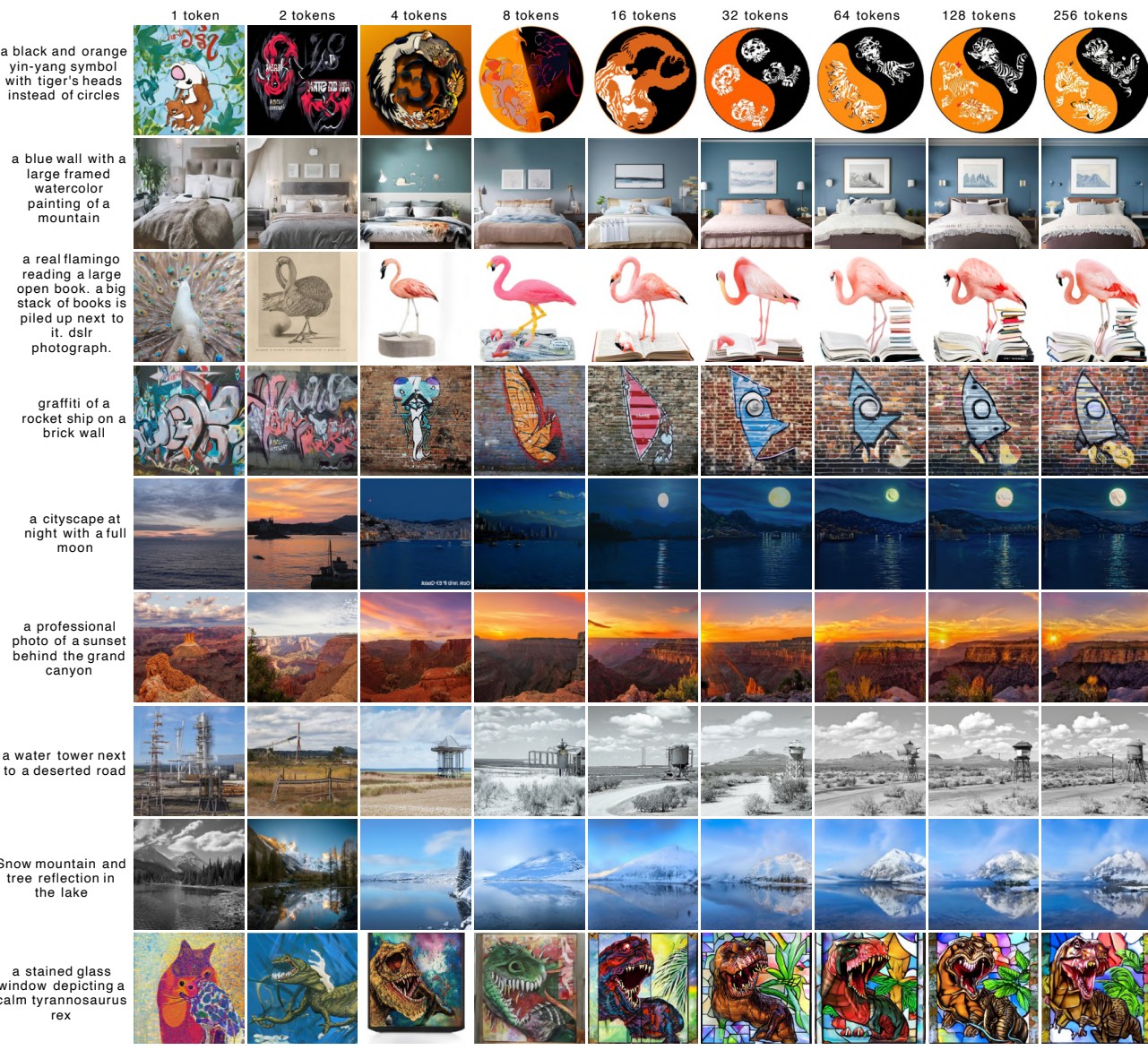

*Figure 47.* **FlexTok text-conditional image generations with varying numbers of tokens.** From left to right the number of tokens used increases in powers of 2 from 1 up to 256 tokens. Images are generated using the `FlexTok d18-d28` tokenizer trained on DFN combined with a 3.06B parameter AR Transformer also trained on DFN. For each row the AR Transformer is conditioned on the text embeddings of a different PartiPrompt (Yu et al., 2022).

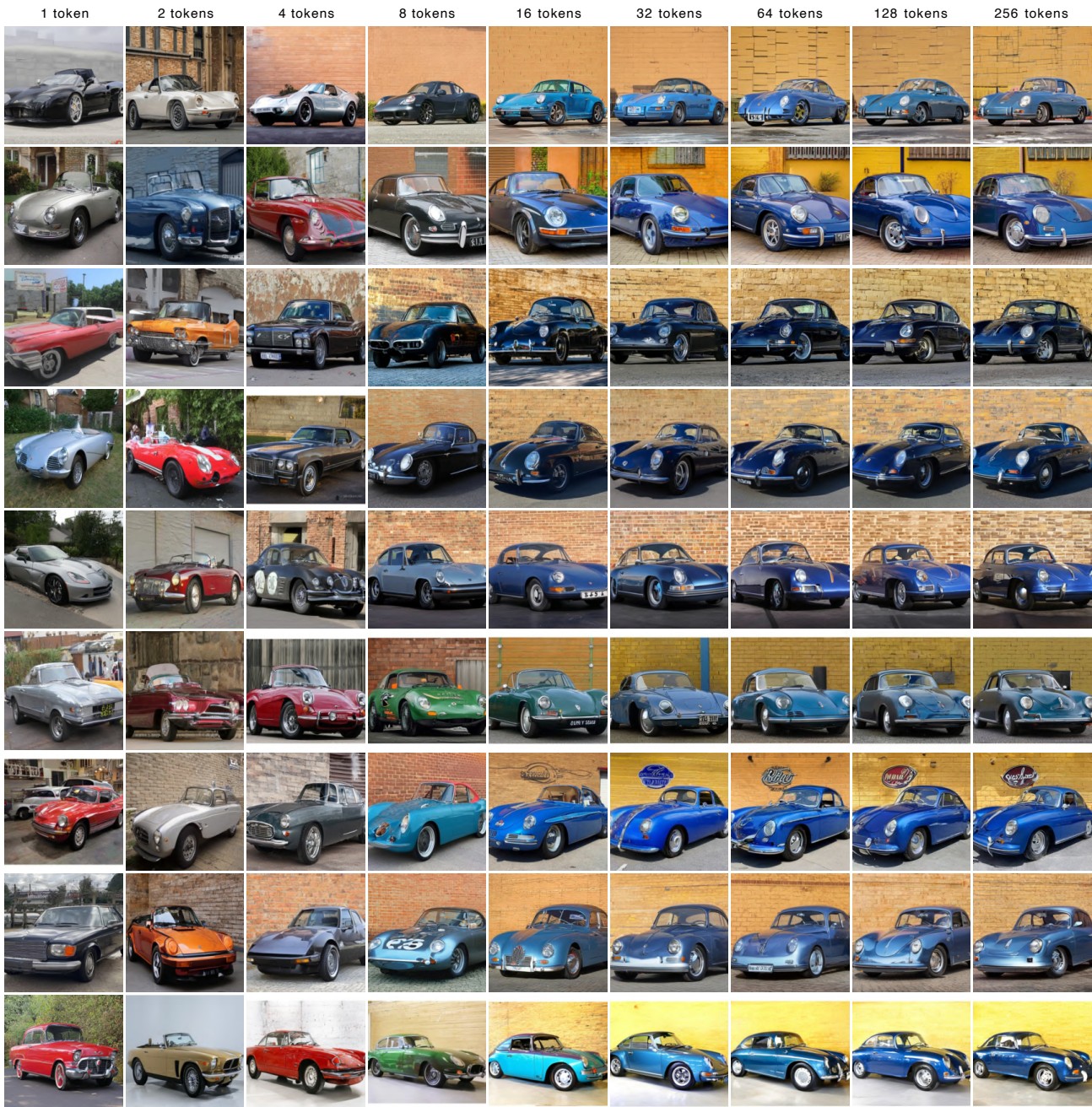

*Figure 48.* **FlexTok text-conditional image generations with varying numbers of tokens.** From left to right the number of tokens used increases in powers of 2 from 1 up to 256 tokens. Images are generated using the `FlexTok d18-d28` tokenizer trained on DFN combined with a 3.06B parameter AR Transformer also trained on DFN. For each row the AR Transformer is conditioned on the text embeddings of the prompt *"A blue Porsche 356 parked in front of a yellow brick wall"*, but uses a different random seed.

