# OpenReview forum: "FlexTok: Resampling Images into 1D Token Sequences of Flexible Length"
_ICML.cc/2025/Conference — ICML 2025 poster_

### Official Review · Reviewer_jx6L · 2025-02-26

**Overall Recommendation:** 3

**Summary:**

This paper presents a novel method for tokenizing an image into a one-dimensional token sequence, which allows for flexible image representation and processing. Most of the existing VAE/VQVAE methods employ quantization on 2D grids, thus the token size is proportional to the image size. This paper proposes a novel VQ method that resamples images of varying sizes into a fixed-size sequence. It can be combined with AR architecture for image generation.
This paper presents thorough experiments to analyze different modules and hyperparameters for the generation. From experiments, the paper achieves performance comparable to that of existing SOTA image generation methods.


## update after rebuttal
I thank the authors for their detailed response. Most of my concerns have been solved. However, as the reconstruction uncertainty and diffusion/AR sampling will affect the method's performance, the main paper should enclose a detailed discussion on them in the final version.  Considering several revisions are required, I will keep my initial rate.

**Claims And Evidence:**

1. The paper proposes the token numbers of previous methods are proportional to image size, but the method is independent of size and is affected by the complexity of the image. The claim is clear but it lacks experiments to support this. The method quantizes the image to 1-D sequence tokens, but all experiments are done on 256*256 size. The paper does not discuss: a) Does the image size affect the token sizes? For larger images, does it need more tokens for representation? b) How to define the complexity of the image? The paper lacks an experiment to analyze this. If an image' content is 'simple', does the method only require a fixed-size token to represent it no matter the size of the image?

2. FlixTok is an image tokenizer, which should losslessly compress and reconstruct an image. However, in Figure 3, when the tokens are less than 16, the reconstructed image is different from the original. It is normal the reconstruction should be worse with limited tokens, but the content should be similar to the original images. From this result, could it be called a tokenizer? It is more like a generation model.

**Essential References Not Discussed:**

All important references have been enclosed in the paper.

**Experimental Designs Or Analyses:**

1. The paper lacks the experiments present in the 'Claims And Evidence' part.

2. The FlexTok is an image tokenizer. Ideally, it should losslessly compress an image to tokens and then reconstruct it. In experiments, it only presents two figures to analyze the reconstruction. It lacks enough experiments and comparisons with other methods, especially 2-D grid tokenization methods. Furthermore, Figure 4, only presents rFiD, which can only analyze the GT and prediction distribution. PSNR can better compare the compression losses.

3. The decoder of FlexTok is a diffusion-type model. Does the initial noise affect the reconstruction quality and appearance during inference? Is this uncertainty in inference optimal for the reconstruction? Why not employ a VQGAN decoder and corresponding supervision? The paper should give more discussions.

4. In generation, it employs an AR architecture. The sampling strategy of AR (such as TopK and TopP) and diffusion tricks both affect generation quality and diversity. Which one is more important? The paper lacks a discussion on this.

**Methods And Evaluation Criteria:**

The problem is present clearly and the method can solve it.

**Other Comments Or Suggestions:**

No more comments and suggestions.

**Other Strengths And Weaknesses:**

All weaknesses have been present in'Experimental Designs Or Analyses' and 'Claims And Evidence'.

**Questions For Authors:**

No more questions.

**Relation To Broader Scientific Literature:**

No more specific contributions to a broader scientific literature.

**Theoretical Claims:**

The theoretical claims are clear and correct.

---

> ### Author Rebuttal · Authors · 2025-04-01
>
> We thank reviewer jx6L for the thoughtful and constructive feedback. Below we address the main points raised:
>
> **1. Relation between image size, complexity, and token count**
>
> This is a good point, and we haven't explored this explicitly yet. We performed all our experiments at 256x256 resolution specifically to enable direct comparison with standard methods (e.g., VQ-GAN, LlamaGen). While FlexTok could support variable-resolution inputs (e.g. using architectures like NaViT), that's a separate research direction we leave for future work. We also agree the interplay between image complexity, resolution, and token counts is an interesting open question.
>
> **2. “Lossless” compression**
>
> We appreciate this observation about reconstruction quality at low token counts. It's important to clarify that all image tokenizers are inherently lossy, just to different degrees. Some tokenizers (e.g., SEED (Ge et al. 2023)) focus purely on semantic features without pixel accuracy, while others (e.g., VQ-GAN) are more pixel-aligned. FlexTok sits in between, explicitly transitioning from semantic-level reconstructions at low token counts (<16 tokens) toward highly pixel-aligned reconstructions as token count increases (e.g. at 256 tokens). We explicitly measure pixel-level reconstruction fidelity using Mean Absolute Error (MAE), which steadily improves with more tokens (Fig. 4). Regarding the rFID metric, we agree that additional reconstruction metrics would strengthen our analysis; we'll include PSNR and SSIM results compared directly to baseline tokenizers like VQ-GAN (see the comparison in response to reviewer ekCv).
>
> **3. Why a diffusion-type decoder rather than VQGAN?**
>
> We chose a diffusion-type decoder (rectified flow) exactly because it models conditional uncertainty. When using few tokens, the degree of compression is high and reconstructions naturally have uncertainty; the diffusion decoder handles this gracefully, producing plausible, semantically coherent outputs rather than blurry averages. As the number of tokens increases, the uncertainty naturally reduces, making reconstructions progressively more accurate and deterministic (see image reconstructions in Appendix J.1). A VQGAN-style decoder wouldn't offer this flexible control over reconstruction uncertainty.
>
> **4. AR sampling vs. diffusion sampling impact**
>
> We found both AR sampling (top-k, top-p, temperature) and diffusion sampling parameters influence image quality and diversity. AR sampling tends to be stable across reasonable settings (see Appendix F), while diffusion decoding, particularly adaptive projected guidance (Sadat et al., 2024), had a significant impact on final image quality. In short: both matter, but diffusion guidance is particularly critical. We'll clarify this explicitly.

---

> > ### Comment · Reviewer_jx6L · 2025-04-02
> >
> > Thanks for the detailed rebuttal.
> > Another question on the uncertainty. As the VAE/VQVAE attempts to encode the image and reconstruct it. Although your method can use all tokens for reconstruction, your proposed diffusion decoder still encloses uncertainty in reconstruction. Different initial noise would result in a different image, maybe in some details. Have you compared variant initial-noise reconstruction? This part should be discussed in the paper.

---

> > > ### Author Response · Authors · 2025-04-02
> > >
> > > Thanks for the interesting follow-up question.
> > >
> > > To some extent, this reconstruction uncertainty is quantified through per-image reconstruction metrics such as MAE and DreamSim, measured between input images and their corresponding k-token reconstructions. As shown in Fig. 4, these metrics demonstrate a roughly log-linear improvement with increasing token counts, indicating that reconstructions become progressively closer to the original input and therefore necessarily more deterministic. This behavior is also visually apparent in the reconstructions provided in Appendix J.1. We expect that providing even more tokens as conditioning would further reduce reconstruction variance.
> > >
> > > To explicitly quantify the effect of initial noise variation, we conducted an additional experiment where we decoded identical token sequences 10 times using different random seeds and measured the average pairwise DreamSim similarity across reconstructions. We observed that reconstruction variability rapidly decreases with an increased number of conditioning tokens, highlighting that stronger conditioning signals lead to more deterministic outputs. We will include this analysis and discussion in the camera-ready version.

---

### Official Review · Reviewer_ekCv · 2025-03-04

**Overall Recommendation:** 4

**Summary:**

This paper introduces FlexTok, a novel 1D tokenizer that can encode images with variable token lengths. It combines casual masking and nested dropout in training to force the tokenizer to learn to reconstruct an image with a varying number of tokens. This strategy further promotes the tokenizer to encode images in a coarse-to-fine order, where the initial tokens encapsulate semantic and geometric concepts, and the subsequent tokens progressively capture finer details. To ensure reconstruction quality at extreme compression rates (e.g., using only 1-2 tokens), FlexTok employs a rectified flow model as its decoder. The method achieves strong performance in both reconstruction and generative tasks on ImageNet and COCO.

## update after rebuttal

Thank the authors for their detailed response. Considering that FlexTok shows diminishing improvements for long token sequences, which limits the method's upper bound, and considering the inefficiency issues raised by other reviewers, I will maintain my initial rating.

**Claims And Evidence:**

The claims in this paper supported by experimental results or prior studies.

**Essential References Not Discussed:**

N/A

**Experimental Designs Or Analyses:**

The experimental designs are solid and fair.

**Methods And Evaluation Criteria:**

The paper mainly uses rFID, MAE, and DreamSim for reconstruction quality evaluation, and leverages gFID, top-1 accuracy, and CLIP score for generation quality evaluation. These metrics are proper for the study of an visual tokenizer.

**Other Comments Or Suggestions:**

I think this is an overall good paper, but could benefit from further enhancing the flexibility in tokenization and generation. For example, the default setting of FlexTok only supports reconstruction for a predefined set of token lengths, rather than a truly random number of tokens. Besides, instead of having a predefined token length in generation, it would be more exciting to see the generator evolve to decide the number of tokens to generate, similar to how language models generate texts.

**Other Strengths And Weaknesses:**

**Strengths:**
- This paper studies a very important problem in image tokenization. The idea of 1D and variable length tokenization in FlexTok is novel and interesting, which enables images to be modeled in a similar way to language sequences.
- Compared to traditional 2D tokenizers and prior 1D tokenizers, FlexTok demonstrates improved reconstruction and generation performance, while using fewer number of tokens.
- The paper is well-written. The experiments are solid and comprehensive.

**Weaknesses:**
- It seems that variable length in the tokenization stage does not generalize to the generation stage. That is, although a single FlexTok tokenizer can handle variable length of tokens, the generator is limited to a fixed token length. As a result, it basically needs multiple generators to support variable length generation. This constrains the practical use of FlexTok in generation. (Maybe I have misunderstandings here, the authors can correct me.)
- FlexTok is relatively weak in long-sequence reconstruction and generation. In Figure 4, 6, 7, it can be seen that increasing the number of tokens beyond 32 has minor improvements in rFID, and even negatively impacts gFID. This may be attributed to the trade-off in performance between low number and high number of tokens, as shown in Appendix Table 4. FlexTok adopts the “Pow2” dropout strategy, which is favored for short token length but under-samples long sequences in training.

**Questions For Authors:**

1. From my experience, 1D tokenizers like TiTok are more adept at 'semantic-level' reconstruction than 'pixel-level' reconstruction. Therefore, having a low rFID does not guarantee that the tokenizer faithfully reconstruct the images. I wonder whether FlexTok suffers from the same problem. Could the authors provide a comparison with other tokenizers on the PSNR and SSIM metrics?

2. As shown in Appendix Figure 10, REPA greatly accelerates the convergency of FlexTok training. I am curious about whether FlexTok could achieve similar performance without REPA but longer training schedules. It would be better if the authors could provide some visualizations of the reconstruction results without REPA.

**Relation To Broader Scientific Literature:**

In recent literatures, there is a growing interest in 1D tokenizers which improve computational efficiency (e.g., encoding an image with fewer tokens) while maintaining competitive quality. Prior works like TiTok are fixed in token length. FlexTok takes a step forward to enable variable length in tokenization. Besides, by incorporating advancements from tokenizers with rectified flow decoding, FlexTok further reduces the minimum token length from 32 to 1.

**Theoretical Claims:**

There is no proof or theoretical claim in this paper.

---

> ### Author Rebuttal · Authors · 2025-04-01
>
> We thank reviewer ekCv for the thoughtful and constructive feedback. Below we address the main points raised:
>
> **1. Variable-length generation limitation (fixed token length in generator)**
>
> To clarify: in our current setup, we train a single autoregressive (AR) model capable of generating a full 256-token sequence. During inference, shorter outputs are obtained simply by early stopping this sequence at the desired length. While effective, you're right that it could be even more flexible if the AR model itself determined when to halt generation, and this is an interesting future research direction. We expect that we can create such halting conditions by augmenting the training data with per-token-subsequence reconstruction metrics (e.g., MSE, DreamSim), and when training the stage 2 AR model, simply truncating the training sequences if the score reaches a certain pre-defined threshold.
>
> **2. Weakness at longer token sequences (limited improvement beyond ~32 tokens)**
>
> The reviewer raises an insightful point about long-sequence performance. Indeed, our "Pow2" nested dropout schedule intentionally emphasizes shorter sequences during training. This design optimizes FlexTok’s ability to reconstruct images effectively at extreme compression (few tokens), a core contribution of our work. However, it is correct that this strategy results in diminishing improvements for longer sequences (>32 tokens). Adjusting dropout sampling strategies to more evenly balance short and long sequences could potentially mitigate this trade-off, and we appreciate the suggestion here.
>
> **3. Semantic-level vs. pixel-level reconstruction (PSNR/SSIM comparisons)**
>
> We agree, rFID is not a good measure for pixel alignment. We measure rFID to demonstrate that the FlexTok decoder is capable of producing outputs that could "plausibly" come from the image distribution, no matter the number of tokens given. To measure pixel-level reconstruction alignment, we show MAE and DreamSim in Fig 4, and observe a roughly log-linear relationship between the scores number of tokens. We additionally show PSNR and SSIM in the table below for various number of tokens, and find that at 256 tokens used, FlexTok reaches comparable compression performance to common 2D-grid tokenizers that use 16x16 discrete tokens.
>
> FlexTok d18-d28 reconstruction metrics on IN1K validation set, resolution 256x256:
>
> | # Tokens | PSNR | SSIM |
> | --- | --- | --- |
> | 1 | 9.35 | 0.187 |
> | 2 | 10.25 | 0.222 |
> | 4 | 11.51 | 0.254 |
> | 8 | 11.90 | 0.269 |
> | 16 | 13.05 | 0.304 |
> | 32 | 13.96 | 0.330 |
> | 64 | 14.34 | 0.343 |
> | 128 | 15.90 | 0.407 |
> | 256 | 17.70 | 0.489 |
>
> Comparison with common discrete tokenizer baselines (numbers from Cosmos paper):
>
> | Model | # Tokens | PSNR | SSIM |
> | --- | --- | --- | --- |
> | Open-MAGVIT2 | 16x16 | 17.00 | 0.398 |
> | LlamaGen | 16x16 | 18.38 | 0.338 |
> | Cosmos-0.1 | 16x16 | 20.49 | 0.518 |
>
> **4. Impact of REPA vs. longer training schedules**
>
> This is a good question, however, the difference in convergence speed is so significant that we found it computationally too expensive to ablate this. We expect that the ~17x convergence speedup (in terms of FID) demonstrated in the original REPA paper may roughly translate to our setting too. Unfortunately we are unable to add qualitative examples to this text-only reply, but we find the non-REPA reconstructions (after training for the same number of steps) to be significantly worse in terms of fidelity, and overall less semantic. We will add a discussion of these points as well as qualitatives to the camera-ready.

---

### Official Review · Reviewer_6zZ5 · 2025-03-13

**Overall Recommendation:** 4

**Summary:**

In this paper, the authors introduce a tokenizer that maps 2D images into variable-length, ordered 1D token sequences. This tokenizer allows images to be represented with a flexible number of tokens based on their content. In addition, an autoregressive model leverages this approach to achieve high-quality generation results with fewer image tokens. The authors conduct extensive experiments to validate the effectiveness of the proposed method.

## update after rebuttal
I appreciate the authors' thorough response and the effort they put into the rebuttal. However, I still have concerns regarding the limitations of using a fixed number of register tokens, as well as the lack of a comprehensive system-level comparison for text-conditional image generation. Therefore, I am keeping my score unchanged.

**Claims And Evidence:**

Strengths
+ The claims regarding the limitations of existing generative models are correct and wildly recognized in the field of generative models.
+ The proposed method is reasonable  and easy to understand. It encodes images with variable-length tokens based on image complexity.

Weaknesses

None

**Essential References Not Discussed:**

To my knowledge, there is no other works that should be discussed.

**Experimental Designs Or Analyses:**

Strengths
+ The superior results on both class-conditonal and text-conditional image generation demonstrate the effectiveness of the proposed method. This paper also provides exhaustive ablation studies to assess each key component.
+ The qualitative results across different numbers of tokens make the efficiency of FlexTok more apparent and intuitive.

Weaknesses
- It is better to show system-level comparison on text-conditional image generation, not only the ablation studies in the main text.
- For class-conditional generation, Inception score (IS), Precision, and Recall are the primary evaluation metrics, which is widely used in the generative field. Thus, it is essential to include them in this paper.

**Methods And Evaluation Criteria:**

Strengths
+ The proposed method is both intuitive and reasonable. It is an effective technique to employing rectified flow decoder to alleviate the blurry reconstruction caused by fewer register tokens. The nested dropout and causal attention masks also benefit the learning of visual vocabulary and AR generation.
+ The evaluation benchmark is reasonably appropriate for assessing the effectiveness of the proposed method.

Weaknesses
- Compared to the decoders used in existing tokenizers, the rectified flow decoder requires higher computation costs, due to its extensive denoising steps.
- The pre-defined maximum number of register tokens may not suitable for extremely complex images, and determining an optimal setting remains an open and difficult problem.

**Other Comments Or Suggestions:**

It would be beneficial to provide a summary of the appendix content at the beginning.

**Other Strengths And Weaknesses:**

None

**Questions For Authors:**

I acknowledge the novelty of the proposed method, despite its limitations regarding the pre-defined number of register tokens and the computation cost of the decoder. Thus, I am inclined to rate this paper as accept.

**Relation To Broader Scientific Literature:**

To my knowledge, the proposed method in this paper is new.

**Theoretical Claims:**

This paper does not include any theoretical claims.

---

> ### Author Rebuttal · Authors · 2025-04-01
>
> We thank reviewer 6zZ5 for the thoughtful and constructive feedback. Below we address the main points raised:
>
> **1. Higher computational cost of rectified flow decoder and token-count limitations**
>
> Please see our response to reviewer MMdz. Our preliminary experiments suggested that higher token counts can significantly reduce reconstruction errors, and enable decoding with fewer steps, however, for simplicity of training subsequent autoregressive models on the token sequences, we decided to choose 256 tokens as the upper-bound for this submission.
>
> **2. System-level comparison for text-to-image generation**
>
> We fully understand the motivation for requesting comparisons against external text-to-image baselines. However, unlike the somewhat standardized class-conditional ImageNet setting, proper comparison on text-to-image generation with external baselines is usually extremely difficult and nuanced due to differences in compute and dataset (size, caption quality, diversity, aesthetics, similarity to COCO, etc.) having significant impact on downstream evaluations (e.g. see "On the Scalability of Diffusion-based Text-to-Image Generation", Li et al. 2024). For that reason we decided to perform a controlled experiment in which we train a 2D grid tokenizer with the same data, compute, and rectified flow decoder objective, and perform autoregressive generation on its tokens. The results in Fig. 6 suggest that FlexTok performs comparably to classical 2D grid tokenizers at 256 tokens, but offers more flexibility overall.
>
> **3. Additional standard metrics for class-conditional generation**
>
> We appreciate the suggestion and provide the requested metrics (Inception Score, Precision, Recall, and gFID) in the table below. Our results are comparable to state-of-the-art methods such as VAR-d30 (323.1 IS, 0.82 precision, 0.59 recall, 1.92 gFID) and TiTok (gFID between 1.97 and 2.77 depending on tokenizer choice) across a broad range of token counts. We will incorporate these results into the camera-ready version of the paper.
>
> | #Tokens | IS | Precision | Recall | gFID |
> | --- | --- | --- | --- | --- |
> | 1 | 236.47 | 0.83 | 0.53 | 3.14 |
> | 2 | 238.07 | 0.82 | 0.57 | 2.51 |
> | 4 | 226.77 | 0.80 | 0.60 | 2.00 |
> | 8 | 266.48 | 0.82 | 0.61 | 1.82 |
> | 16 | 277.45 | 0.82 | 0.61 | 1.75 |
> | 32 | 284.99 | 0.82 | 0.61 | 1.71 |
> | 64 | 286.40 | 0.82 | 0.61 | 1.76 |
> | 128 | 275.63 | 0.82 | 0.61 | 1.89 |
> | 256 | 258.33 | 0.80 | 0.61 | 2.45 |
>
> **4. Additional suggestion: summary of appendix content**
>
> Thanks for the helpful suggestion. Adding a concise appendix summary at the start is indeed beneficial. We will add this to the camera-ready version.

---

### Official Review · Reviewer_MMdz · 2025-03-14

**Overall Recommendation:** 3

**Summary:**

The paper proposes FlexTok, a method for improving the tokenizer (VAE compression) used in image generation frameworks. Like previous approaches (TiTok, ALIT), FlexTok compresses 2D images into 1D tokens initialized as learnable registers. These tokens interact with encoded image patch tokens via attention mechanisms. Unlike TiTok and ALIT, FlexTok’s decoder is trained with a flow matching loss rather than standard reconstruction loss, enabling multi-step denoising decoding. Additionally, FlexTok introduces a unique causal attention masking structure: encoded image patches attend only among themselves and not to registers, whereas register tokens attend to all patches but follow a causal pattern among themselves (the i-th register attends to j-th registers only if i ≥ j). The model further uses token-dropping techniques during training, similar to ElasticTok, allowing variable-length token representations with earlier tokens representing more general information and later tokens representing details. The second-stage generative model, based on an autoregressive approach, progressively generates finer-grained images as more tokens are used.

## update after rebuttal
I am willing to update my score to weak accept since ALIT and ElasticTok are concurrent works. However, my concerns about reconstruction quality, the inference cost, and the motivation are not fully addressed.

**Claims And Evidence:**

The authors claim significant benefits from their method, particularly highlighting their capability of generating images using even a single token. However, several of these claims are problematic:

- **Single-token generation claim:**
  The authors assert that their model can encode an image into just one token. However, this claim is unfair or misleading, as the decoding process itself is multi-step flow matching (a generative process rather than a faithful reconstruction). Thus, the single-token encoding is effectively used as a conditioning input similar to class or text tokens, rather than a compressed latent representation.

- **Reduction of compute or acceleration claim:**
  Due to the iterative, generative nature of the decoder (25-step flow matching), this method requires significantly more computational resources. This contradicts one key motivation for employing latent-based generative models, which typically aim at computational efficiency.

- **Token-level faithfulness:**
  Reconstruction quality of individual images (crucial for downstream tasks such as editing or conditional generation) appears limited. For instance, Figure 3 shows significant discrepancies between generated images and ground truth, even with 256 tokens (e.g., misaligned dog tails), indicating poor pixel-level or even patch-level alignment.

**Essential References Not Discussed:**

No essential missing references identified.

**Experimental Designs Or Analyses:**

Key weaknesses identified in experimental design:

- Lack of adequate evaluation of single-image reconstruction fidelity. Given the latent-based approach, individual reconstruction quality is crucial but is largely overlooked.
- The experiments presented (particularly in Figure 3) clearly illustrate significant quality gaps even at relatively high token counts (256 tokens), undermining claims of efficient and faithful representation.

**Methods And Evaluation Criteria:**

The methods and evaluation criteria are reasonable in the context of image generation research. However, the experimental evaluation has critical limitations:

- Reconstruction quality at the single-image level is not sufficiently addressed or emphasized.
- Claims of single-token generation and compression efficiency are misleading due to the generative, iterative decoder.

**Other Comments Or Suggestions:**

The authors should clarify their claims, explicitly distinguishing their method as employing an additional generative decoding process rather than true latent compression. More focus should be placed on improving reconstruction fidelity and clearly discussing limitations inherent to the proposed approach.

**Other Strengths And Weaknesses:**

**Strengths:**

- Causal attention mask is a novel modification.
- Interesting concept of flexible token lengths with importance ordering.

**Weaknesses:**

- Misleading or unfair claims regarding single-token generation and compression effectiveness.
- Decoder complexity and computational overhead contradict original VAE motivations (speed and efficiency).
- Poor single-image reconstruction fidelity limiting applicability to editing or conditional generation tasks.
- Limited overall technical novelty given strong reliance on prior works (TiTok, ALIT, DALL-E 3).

**Questions For Authors:**

1. Could you clarify why you claim that encoding images to a single token is feasible or meaningful, given your decoder itself is a multi-step generative model rather than a direct reconstruction?

2. Given the computational overhead introduced by iterative flow matching decoding (25 steps), how do you justify the additional complexity against typical VAE motivations (speed, compression)?

3. Reconstruction fidelity appears severely limited even at relatively high token counts (e.g., 256 tokens). Could you provide deeper analysis or experimental insights into how your method might address pixel-level or patch-level misalignments, which are crucial for tasks like conditional generation or editing?

4. Apart from the novel causal attention masking, could you clearly summarize the distinctive technical contributions of FlexTok beyond existing works such as TiTok, ALIT, and the diffusion-based decoding approaches found in DALL-E 3?

**Relation To Broader Scientific Literature:**

FlexTok builds upon existing literature, notably TiTok, ALIT, and ElasticTok, which have previously introduced concepts of variable-length, learnable register tokens. The flow matching decoding strategy aligns closely with ideas presented earlier (e.g., OpenAI's DALL-E 3), where diffusion-based decoders was used (see Consistency Decoder). Thus, the technical novelty of FlexTok appears limited, with incremental advances primarily in the causal attention masking strategy.

**Theoretical Claims:**

The paper does not present explicit theoretical claims or analyses.

---

> ### Author Rebuttal · Authors · 2025-04-01
>
> We thank reviewer MMdz for the thoughtful and constructive feedback. Below we address the main points raised:
>
> **1. Single-token generation claim**
>
> Image tokenization is commonly performed with lossy autoencoders that abstract away imperceptible information, meaning all tokenizer decoders (whether trained with diffusion/flow methods or GANs) are inherently generative to some extent. The degree to which they must be "generative" directly corresponds to the amount of compression. At the coarsest level (single FlexTok token = 2 bytes), our representation necessarily operates at a highly semantic level, conceptually similar to semantic tokenizers like SEED (Ge et al. 2023). At the finest level (256 tokens = 512 bytes), we achieve compression comparable to classical 2D grid tokenizers like VQ-GAN (also, see response to reviewer ekCv). FlexTok's unique strength is providing a single model that learns these different hierarchies, effectively offering an alternative way to describe images in a coarse-to-fine manner. The single-token scenario is indeed best interpreted as semantic conditioning rather than pixel-level compression.
>
> **2. Computational overhead (25-step flow matching decoder)**
> We acknowledge that FlexTok's rectified flow decoder adds computational complexity during inference (though encoding remains efficient). We explicitly chose this architecture because it consistently maintains high reconstruction fidelity across a wide range of token counts (Fig. 4). Importantly, decoding happens after the autoregressive (AR) generation step, which is already computationally intensive, so the flow decoding step adds a constant overhead rather than introducing an entirely new computational bottleneck. We also anticipate that common distillation methods (e.g., consistency decoders, Reflow) can substantially lower inference complexity.
>
> **3. Token-level faithfulness (single-image reconstruction quality)**
> Regarding reconstruction quality at 256 tokens (512 bytes), it's important to recognize that this still represents extremely high compression: a full-color image compressed to just 512 bytes will naturally show some loss of detail. This token count was explicitly chosen to match standard tokenizers (e.g., VQ-GAN, LlamaGen), allowing direct and fair comparisons. At this standard token count, FlexTok achieves reconstruction fidelity comparable to these established methods (see the comparison in response to reviewer ekCv). The visible imperfections at this compression rate are expected, and our results show a clear log-linear trend (Fig. 4), strongly suggesting that higher token counts (e.g., 1024 tokens or more) would yield increasingly faithful reconstructions. This trade-off between compression and fidelity is fundamental to all image tokenization approaches, and exploring higher-token-count scenarios is a natural next step.
>
> **4. Technical novelty compared to TiTok, ALIT, ElasticTok, and DALL-E 3**
> Regarding FlexTok's novelty beyond causal attention masking: while ElasticTok and ALIT indeed share similar concepts, these methods were developed concurrently and independently. FlexTok directly builds on TiTok's 1D tokenizer framework and diffusion/flow-matching ideas, but introduces several critical new technical components: specifically, causal attention masking combined explicitly with nested token dropout and rectified flow decoding. Crucially, these innovations collectively enable hierarchical tokenization, smoothly transitioning from coarse, semantic-level representations to detailed pixel-level reconstructions within one tokenizer. This hierarchical tokenization was not shown by TiTok or other prior methods. Additionally, we provide a detailed analysis of this hierarchical behavior in generative settings, clearly showing its practical benefits and trade-offs.

---

> > ### Comment · Reviewer_MMdz · 2025-04-07
> >
> > Thanks to the authors for their detailed response.
> >
> > First, I went back and checked the ALIT and ElasticTok papers, and indeed these were published within three months of the ICML deadline, making them concurrent works. Given this, I’m happy to withdraw my earlier concerns regarding novelty.
> >
> > However, I still have doubts regarding the practical utility and the intended application scenario of this work:
> > - If the authors’ definition of "image tokenization" relaxes the strict reconstruction requirement, then it would make sense to compare your approach with semantic tokenizers such as CLIP or DINO, specifically evaluating downstream tasks like image understanding.
> > - On the other hand, if the authors intend to compare against generative model VAEs, maintaining faithful reconstruction becomes critical—but as clearly shown, your current method struggles in this aspect.
> > - Regarding the two-stage generation approach, while using a single token might initially seem to improve efficiency, the second-stage diffusion decoder actually shifts the computational cost from the first stage to the second. In my view, this resembles more of a two-stage cascaded generation rather than genuinely improved efficiency.
> >
> > Considering your clarifications and my concerns above, I can raise my rating to a weak accept, but the motivation behind this work remains somewhat unclear to me.

---

> > > ### Author Response · Authors · 2025-04-08
> > >
> > > We thank the reviewer for their thoughtful follow-up and the reconsideration of novelty given the concurrent publication timeline of ALIT and ElasticTok.
> > >
> > > Regarding the remaining points about practical utility and application scenarios:
> > >
> > > **1. Semantic vs. pixel-level reconstruction**
> > >
> > > Our method strives for reconstructions as faithful as possible given the inherent information bottleneck defined by token count. At low token counts (1-16 tokens), reconstructions necessarily capture high-level semantics due to extreme compression (as low as 2 bytes per image). At higher counts (up to 256 tokens/512 bytes), reconstructions naturally become more detailed. All discrete tokenizers operate under similar trade-offs, optimizing reconstruction quality within information constraints rather than guaranteeing pixel-perfect fidelity. Our metrics (e.g. MAE, DreamSim) transparently quantify this trade-off, showing clear improvement with increased token counts.
> > >
> > > While comparing with semantic vision encoders like CLIP or DINO for image understanding could be interesting, this direction is intentionally out of scope for our paper. Like most discrete tokenizer literature, our primary focus is on generative tasks where discrete representations particularly excel. As a future direction, exploring our hierarchical representations for image understanding could indeed be valuable (potentially without the discrete bottleneck if generation isn't also needed).
> > >
> > > **2. Faithful reconstruction & generative performance**
> > >
> > > Our strong performance on established benchmarks like ImageNet FID demonstrates that FlexTok's reconstructions effectively support practical generative tasks. The information bottleneck inherent in extreme compression (2-512 bytes per image) naturally affects pixel-level fidelity, but this is true of all tokenization methods. What distinguishes FlexTok is its ability to operate across this entire spectrum within a single model, allowing users to choose the appropriate trade-off between semantic-level and pixel-level representation based on their specific use case.
> > >
> > >
> > > **3. Computational efficiency & two-stage generation**
> > >
> > > We agree with the reviewer that our approach can be viewed as a two-stage cascaded generation, and the computational cost does shift between stages depending on token count. Our paper explores this trade-off, showing how fewer tokens (where the AR model does less work and the flow decoder does more) can be sufficient for simpler conditioning scenarios like class labels, while more complex conditioning like detailed captions benefits from additional tokens. We believe this flexibility offers practical utility in adapting to different generation tasks, though we agree the flow decoder adds computational overhead that could be optimized in future work.
> > >
> > > We hope these clarifications address the main concerns and better convey the motivations behind our work.

---

### Decision · Program_Chairs · 2025-05-01

**Decision:**

Accept (poster)

**Comment:**

The paper proposes a method for improving the tokenizer used in image generation frameworks. The decoder is trained with a flow matching loss, which enables multi-step denoising decoding.

The reviewers found the flexible token lengths interesting, and the evaluation reasonable despite the shortcomings in the reconstruction. There were some concerns raised during the rebuttal, but the authors' discussion resolved most of them. Minor concerns, such as the motivation for the work, remained open.

Despite the remaining concerns, the reviewers recommend the acceptance of the paper. Thus, I also recommend it.